# LOWER-LEVEL DUALITY BASED PENALTY METHODS FOR NONSMOOTH BILEVEL HYPERPARAMETER OPTIMIZATION

## ABSTRACT

Hyperparameter optimization (HO) is a critical task in machine learning and can be naturally formulated as bilevel optimization (BLO) with nonsmooth lower-level (LL) problems. However, many existing approaches rely on smoothing strategies or sequential subproblem solvers, both of which introduce significant computational overhead. To address these challenges, we develop a penalization framework that exploits strong duality of the LL problem and its dual. Building on this, we design first-order single-loop projection-based algorithms to solve the penalized problems efficiently. Our methods avoid smoothing and off-the-shelf solvers, thereby greatly reducing per-iteration complexity and overall runtime. We provide rigorous convergence guarantees and analyze the stationary conditions of BLO with nonsmooth LL problems under penalty perspective. Through extensive numerical experiments on a variety of benchmark and real-world tasks, we demonstrate the efficiency, scalability and superiority of our method over existing BLO algorithms.

## 1 INTRODUCTION

Hyperparameter optimization (HO) arises in many diverse fields, including neural architecture search [1; 2; 3], feature learning [4], ensemble models [5], semi-supervised learning [6] and sample-weighting schemes [7; 8; 9; 10]. The hyperparameters control model complexity, stability and convergence and they need to be chosen externally. A poor choice can cripple performance, whereas good ones greatly enhance accuracy, robustness and generalization. **Regularization** is a common way to guide hyperparameter tuning, especially in regression and classification [11]. By incorporating a penalty term into the empirical risk, regularization balances data fitting with model complexity, thereby mitigating overfitting, which can be formulated as

$$\min_{\mathbf{x}} \ l(\mathbf{x}) + \sum_{i=1}^{M+1} \lambda_i R_i(\mathbf{x}), \tag{1}$$

where $l(\mathbf{x})$ represents loss functions and $\boldsymbol{\lambda} = (\lambda_1, \lambda_2, ..., \lambda_{M+1})$ encompasses hyperparameters. Meanwhile, $R_i(\mathbf{x}), i = 1, 2, ..., M + 1$ denotes regularizers related to norms, which can be categorized as follows:

$$R_i(\mathbf{x}) = \|\mathbf{x}\|_{(i)}, \ i = 1, 2, ..., M, \ R_{M+1}(\mathbf{x}) = \frac{1}{2}\|\mathbf{x}\|_2^2. \tag{2}$$

For each $i$, $\|\cdot\|_{(i)}$ represents a specific norm, such as the $\ell_1$, $\ell_2$, $\ell_\infty$, $\ell_{1,2}$ norm for vectors, the spectre or nuclear norm for matrices, or other commonly used norms, most of which are nonsmooth.

Based on formulation (1), the training/validation approach optimizes parameters on the training set while evaluating error on the validation set. This can be cast as a bilevel optimization (BLO) framework [3; 12], which has shown strong empirical performance [13; 14; 4; 15]. Formally,

$$\min_{\mathbf{x}\in\mathbb{R}^{d_x}, \boldsymbol{\lambda}\in\mathbb{R}_+^{M+1}} \ L(\mathbf{x}) \quad \text{s.t. } \mathbf{x} \in \arg\min_{\hat{\mathbf{x}}} \left\{ l(\hat{\mathbf{x}}) + \sum_{i=1}^{M+1} \lambda_i R_i(\hat{\mathbf{x}}) \right\}, \tag{3}$$

where $L, l, R_i : \mathbb{R}^{d_x} \to \mathbb{R} \cup \{+\infty\}$ are proper, closed functions. In this BLO, the lower-level (LL) variable $\mathbf{x}$ is the parameter to learn, and the upper-level (UL) variable $\boldsymbol{\lambda}$ is hyperparameter. This form naturally arises in structural risk minimization, which is one of the most common and foundational frameworks in machine learning. The LL base learner determines the optimal hypothesis for a given hyperparameter, while the UL problem selects the hyperparameter–hypothesis pair minimizing the given criteria on the validation set. Representative examples include elastic net [16], sparse group Lasso [17], low-rank matrix completion [18], smoothed support vector machine (SVM) [19; 20], robust regression [21; 22].

Table 1: Examples of bilevel hyperparameter optimization [23; 14; 24] with norm regularizers.

| Machine learning algorithm | Upper Criteria | Base Learner |
|---|---|---|
| Elastic net | $\frac{1}{2}\sum_{i \in I_{val}} \|b_i - \mathbf{x}^T \mathbf{a}_i\|^2$ | $\frac{1}{2}\sum_{i \in I_{tr}} \|b_i - \mathbf{x}^T \mathbf{a}_i\|^2 + \lambda_1 \|\mathbf{x}\|_1 + \frac{\lambda_2}{2}\|\mathbf{x}\|_2^2$ |
| Sparse group Lasso | $\frac{1}{2}\sum_{i \in I_{val}} \|b_i - \mathbf{x}^T \mathbf{a}_i\|^2$ | $\frac{1}{2}\sum_{i \in I_{tr}} \|b_i - \mathbf{x}^T \mathbf{a}_i\|^2 + \sum_{m=1}^{M} \lambda_m \|\mathbf{x}^{(m)}\|_2 + \lambda_{M+1}\|\mathbf{x}\|_1$ |
| Smoothed support vector machine | $\sum_{i \in I_{val}} l_h(b_i \mathbf{x}^T \mathbf{a}_i)$ | $\sum_{i \in I_{tr}} l_h(b_i \mathbf{x}^T \mathbf{a}_i) + \frac{\lambda}{2}\|\mathbf{x}\|^2$ (with constraint $-\bar{\mathbf{x}} \leq \mathbf{x} \leq \bar{\mathbf{x}}$.) |
| Low-rank matrix completion | $\sum_{(i,j) \in \Omega_{val}} \frac{1}{2}\|M_{ij} - \mathbf{x}_i\theta - \mathbf{z}_j\beta - \Gamma_{ij}\|^2$ | $\sum_{(i,j) \in \Omega_{tr}} \frac{1}{2}\|M_{ij} - \mathbf{x}_i\theta - \mathbf{z}_j\beta - \Gamma_{ij}\|^2 + \lambda_0\|\Gamma\|_* + \sum_{g=1}^{G} \lambda_g\|\theta^{(g)}\|_2 + \sum_{g=1}^{G} \lambda_{g+G}\|\beta^{(g)}\|_2$ |
| Robust regression | $\sum_{j \in I_{val}} l_\delta(b_i - \mathbf{x}^T \mathbf{a}_i)$ | $\sum_{j \in I_{tr}} l_\delta(b_i - \mathbf{x}^T \mathbf{a}_i) + \lambda_1\|\mathbf{x}\|_1 + \frac{\lambda_2}{2}\|\mathbf{x}\|_2^2$ |

$l_h$ denotes the smoothed hinge loss given by $l_h(t) = \frac{1}{2} - t$ if $t \leq 0$, $\frac{1}{2}(1-t)^2$ if $0 \leq t \leq 1$ and 0 else.
$l_\delta$ denotes Huber loss given by $l_\delta(t) = \frac{1}{2}t^2$ if $|t| \leq \delta$, $\delta|t| - \frac{1}{2}\delta^2$ if $|t| > \delta$.

## 1.1 RELATED WORK

**Hyperparameter Optimization.** A variety of approaches have been developed for HO [25; 14]. Model-free methods such as grid search [26] and random search [27] are simple but limited. More advanced approaches like Bayesian optimization [28; 29] leverage prior observations to guide evaluations, yet often face scalability issues in high-dimensional spaces.

**Bilevel Optimization.** BLO underpins many machine learning tasks, including meta-learning [30], adversarial learning [31; 32; 33], model selection [34; 35], generative adversarial networks [36; 37], game theory [38]. BLO is challenging to solve in practice due to the inherently nested structure. Many existing methods assume strong convexity of the LL problem, which ensures implicit differentiation based on unique LL solution and simplifies analysis [7; 4; 8; 39; 40; 41; 42; 43]. However, this assumption is often restrictive. When the LL problem is merely convex, multiple optimal solutions may arise, which introduces additional difficulties. To mitigate this, alternative approaches have been developed, including value function-based methods[44; 45; 46; 47; 48], primal–dual frameworks [49] and penalty-based techniques [50; 51; 52; 53].

Beyond convex settings, nonsmooth LL problems present additional difficulties. Further extensions have been proposed, including implicit differentiation based on partial smoothness [54; 15], difference-of-convex (DC) and penalized DC methods [55; 56; 57], gradient-free approaches [58], duality-based cone programming [59]. A separate stream introduces smoothing strategies [60; 61; 62], with Moreau-envelope formulations further enabling efficient single-loop algorithms [63; 64; 65].

## 1.2 OUR NOVELTY AND CONTRIBUTIONS

In this work, we focus on unified framework and efficient algorithms for nonsmooth regularized BLO (3) that avoids smoothing techniques and off-the-shelf solvers, while retaining a single-loop structure. We highlight our novelty in Table 3 and summarize contributions as follows.

- We propose **L**ower-level **D**uality based **P**enalty **M**ethods (LDPM), along with single-loop Hessian-free algorithms LDP-PGM and LDP-ADMM, in which utilize effective epigraphic projections to handle nonsmooth components and significantly reduce computational cost.

- We provide analysis of stationary conditions for nonsmooth regularized BLO in penalty framework, and establish non-asymptotic convergence guarantees for our methods under mild assumptions

- We conduct extensive experiments on both synthetic and real-world tasks, which consistently demonstrate the efficiency and robustness of our approach compared with existing methods.

## 2 LOWER-LEVEL DUALITY BASED PENALIZATION FRAMEWORK

In this section, we propose our framework with lower-level duality based penalization method (LDPM). Prior to this, we observe that the loss functions of base learners in Table 1 share a unified structure of the form $\varphi(A\mathbf{x} - \mathbf{b})$ where $A\mathbf{x} - \mathbf{b}$ abstracts the linear data-sample relationship and $\varphi$ may in general be nonlinear. Accordingly, we denote that

$$l(\mathbf{x}) = \varphi(A_t\mathbf{x} - \mathbf{b}_t). \tag{4}$$

We provide specific forms of $\varphi$ and corresponding $(A_t, \mathbf{b}_t)$ associated with examples in Table 1 as follows:

**Least squares loss**: $\varphi(t) = \frac{1}{2}t^2$, with $A_t\mathbf{x} - \mathbf{b}_t = A_{tr}\mathbf{x} - \mathbf{b}_{tr}$.

**Smoothed hinge loss**: $\varphi(t) = l_h(t)$ with $A_t\mathbf{x} - \mathbf{b}_t = (\mathbf{b}_{tr}A_{tr})\mathbf{x}$. ($l_h$ is defined in Table 1)

**Huber loss**: $\varphi(t) = l_\delta(t)$, with $A_t\mathbf{x} - \mathbf{b}_t = A_{tr}\mathbf{x} - \mathbf{b}_{tr}$. ($l_\delta$ is defined in Table 1)

Building on the structure of $\varphi$, we reformulate (3) based on lower-level duality. Embracing the idea initially proposed by [59], we state the following lemma as a modification and extension of [59, Theorem 2.1].

**Lemma 2.1.** *Given the convex lower semi-continuous[1] functions $l$ and $R_i$, if $\mathrm{ri}(\mathrm{dom}\, l \cap (\cap_{i=1}^{M+1} \mathrm{dom}\, R_i)) \neq \emptyset$[2], then problem (3) has the following equivalent form:*

$$\min_{\mathbf{x}, \boldsymbol{\lambda}, \boldsymbol{\rho}, \boldsymbol{\xi}} L(\mathbf{x}) \quad s.t. \quad \begin{cases} l(\mathbf{x}) + \sum_{i=1}^{M+1} \lambda_i R_i(\mathbf{x}) + \varphi^*(\boldsymbol{\xi}) + \sum_{i=1}^{M+1} \lambda_i R_i^*\left(\frac{\rho_i}{\lambda_i}\right) + \boldsymbol{\xi}^\top \mathbf{b}_t \leq 0, \\ A_t\,\boldsymbol{\xi} + \sum_{i=1}^{M+1} \boldsymbol{\rho}_i = \mathbf{0}. \end{cases} \tag{5}$$

*where $\boldsymbol{\rho} = (\boldsymbol{\rho}_1, ..., \boldsymbol{\rho}_{M+1})$. $\varphi^*$ and $R_i^*$ are the conjugate functions of $\varphi$ and $R_i$, respectively.*

*Remark* 2.2. The equivalence holds both in terms of the set of minimizers and the optimal objective value. A detailed proof and further explanations for Lemma 2.1 are provided in Appendix B.2.

*Remark* 2.3. Slater's condition is broadly satisfied by all examples in Table 1, ensuring strong duality for the LL problem in (3) without requiring strong convexity. For instance, the least squares loss is not strongly convex, yet strong duality still holds under this condition.

Our reformulation differs from [59, Theorem 2.1] in that it explicitly exploits inner structures of $l(\mathbf{x})$. We emphasize that $\varphi^*$ and $R_i^*$ admit closed-form expressions for all problems in Table 1, making Lemma 2.1 applicable. For clarity, the explicit forms of $\varphi^*$ are provided in Appendix B.4.

In particular, for $i = 1, \ldots, M$ with $R_i(\mathbf{x}) = \|\mathbf{x}\|_{(i)}$, the conjugate is the indicator of the dual-norm unit ball $\{\|\mathbf{y}\|_{*(i)} \leq 1\}$ [66, Example 3.26][3]. For $R_{M+1}(\mathbf{x}) = \frac{1}{2}\|\mathbf{x}\|_2^2$, one has $\lambda_{M+1} R_{M+1}^*\left(\frac{\boldsymbol{\rho}_{M+1}}{\lambda_{M+1}}\right) = \frac{\|\boldsymbol{\rho}_{M+1}\|_2^2}{2\lambda_{M+1}}$ [66, Example 3.27]. To refine the constraints in (5), we introduce $r_i$ and $s$ such that $R_i(\mathbf{x}) \leq r_i$ and $\frac{\|\boldsymbol{\rho}_{M+1}\|_2^2}{2\lambda_{M+1}} \leq s$, yielding the following reformulation.

---

[1]The definitions of lower semi-continuity and conjugate are provided in Definition B.2, B.3.

[2]This condition is commonly known as Slater's condition. $\mathrm{ri}(\cdot)$ denotes the relative interior of the set.

[3]$\|\cdot\|_{*(i)}$ denoted the dual norm of $\|\cdot\|_{(i)}$

**Proposition 2.4.** *Under the assumptions of Lemma 2.1, problem* (3) *can be further reformulated as*

$$\min_{\mathbf{x},\boldsymbol{\lambda},\boldsymbol{\rho},\mathbf{r},\boldsymbol{\xi},s} L(\mathbf{x}) \quad \text{s.t.} \quad \begin{cases} l(\mathbf{x}) + \sum_{i=1}^{M+1} \lambda_i r_i + \varphi^*(\boldsymbol{\xi}) + \boldsymbol{\xi}^T \mathbf{b}_t + s \leq 0, \ A_t \boldsymbol{\xi} + \sum_{i=1}^{M+1} \boldsymbol{\rho}_i = \mathbf{0}, \\ \|\mathbf{x}\|_{(i)} \leq r_i, \|\boldsymbol{\rho}_i\|_{*(i)} \leq \lambda_i, \ i = 1, 2, ..., M, \\ \frac{1}{2}\|\mathbf{x}\|_2^2 \leq r_{M+1}, \|\boldsymbol{\rho}_{M+1}\|_2^2 \leq 2\lambda_{M+1}s. \end{cases} \tag{6}$$

For simplicity, we rewrite the left-hand of the first inequality constraint in (6) as:

$$p(\mathbf{x},\boldsymbol{\lambda},\mathbf{r},\boldsymbol{\xi},s) = l(\mathbf{x}) + \sum_{i=1}^{M+1} \lambda_i r_i + \varphi^*(\boldsymbol{\xi}) + \boldsymbol{\xi}^T \mathbf{b}_t + s. \tag{7}$$

We then consider the following penalized problem,

$$\min_{\mathbf{z}:=(\mathbf{x},\boldsymbol{\lambda},\boldsymbol{\rho},\mathbf{r},\boldsymbol{\xi},s)} F_k(\mathbf{z}) \quad \text{s.t.} \quad \begin{cases} \|\mathbf{x}\|_{(i)} \leq r_i, \quad \|\boldsymbol{\rho}_i\|_{*(i)} \leq \lambda_i, \quad i = 1, 2, \ldots, M, \\ \frac{1}{2}\|\mathbf{x}\|_2^2 \leq r_{M+1}, \quad \|\boldsymbol{\rho}_{M+1}\|_2^2 \leq 2\lambda_{M+1}s. \end{cases} \tag{8}$$

where $F_k(\mathbf{z}) := L(\mathbf{x}) + \beta_k p(\mathbf{x},\boldsymbol{\lambda},\mathbf{r},\boldsymbol{\xi},s) + \frac{\beta_k}{2}\|A_t\boldsymbol{\xi} + \sum_{i=1}^{M+1} \boldsymbol{\rho}_i\|^2$ with penalty parameter $\beta_k$. This penalty approach is standard in BLO [50; 63; 64; 65; 44]. Inspired by [67, Theorem 17.1], the following theorem establishes the connection between the optimal solutions of the penalized problem and reformulation (6).

**Theorem 2.5.** *Assume $L, l$ and $R_i$ are lower semi-continuous, with $l$ and $R_i$ convex. Let $\beta_k \to \infty$ and let $\mathbf{z}^{k+1}$ denote a minimizer of* (8) *with $\beta_k$, then every limit point $\mathbf{z}^*$ of the sequence $\{\mathbf{z}^k\}$ is a solution to* (6).

The proof of Theorem 2.5 is provided in Appendix B.3. From the equivalence between (5) and (6), Theorem 2.5 also reveals the connection between (5) and (8).

## 3 PROJECTION-BASED FIRST-ORDER ALGORITHMS

In this section, we develop our algorithms under the penalization framework LDPM. We begin with the following assumptions to support analysis and algorithm design.

**Assumption 3.1.** The UL objective $L$ is bounded below and $\alpha_L$-smooth with respect to $\mathbf{x}$.

**Assumption 3.2.** The function $\varphi$ is convex. $\varphi$ and $\varphi^*$ are $\alpha_p$- and $\alpha_d$-smooth in their domains, respectively.

*Remark* 3.3. Assumptions 3.1 and 3.2 are satisfied by common loss functions, including those in Table 1. They are also suitable for the framework in Section 2. We emphasize that UL objective $L$ *can be nonconvex*.

*Remark* 3.4. Assumption 3.2 implies strong convexity of $\varphi$ and $\varphi^*$, which is detailed in Appendix D.5.1. We emphasize that it does not force the LL objective to be strongly convex. In fact, $l(\mathbf{x}) = \varphi(A_t\mathbf{x} - \mathbf{b}_t)$ is convex but not strongly convex when $A_t$ is not of full row rank.

To handle the nonsmooth constraints of (8) induced by different norms, we introduce the cone sets

$$\begin{aligned} &\mathcal{K}_i := \{(\mathbf{x},\mathbf{r}) \mid \|\mathbf{x}\|_{(i)} \leq r_i\}, \ \mathcal{K}_i^d := \{(\boldsymbol{\rho}_i, \lambda_i) \mid \|\boldsymbol{\rho}_i\|_{*(i)} \leq \lambda_i\}, \quad i = 1, 2, ..., M, \\ &\mathcal{K}_{M+1} := \{(\mathbf{x},\mathbf{r}) \mid \|\mathbf{x}\|_2^2 \leq 2r_{M+1}\}, \ \mathcal{K}_{M+1}^d := \{(\boldsymbol{\rho}_{M+1}, \lambda_{M+1}, s) \mid \|\boldsymbol{\rho}_{M+1}\|_2^2 \leq 2\lambda_{M+1}s\}. \end{aligned} \tag{9}$$

Each of these set is projection-friendly, which enables efficient epigraphic projections. The details of the projection operations are provided in Appendix C. Hence, a natural strategy is to manage the constraints of (8) via projections onto $\mathcal{K}_i$ and $\mathcal{K}_i^d$. We develop algorithms for two settings: (i) single-round global regularization on $\mathbf{x}$ in Section 3.1 and (ii) multiple interacting regularizers in Section 3.2.

## 3.1 SEPARABLE REGULARIZERS

In this subsection, we present the algorithm for (3) when the LL problem involves separate regularizers as a group of component-wise terms. In this case, the LL problem is written as

$$\mathbf{x} \in \arg\min_{\hat{\mathbf{x}}} \{l(\hat{\mathbf{x}}) + \sum_{i=1}^{M} \lambda_i \|\hat{\mathbf{x}}^{(i)}\|_{(t)}\}, \text{ with } \mathbf{x} = (\mathbf{x}^{(1)}, ..., \mathbf{x}^{(M)}), \ M \geq 1,$$

where $\mathbf{x}^{(i)}$ represents the $i$-th subvector of $\mathbf{x}$ and $\|\cdot\|_{(t)}$ represents a prescribed norm applied to each group.

**When $M = 1$,** it involves a single regularizer $R_1(\mathbf{x})$, corresponding to simpler models such as toy Lasso or logistic regression. If $R_1(\mathbf{x}) = \|\mathbf{x}\|_{(t)}$, the constraints of (8) reduce to:

$$\|\mathbf{x}\|_{(t)} \leq r_1, \ \|\boldsymbol{\rho}\|_{*(t)} \leq \lambda_1. \tag{10}$$

If $R_1(\mathbf{x}) = \frac{1}{2}\|\mathbf{x}\|_2^2$, the constraints of (8) simplify to:

$$\frac{1}{2}\|\mathbf{x}\|_2^2 \leq r_1, \ \frac{1}{2}\|\boldsymbol{\rho}\|_2^2 \leq \lambda_1 s. \tag{11}$$

(10) and (11) are consistent with the structure in (9) and can be compactly expressed as

$$\mathbf{z} \in \mathcal{K} := \mathcal{K}_1 \times \mathcal{K}_1^d. \tag{12}$$

**When $M > 1$,** the LL problem in (3) involves group regularization, with group-wise $\ell_2$-regularization as the most common example such as group Lasso. Although multiple terms appear, this setting essentially corresponds to a single-round regularization over the entire $\mathbf{x}$, under which the constraints in (8) reduce to:

$$\|\mathbf{x}^{(i)}\|_{(t)} \leq r_i, \ \|\boldsymbol{\rho}^{(i)}\|_{(t)} \leq \lambda_i, \text{ with } \boldsymbol{\rho} = (\boldsymbol{\rho}^{(1)}, ..., \boldsymbol{\rho}^{(M)}), \ i = 1, \ldots, M,$$

where $\boldsymbol{\rho}^{(i)}$ is the $i$-th subvector of $\boldsymbol{\rho}$. The above constraints are separable in $i$ and equivalent to:

$$(\mathbf{x}^{(1)}, r_1, \mathbf{x}^{(2)}, r_2, ..., \mathbf{x}^M, r_M) \in \mathcal{K}_1 \times \cdots \times \mathcal{K}_M, \ (\boldsymbol{\rho}, \boldsymbol{\lambda}) \in \mathcal{K}_1^d \times \cdots \times \mathcal{K}_M^d, \\ \mathbf{z} \in \mathcal{K} := (\mathcal{K}_1 \times \cdots \times \mathcal{K}_M) \times (\mathcal{K}_1^d \times \cdots \times \mathcal{K}_M^d). \tag{13}$$

In summary, the penalized problem (8) can be uniformly expressed as

$$\min_{\mathbf{z}} \frac{1}{\beta_k} F_k(\mathbf{z}) \quad \text{s.t.} \quad \mathbf{z} \in \mathcal{K},$$

where $\mathcal{K}$ is defined in (12) or (13) and remains projection-friendly. Accordingly, we adopt projection gradient descent to solve it in this setting, as outlined in Algorithm 1. In each iteration, we update $\mathbf{z}$ as

$$\mathbf{z}^{k+1} = \text{proj}_{\mathcal{K}}(\mathbf{z}^k - \frac{e_k}{\beta_k}\nabla_{\mathbf{z}} F_k(\mathbf{z}^k)), \tag{14}$$

where $e_k > 0$ are the step sizes and $\text{proj}_{\mathcal{K}}(\mathbf{x})$ is the projection of $\mathbf{x}$ onto $\mathcal{K}$.

---

**Algorithm 1** **L**ower-level **D**uality **P**enalization **P**rojection **G**radient **M**ethod (LDP-PGM)

---
1: Input $\boldsymbol{\lambda}^0 > 0, \boldsymbol{\xi}^0$, step sizes $\{e_k\}$, penalty parameters $\{\beta_k\}$. Initialize $\mathbf{x}^0, \mathbf{r}^0, \boldsymbol{\rho}^0, s^0$.
2: **for** $k = 0, 1, 2, ...$ **do**
3:     Update $\mathbf{z}^{k+1}$ with (14).
4: **end for**

---

We remark that Algorithm 1 is a single loop algorithm that does not require solving any subproblem. The initialization is detailed in Appendix D.1. We now turn to the non-asymptotic convergence analysis of

Algorithm 1. From the proof of Lemma 2.1 and the definition of $p$, it follows that $p(\mathbf{x}, \boldsymbol{\lambda}, \mathbf{r}, \boldsymbol{\xi}, s) \geq 0$ and the feasible set has no interior point [68; 44; 59]. Consequently, the classical KKT conditions for nonsmooth constrained optimization [69] are inapplicable. Instead, we adopt the approximate KKT conditions [70] and introduce the following merit functions,

$$
\begin{aligned}
\phi_{res}^k(\mathbf{z}) &:= \text{dist}\left(0, \nabla_{\mathbf{z}} F_k(\mathbf{z}) + \mathcal{N}_{\mathcal{K}}(\mathbf{z})\right), \\
\phi_{fea}(\mathbf{z}) &:= \max\{p(\mathbf{x}, \boldsymbol{\lambda}, \mathbf{r}, \boldsymbol{\xi}, s), \|A_t \boldsymbol{\xi} + \boldsymbol{\rho}\|^2\}.
\end{aligned}
\tag{15}
$$

The residual function $\phi_{res}^k(\mathbf{z})$ quantifies the stationarity for (8), because $\phi_{res}^k(\mathbf{z}) = 0$ if and only if $\mathbf{z}$ is a stationary point of (8). Meanwhile, the function $\phi_{fea}(\mathbf{z})$ is interpreted as a feasibility measure for the penalized constraints [71]. Combined with the structure of BLO, $\phi_{fea}(\mathbf{z})$ regulates optimality conditions of LL problem of (3). We clarify corresponding conclusions and explanations in Appendix D.2.

**Theorem 3.5.** *Under Assumptions 3.1 and 3.2, suppose $\beta_k = \underline{\beta}(1+k)^p$ with $\underline{\beta} > 0$ and $p \in (0, 1/2)$. If the step sizes $\{e_k\}$ in Algorithm 1 satisfy $0 < \underline{e} \leq e_k \leq \min\{\frac{\underline{\beta}}{\alpha_L + \underline{\beta}\|A_t\|_2^2 \alpha_p}, 1, \frac{1}{\alpha_d + \|A_t\|_2^2}\}$, the sequence $\{\mathbf{z}^k\}$ generated by Algorithm 1 satisfies*

$$
\min_{0 \leq k \leq K} \phi_{res}^k(\mathbf{z}^{k+1}) = \mathcal{O}\left(\frac{(L_c + 1/\underline{e})\underline{\beta}}{K^{\frac{1}{2} - p}}\right),
$$

*where $L_c := \max\{\frac{1}{\underline{\beta}}\alpha_L + \|A_t\|_2^2 \alpha_p, \alpha_d + \|A_t\|_2^2, 1\}$. Furthermore, if the sequence $\{F_k(\mathbf{z}^k)\}$ is bounded, then it holds that*

$$
0 \leq \min_{0 \leq k \leq K} \phi_{fea}(\mathbf{z}^k) = \mathcal{O}(1/K^p).
$$

We remark that the lower bound $\underline{e}$ and the boundedness assumption on $\{F_k(\mathbf{z}^k)\}$ are widely adopted in single-loop Hessian-free BLO algorithms [64; 65; 63]. We provide detailed proofs in Appendix D.3.

Meanwhile, the boundedness assumption on $\{F_k(\mathbf{z}^k)\}$ is standard and necessary in single-loop penalty-based methods for BLO without lower-level strong convexity. Relaxing this assumption in nonconvex, nonsmooth single-loop bilevel settings would require substantially stronger analytical tools derived from structural properties of the problem, such as global error bounds, Kurdyka–Łojasiewicz (KL) inequalities. To the best of our knowledge, establishing such results for nonconvex nonsmooth bilevel penalty methods remains open. In this sense, the boundedness of $\{F_k(z^k)\}$ should be regarded as a available technical condition, rather than any undesirable behavior of our algorithms.

## 3.2 Nonseparable Regularizers

In this subsection, we focus on the scenarios that LL problem in (3) involves multiple interacting regularizers, where several regularization terms are applied to the entire vector, such as elastic net or sparse group Lasso. Using the definitions of $\mathcal{K}_i$ and $\mathcal{K}_i^d$ from (9), the constraints of (8) can be written as

$$
\begin{aligned}
(\mathbf{x}, \mathbf{r}) \in \mathcal{K}_i, (\boldsymbol{\rho}_i, \lambda_i) \in \mathcal{K}_i^d, \ i = 1, 2, ..., M, \\
(\mathbf{x}, \mathbf{r}) \in \mathcal{K}_{M+1}, (\boldsymbol{\rho}_{M+1}, \lambda_{M+1}, s) \in \mathcal{K}_{M+1}^d,
\end{aligned}
$$

which can be further expressed as

$$
(\mathbf{x}, \mathbf{r}) \in \mathcal{K}_1 \cap \cdots \cap \mathcal{K}_{M+1}, \ (\boldsymbol{\rho}, \boldsymbol{\lambda}, s) \in \mathcal{K}_1^d \times \cdots \times \mathcal{K}_{M+1}^d.
\tag{16}
$$

We denote $\mathcal{K}_*^d := \mathcal{K}_1^d \times \cdots \times \mathcal{K}_{M+1}^d$. (16) can be equivalently expressed as

$$
\mathbf{z} \in \mathcal{K} := (\mathcal{K}_1 \cap \cdots \cap \mathcal{K}_{M+1}) \times \mathcal{K}_1^d \times \cdots \mathcal{K}_{M+1}^d = (\mathcal{K}_1 \cap \cdots \cap \mathcal{K}_{M+1}) \times \mathcal{K}_*^d.
$$

Since each $\mathcal{K}_i^d$ is projection-friendly, the product set $\mathcal{K}_*^d$ inherits this property. In contrast, the intersection $\cap_{i=1}^{M+1}\mathcal{K}_i$ defined over the shared variable $(\mathbf{x}, \mathbf{r})$ may not be projection-friendly. Although projection onto such intersections has been studied [72; 73], the required iterations are often complex. To address this, we reformulate the constraint to avoid direct projection onto the intersection:

$$\mathbf{z} \in \mathcal{K}_i \times \mathcal{K}_*^d,\ i = 1, 2, ..., M + 1. \tag{17}$$

For each $i$, since $\mathcal{K}_*^d$ and $\mathcal{K}_i$ are projection-friendly, the product set $\mathcal{K}_i \times \mathcal{K}_*^d$ remains projection-friendly. Hence, we introduce auxiliary variables $\mathbf{u}_i$, leading to uniform expression of (8):

$$\min_{\mathbf{z},\mathbf{u}}\quad \frac{1}{\beta_k}F_k(\mathbf{z})\quad \text{s.t.}\quad \mathbf{z} = \mathbf{u}_i,\ \mathbf{u}_i \in \mathcal{K}_i \times \mathcal{K}_*^d,\ i = 1, ..., M + 1, \tag{18}$$

where $\mathbf{u} = (\mathbf{u}_1, ..., \mathbf{u}_{M+1})$. We define the indicator function as $g_i(\mathbf{z}) = I_{\mathcal{K}_i \times \mathcal{K}_*^d}(\mathbf{z}), i = 1, 2, ..., M + 1$. The augmented Lagrangian function of problem (18) is given by:

$$\mathcal{L}_\gamma^k(\mathbf{z}, \mathbf{u}, \boldsymbol{\mu}) = \frac{1}{\beta_k}F_k(\mathbf{z}) + \sum_{i=1}^{M+1} g_i(\mathbf{u}_i) + \sum_{i=1}^{M+1} \langle\boldsymbol{\mu}_i, \mathbf{u}_i - \mathbf{z}\rangle + \frac{\gamma}{2}\sum_{i=1}^{M+1}\|\mathbf{u}_i - \mathbf{z}\|^2, \tag{19}$$

where $\boldsymbol{\mu} := (\boldsymbol{\mu}_1, ..., \boldsymbol{\mu}_{M+1})$ denotes the Lagrangian multiplier associated with constraint $\mathbf{z} = \mathbf{u}_i$. Based on $\mathcal{L}_\gamma^k(\mathbf{z}, \mathbf{u}, \boldsymbol{\mu})$, we adopt an alternative approach to solve (18) inspired by the core idea of the Alternating Direction Method of Multipliers (ADMM). Here, $\gamma$ serves as a penalty parameter and is taken to be a independent positive constant. This is because ADMM is well known to be robust to the choice of $\gamma$, and convergence is guaranteed for any fixed $\gamma > 0$ [74; 75]. The method alternately updates the primal variables $\mathbf{z}$ and $\mathbf{u}$, followed by a dual ascent step on $\boldsymbol{\mu}$. At iteration $k$, we perform a gradient update on $\mathbf{z}$ initialized at $\mathbf{z}^k$:

$$\mathbf{z}^{k+1} = \mathbf{z}^k - e_k\mathbf{d}_\mathbf{z}^k, \tag{20}$$

where the update direction $\mathbf{d}_\mathbf{z}^k$ corresponds to the gradient of $\mathcal{L}_\gamma^k$ with respect to $\mathbf{z}$ evaluated at $(\mathbf{z}^k, \mathbf{u}^k, \boldsymbol{\mu}^k)$ and $e_k$ is the step size of $k$-th iteration. This is equivalent to minimize the proximal subproblem of $\mathcal{L}_\gamma^k$:

$$\mathbf{z}^{k+1} = \arg\min_{\mathbf{z}}\{\mathcal{L}_\gamma^k(\mathbf{z}^k, \mathbf{u}^k, \boldsymbol{\mu}^k) + \langle\nabla_z\mathcal{L}_\gamma^k(\mathbf{z}^k, \mathbf{u}^k, \boldsymbol{\mu}^k), \mathbf{z} - \mathbf{z}^k\rangle + \frac{1}{2e_k}\|\mathbf{z} - \mathbf{z}^k\|^2\}.$$

Next, for the $\mathbf{u}$-subproblem, we update $\mathbf{u}_i$ by minimizing $\mathcal{L}_\gamma^k$ with respect to $\mathbf{u}_i$ as

$$\mathbf{u}_i^{k+1} = \arg\min_{\mathbf{u}_i}\{g_i(\mathbf{u}_i) + \frac{\gamma}{2}\|\mathbf{u}_i - \mathbf{z}^{k+1} + \frac{\boldsymbol{\mu}_i^k}{\gamma}\|^2\}, \tag{21}$$

which is equivalent to performing the direct projection onto $\mathcal{K}_i \times \mathcal{K}_*^d$, yielding:

$$\mathbf{u}_i^{k+1} = \mathrm{proj}_{\mathcal{K}_i \times \mathcal{K}_*^d}(\frac{\boldsymbol{\mu}_i^k}{\gamma} - \mathbf{z}^{k+1}),\ i = 1, ..., M + 1. \tag{22}$$

Finally, for the dual multipliers $\boldsymbol{\mu}_i$, we update them as

$$\boldsymbol{\mu}_i^{k+1} = \boldsymbol{\mu}_i^k + \gamma(\mathbf{u}_i^{k+1} - \mathbf{z}^{k+1}),\ i = 1, ..., M + 1. \tag{23}$$

We summarize these iterations in Algorithm 2 and the initialization is detailed in Appendix D.1.

Algorithm 2 differs from standard ADMM in two key aspects: (i) the augmented Lagrangian $\mathcal{L}_\gamma^k$ varies with the iterative $\beta_k$. (ii) instead of exactly minimizing $\mathcal{L}_\gamma^k$ in the $\mathbf{z}$-subproblem, we adopt its first-order approximation at $\mathbf{z}^k$. The strategy is commonly employed in gradient-based ADMM [76; 77]. We now discuss the non-asymptotic convergence property of Algorithm 2. Similar to Theorem 3.5, we define the following merit functions analogous to (15):

$$\begin{aligned}\phi_{res}^k(\mathbf{z}) &:= \mathrm{dist}\left(0, \nabla F_k(\mathbf{z}) + \mathcal{N}_\mathcal{K}(\mathbf{z})\right),\\ \phi_{fea}(\mathbf{z}) &:= \max\{p(\mathbf{x}, \boldsymbol{\lambda}, \mathbf{r}, \boldsymbol{\xi}, s), \|A_t\boldsymbol{\xi} + \sum_{i=1}^{M+1}\boldsymbol{\rho}_i\|^2\}.\end{aligned} \tag{24}$$

To establish the convergence results for Algorithm 2, we invoke the following assumption.

---

**Algorithm 2** **L**ower-level **D**uality **P**enalization **A**lternating **D**irection **M**ethod of **M**ultipliers (LDP-ADMM)

---

1: Input $\boldsymbol{\lambda}^0 > 0, \boldsymbol{\xi}^0$, constant $\gamma > 0$. Initialize $\mathbf{x}^0, \mathbf{r}^0, \boldsymbol{\rho}_i^0, s^0$ and $\mathbf{u}_i^0 = \mathbf{z}^0$. Input sequences $\{e_k\}, \{\beta_k\}$.
2: **for** $k = 0, 1, 2, ...$ **do**
3:     Update $\mathbf{z}^{k+1}$ with (20).
4:     Update $\mathbf{u}^{k+1}$ with (22).
5:     update $\boldsymbol{\mu}^{k+1}$ with (23).
6: **end for**

---

**Assumption 3.6.** The sequence of multipliers $\{\boldsymbol{\mu}^k\}$ is bounded.

As discusses in [74; 75], the convergence of ADMM for nonconvex nonsmooth composite optimization is highly challenging without imposing assumptions like Assumption 3.6, which is an open question. Assumption 3.6 is popularly employed in ADMM approaches [78; 79; 80; 81].

**Theorem 3.7.** *Under Assumptions 3.1, 3.2, 3.6, let* $\beta_k = \underline{\beta}(1 + k)^p$ *with* $\underline{\beta} > 0$ *and* $p \in (0, 1/2)$. *If step sizes* $e_k$ *satisfy* $0 < \underline{e} \le e_k \le 1/M_e$, *where* $M_e$ *is a constant defined as*

$$M_e = \max\{\frac{1}{\underline{\beta}}\alpha_L + \|A_t\|_2^2 \alpha_p, \alpha_d + \|A_t\|_2^2, 1\} + (M + 1)\gamma,$$

*then the sequence* $\{\mathbf{z}^k\}$ *in Algorithm 2 satisfies* $\lim_{k\to\infty} \phi_{res}^k(\mathbf{z}^{k+1}) = 0$. *Moreover, if* $\{F_k(\mathbf{z}^k)\}$ *is bounded, then* $\lim_{k\to\infty} \phi_{fea}(\mathbf{z}^k) = 0$.

We remark that the lower bound $\underline{e}$ and the boundedness assumption on $\{F_k(\mathbf{z}^k)\}$ are analogous to Theorem 3.5. We provide the detailed explanations and proofs of Theorem 3.7 in Appendix D.4.

## 4 NUMERICAL EXPERIMENTS

In this section, we assess the numerical performance of our proposed algorithms through experiments on both synthetic and real datasets. Specifically, we compare with several existing hyperparameter optimization algorithms under the BLO framework (3), including search methods, TPE [82], IGJO [14], IFDM [54; 15], VF-iDCA [57], LDMMA [59], MEHA [63], BiC-GAFFA [65], as detailed in Appedix E.1.

We evaluate all tasks listed in Table 1. The comparison is based on validation and test errors obtained from the LL minimizers, together with the overall running time. In addition, we also report the lower-level duality gap and the sparsity of the resulting solutions. These metrics are standard in the assessment of (bilevel) hyperparameter optimization [57; 14]. For each task, we conduct experiments across diverse data settings or datasets with 10 independent repetitions, and report aggregated statistical outcomes. Depending on the regularization structure, we employ either LDP-PGM (Algorithm 1) or LDP-ADMM (Algorithm 2). The specific choice of algorithm for each problem is detailed in the corresponding subsection of Appendix E. In all reported experimental results, both variants are uniformly denoted as LDPM in this section.

### 4.1 EXPERIMENTS ON SYNTHETIC DATA

We focus on two prototypical tasks built from simple synthetic data: least squares/Huber regression with various Lasso-type regularizers and low-rank matrix completion, as listed in Table 1. The synthetic data consists of observation matrices sampled from specific distributions and response vectors generated with controlled noise. The detailed data generation process is provided in Appendix E.2.

**Lasso-type Regression.** We consider three regularizers: elastic net [16], group Lasso [83], and sparse group Lasso [17]. These formulations all promote sparsity while balancing model complexity and predictive accu-

racy. Table 2 presents the statistical results for the sparse group Lasso problem, including validation error, test error, and running time. Results for the elastic net and group Lasso problems are reported in Tables 5 and 6, respectively. Detailed experimental settings for each method are provided in the corresponding subsections of Appendix E.2. Overall, LDPM demonstrates superior performance on synthetic data, consistently achieving the lowest test errors while requiring the least computational time compared to baseline methods.

Table 2: Sparse group Lasso problems on synthetic data, where $p$ represents the number of features.

| Settings | $p = 600$ | | | $p = 1200$ | | |
|---|---|---|---|---|---|---|
| | Time(s) | Val. Err. | Test Err. | Time(s) | Val. Err. | Test Err. |
| Grid | $6.36 \pm 1.88$ | $84.73 \pm 5.29$ | $87.34 \pm 15.91$ | $13.68 \pm 2.49$ | $84.68 \pm 4.31$ | $86.00 \pm 18.43$ |
| Random | $6.02 \pm 2.01$ | $135.17 \pm 5.95$ | $147.43 \pm 25.54$ | $12.64 \pm 2.84$ | $137.87 \pm 14.21$ | $146.25 \pm 15.52$ |
| IGJO | $1.58 \pm 0.28$ | $101.93 \pm 4.07$ | $96.36 \pm 13.72$ | $7.35 \pm 1.46$ | $130.56 \pm 14.02$ | $106.70 \pm 4.01$ |
| VF-iDCA | $0.56 \pm 0.15$ | $56.96 \pm 5.58$ | $76.84 \pm 11.33$ | $8.63 \pm 2.91$ | $86.38 \pm 6.40$ | $87.58 \pm 8.90$ |
| LDMMA | $0.57 \pm 0.13$ | $82.70 \pm 5.03$ | $72.44 \pm 14.72$ | $4.72 \pm 2.15$ | $83.93 \pm 7.32$ | $84.03 \pm 9.08$ |
| MEHA | $0.44 \pm 0.04$ | $70.53 \pm 6.34$ | $73.12 \pm 10.98$ | $2.84 \pm 0.22$ | $84.93 \pm 5.74$ | $82.94 \pm 7.91$ |
| BiC-GAFFA | $0.39 \pm 0.02$ | $67.42 \pm 6.28$ | $71.45 \pm 10.74$ | $2.52 \pm 0.29$ | $82.21 \pm 5.03$ | $79.81 \pm 7.66$ |
| LDPM | $\mathbf{0.31 \pm 0.03}$ | $65.11 \pm 6.62$ | $\mathbf{65.91 \pm 8.12}$ | $\mathbf{2.02 \pm 0.11}$ | $76.39 \pm 4.68$ | $\mathbf{74.11 \pm 6.35}$ |

| Settings | $p = 2400$ | | | $p = 4800$ | | |
|---|---|---|---|---|---|---|
| | Time(s) | Val. Err. | Test Err. | Time(s) | Val. Err. | Test Err. |
| Grid | $24.23 \pm 4.05$ | $95.63 \pm 14.13$ | $84.86 \pm 15.09$ | $47.09 \pm 6.34$ | $128.94 \pm 24.11$ | $115.41 \pm 17.62$ |
| Random | $22.17 \pm 6.85$ | $120.04 \pm 15.36$ | $146.77 \pm 16.70$ | $46.3 \pm 5.57$ | $99.41 \pm 16.55$ | $122.49 \pm 19.46$ |
| IGJO | $11.14 \pm 7.44$ | $91.59 \pm 14.97$ | $115.98 \pm 14.94$ | $29.76 \pm 9.44$ | $99.75 \pm 15.14$ | $106.49 \pm 7.48$ |
| VF-iDCA | $14.31 \pm 1.45$ | $63.21 \pm 5.36$ | $81.92 \pm 10.54$ | $45.12 \pm 3.10$ | $73.66 \pm 10.53$ | $96.09 \pm 9.14$ |
| LDMMA | $7.50 \pm 0.21$ | $66.23 \pm 7.47$ | $79.09 \pm 13.75$ | $36.14 \pm 3.65$ | $78.61 \pm 12.32$ | $95.81 \pm 9.43$ |
| MEHA | $6.32 \pm 0.18$ | $74.92 \pm 9.10$ | $77.58 \pm 10.21$ | $5.96 \pm 0.41$ | $87.42 \pm 7.52$ | $93.11 \pm 7.44$ |
| BiC-GAFFA | $5.11 \pm 0.10$ | $86.83 \pm 13.53$ | $76.38 \pm 8.60$ | $5.03 \pm 0.63$ | $94.34 \pm 8.19$ | $92.05 \pm 7.13$ |
| LDPM | $\mathbf{4.61 \pm 0.06}$ | $\mathbf{84.85 \pm 6.21}$ | $72.93 \pm 2.64$ | $\mathbf{4.32 \pm 0.14}$ | $\mathbf{83.12 \pm 5.70}$ | $88.64 \pm 5.11$ |

**Low-rank matrix completion.** For this problem, we conduct the numerical experiments on $60 \times 60$ matrices [57; 14]. The data generation process, detailed statistical results, and corresponding analysis are presented in Appendix E.2.4.

**Robust regression.** For this problem, we consider numerical experiments on regression problems with Huber loss combined with norm regularizers. The data generation process, detailed statistical results, and corresponding analysis are presented in Appendix E.2.5.

**Sensitivity of parameters.** We conduct sensitivity experiments on LDP-PGM and LDP-ADMM. The results summarized in Table 9 show that both algorithms exhibit stable convergence across various parameter settings.

## 4.2 EXPERIMENTS ON REAL-WORLD DATA

To assess robustness of our algorithms, we conduct experiments on larger real-world datasets with more complex sampling distributions. Specifically, we consider experiments on elastic net, smoothing SVM and sparse logistic regression, as listed in Table 1. All datasets are drawn from the LIBSVM repository. For each repetition, we randomly shuffle and split the data into training, validation and test sets.

**Elastic Net.** In this part, we conduct experiments on datasets gisette [84] and sensit [85]. We summarize the comparative experimental results in Table 10 and show the validation and test error curves over time for each algorithm in Figure 1. Even in these high-dimensional settings, LDPM delivers competitive accuracy while maintaining fast convergence. Additional experimental details are provided in Appendix E.4.1.

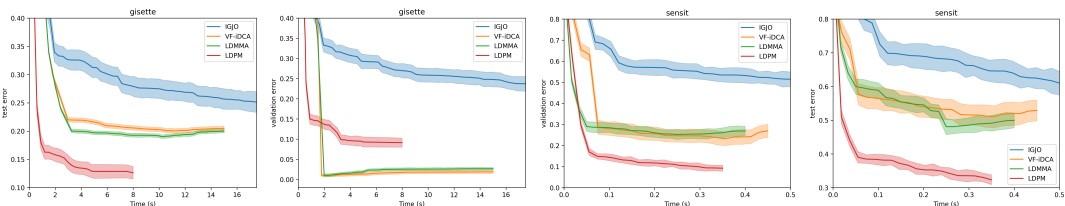

Figure 1: Comparison of the algorithms on Elastic Net problem for real-world datasets.

**Smoothed Support Vector Machine.** We perform 6-fold cross-validation using medical statistics datasets, including diabetes, sonar, a1a [86]. Details of the datasets and experimental setup are given in Appendix E.4.2. We plots the validation and test errors of each algorithm over time in Figure 2, which clearly shows that LDPM converges more rapidly and achieves lower error levels than the competing methods.

**Sparse Logistic Regression.** [87; 24] In this part, we conduct experiments on three large-scale document classification datasets, news20.binary, rcv1.binary and real-sim. Dataset characteristics and experimental details are provided in Appendix E.4.3. We plot the validation and test error curves over time in Figure 3 and report the corresponding final validation and test accuracies in Table 12 for comparison. LDPM consistently converges faster and achieves the lowest validation and test errors.

## 5    CONCLUSION

In this paper, we introduce a penalty framework based on lower-level duality for nonsmooth bilevel hyperparameter optimization (3). Notably, we solve the penalized problem using single-loop first-order algorithms. Theoretically, we establish convergence guarantees for the proposed algorithms. Empirically, through numerical experiments on both synthetic and real-world datasets, our methods exhibit superior performance compared to existing approaches, particularly among the illustrated examples.

## ETHICS STATEMENT

This work does not present any apparent ethical concerns. The proposed algorithms are purelytheoretical and experimental in nature, and they do not involve human subjects, sensitive personaldata, or applications that pose foreseeable risks of harm. Nevertheless, we recognize the importanceof ethical considerations in machine learning research and adhere to the ICLR Code of Ethics.

## REPRODUCIBILITY STATEMENT

To ensure reproducibility, we provide the following: (1) all theoretical results are accompaniedby complete proofs in the appendix; (2) experimental setups, including dataset preprocessing andhyperparameter settings, are described in detail; (3) source code implementing our algorithms will bemade available in the supplementary material. These resources should allow others to fully replicateour findings.

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

## The Use of Large Language Models (LLMs)

No large language models (LLMs) were used in the development of the research ideas, theoretical results, experiments, or writing of this paper. All contents are solely the work of the authors.

## A  Expanded Introduction

BLO underpins many machine learning tasks, including meta-learning [30], adversarial learning [31; 32; 33], reinforcement learning [88; 89; 90; 91], model selection [34; 35], generative adversarial networks [36; 37], and game theory [38]. Early approaches mainly used gradient-based methods, broadly categorized into Iterative Differentiation (ITD), which unrolls the LL problem and computes hypergradients via backpropagation [7; 4; 41; 92; 93; 8], and Approximate Implicit Differentiation (AID), which derives gradients from LL optimality conditions [39; 40; 94; 95; 96].

Recent advances explore fully first-order methods that avoid Hessian or implicit gradients [58; 97; 47]. To handle multiple LL minima, [44] proposed a value-function reformulation, inspiring penalty-based algorithms [45; 50; 52; 48; 53; 51]. Another promising direction employs the Moreau envelope to smooth the bilevel structure, enabling single-loop, Hessian-free algorithms converging to well-defined KKT points [98; 64; 65].

For BLO with nonsmooth LL problems, [54] introduces an implicit differentiation framework via block coordinate descent, later extended to general nonsmooth settings [15]. Alternative approaches include difference-of-convex (DC) and penalized DC methods [55; 56; 57], which rely on the LL value function, and smoothing strategies [60; 61; 62]. More recent work explores gradient-free algorithms with inexact subproblems [58], duality-based cone programming that bypasses the value function [59], and Moreau-envelope methods extended to nonsmooth cases, yielding efficient single-loop algorithms [63]. Compared with ex-

Table 3: Comparison between our algorithm and other single-loop Hessian-free methods

| Method | LL Objective | UL Objective | Nonsmooth | Single-loop | Hessian-free | Non-asymptotic |
|---|---|---|---|---|---|---|
| BOME[51] | L-Smooth Gradient-Bounded | L-Smooth PL Condition | Inapplicable | ✗ | ✓ | ✓ |
| GALET[99] | L-Smooth Gradient-Bounded | L-Smooth PL Condition | Inapplicable | ✗ | ✗ | ✓ |
| V-PBGD[50] | L-Smooth Gradient-Bounded | L-Smooth PL Condition | Inapplicable | ✗ | ✓ | ✓ |
| VF-iDCA[57] | Convex | Convex Nonsmooth | Off-the-shelf Solvers | ✓ | ✓ | ✗ |
| LDMMA[59] | Convex | Convex Nonsmooth | Off-the-shelf Solvers | ✓ | ✓ | ✗ |
| SLM[100] | L-Smooth | L-Smooth PL Condition | Inapplicable | ✗ | ✓ | ✓ |
| MEHA[63] | L-Smooth | Nonsmooth | Smoothing (Moreau) | ✓ | ✓ | ✓ |
| BiC-GAFFA[65] | L-Smooth | L-Smooth Constrained | Smoothing (Moreau) | ✓ | ✓ | ✓ |
| LV-HBA[64] | L-Smooth | L-Smooth Constrained | Smoothing (Moreau) | ✓ | ✓ | ✓ |
| LDPM(Ours) | L-Smooth | Convex Nonsmooth | Projection | ✓ | ✓ | ✓ |

isting methods under the same oracles, our approach demonstrates distinctive advantages. Our framework LDPM does not rely on smoothing schemes. We directly address the nonsmooth components via efficient

projection operations. Therefore, each iteration in our method requires only gradient evaluations and projections. By contrast, BiC-GAFFA depends on smoothing-based reformulation that converts LL problem into a constrained lower-level problem, which also introduces additional computational burden. Similarly, MEHA adopts the Moreau envelope to achieve smooth approximations. Meanwhile, LDMMA and VF-iDCA also differ significantly from our proposed LDPM. We primarily combine gradient descent and projection algorithms, whereas LDMMA and VF-iDCA directly use an off-the-shelf solver to address the subproblem. Furthermore, an important aspect of the LDMMA algorithm is the value function and an additional parameter $\epsilon$, which yields an approximation of the original BLO, while LDPM directly solves the original BLO.

## B  PROOFS AND EXPLANATIONS FOR SECTION 2

In this subsection, we provide the proofs of the results concerning the penalty framework in Section 2. The definitions of convex and lower semi-continuous functions in Lemma 2.1 are given as follows.

**Definition B.1. (Convex Function):** Let $C \subseteq \mathbb{R}^n$ be a convex set. A function $f : C \to \mathbb{R}$ is called **convex** if for all $\mathbf{x}, \mathbf{y} \in C$ and for all $\theta \in [0, 1]$, the following inequality holds:

$$f(\theta \mathbf{x} + (1 - \theta)\mathbf{y}) \leq \theta f(\mathbf{x}) + (1 - \theta) f(\mathbf{y}).$$

**Definition B.2. (Lower semi-continuous Function):** A function $f : \mathbb{R}^n \to \mathbb{R} \cup \{+\infty\}$ is said to be **lower semi-continuous** at a point $\mathbf{x}_0 \in \mathbb{R}^n$ if

$$\liminf_{\mathbf{x} \to \mathbf{x}_0} f(\mathbf{x}) \geq f(\mathbf{x}_0).$$

Equivalently, for all $\alpha \in \mathbb{R}$, the sublevel set $\{\mathbf{x} \in \mathbb{R}^n \mid f(\mathbf{x}) \leq \alpha\}$ is a closed set. If $f$ is lower semi-continuous at every point in its domain, we say that $f$ is a lower semi-continuous function.

These two properties are essential in our framework: convexity ensures the validity of lower-level duality, while the lower semi-continuity guarantees the existence of a minimizer in the LL problem.

**Definition B.3. (Conjugate function):** Given a function $f : \mathbb{R}^n \to \mathbb{R} \cup \{+\infty\}$, the conjugate function $f*$ is defined as

$$f^*(\mathbf{y}) = \sup_{\mathbf{x} \in \mathbb{R}^n} \{\mathbf{y}^T \mathbf{x} - f(\mathbf{x})\}, \ \ \mathbf{y} \in \mathbb{R}^n.$$

**Definition B.4. (Domain):** Let $f : \mathbb{R}^n \to \mathbb{R} \cup \{+\infty\}$. The domain of $f$, also called the effective domain, is the set of points where $f$ takes finite values:

$$\mathrm{dom}(f) := \{\mathbf{x} \in \mathbb{R} \mid f(\mathbf{x}) < +\infty\}.$$

### B.1  PROOF OF LEMMA 2.1

The following proof follows [59].

*Proof.* We prove the conclusion based on the formulation (3). First we introduce augmented variables $\mathbf{z}$ and $\mathbf{z}_i, i = 1, 2, ..., M + 1$ and deduce the equivalent form of LL problem of (3),

$$\min_{\mathbf{x}, \mathbf{z}_i} \ \varphi(\mathbf{z}) + \sum_{i=1}^{M+1} \lambda_i R_i(\mathbf{z}_i) \ \ \text{s.t. } \mathbf{z} = A_t \mathbf{x} - \mathbf{b}_t, \ \mathbf{x} = \mathbf{z}_i, \ i = 1, 2, ..., M + 1 \tag{25}$$

Since $l, R_i$ are convex and the constraints are affine, strong duality holds under Slater's condition. If $\mathrm{ri}(\mathrm{dom} \, l \cap (\cap_{i=1}^{M+1} \mathrm{dom} \, R_i)) \neq \emptyset$, then (25) is equivalent to its Lagrangian dual problem:

$$\max_{\boldsymbol{\xi}, \boldsymbol{\rho}} \min_{\mathbf{x}, \mathbf{z}, \mathbf{z}_i} \ \varphi(\mathbf{z}) + \sum_{i=1}^{M+1} \lambda_i R_i(\mathbf{z}_i) - \boldsymbol{\xi}^T (A_t \mathbf{x} - \mathbf{b}_t - \mathbf{z}) + \sum_{i=1}^{M+1} \boldsymbol{\rho}_i^T (\mathbf{x} - \mathbf{z}_i),$$

where $\boldsymbol{\xi}$ is Lagrangian multiplier of constraint $A_t\mathbf{x} - \mathbf{b}_t = \mathbf{z}$, while $\boldsymbol{\rho}_i$ are those associated with constraints $\mathbf{x} = \mathbf{z}_i$. By adding the negative signs, we obtain

$$\max_{\boldsymbol{\xi},\boldsymbol{\rho}} - \max_{\mathbf{x},\mathbf{z},\mathbf{z}_i} -\varphi(\mathbf{z}) - \sum_{i=1}^{M+1} \lambda_i R_i(\mathbf{z}_i) + \boldsymbol{\xi}^T(A_t\mathbf{x} - \mathbf{b}_t - \mathbf{z}) - \sum_{i=1}^{M+1} \boldsymbol{\rho}_i^T(\mathbf{x} - \mathbf{z}_i).$$

The above problem can be further simplified as,

$$\begin{aligned} \max_{\boldsymbol{\xi},\boldsymbol{\rho}} \quad & -\varphi^*(\boldsymbol{\xi}) - \sum_{i=1}^{M+1} \lambda_i R_i^*(\tfrac{\boldsymbol{\rho}_i}{\lambda_i}) - \boldsymbol{\xi}^T\mathbf{b}_t. \\ \text{s.t.} \quad & A_t\boldsymbol{\xi} + \sum_{i=1}^{M+1} \boldsymbol{\rho}_i = \mathbf{0}. \end{aligned} \tag{26}$$

Meanwhile, leveraging the value function of the lower-level problem, the constraint of (3) is equivalent to

$$l(\mathbf{x}) + \sum_{i=1}^{M+1} \lambda_i R_i(\mathbf{x}) \leq \min_{\mathbf{x}}\{l(\mathbf{x}) + \sum_{i=1}^{M+1} \lambda_i R_i(\mathbf{x})\}. \tag{27}$$

From the equivalence of (25) and (26), (27) is further equivalent to

$$l(\mathbf{x}) + \sum_{i=1}^{M+1} \lambda_i R_i(\mathbf{x}) \quad \leq \quad \max_{\boldsymbol{\xi},\boldsymbol{\rho}}\{-\varphi^*(\boldsymbol{\xi}) - \sum_{i=1}^{M+1} \lambda_i R_i^*(\tfrac{\boldsymbol{\rho}_i}{\lambda_i}) - \boldsymbol{\xi}^T\mathbf{b}_t \mid A_t\boldsymbol{\xi} + \sum_{i=1}^{M+1} \boldsymbol{\rho}_i = \mathbf{0}\}. \tag{28}$$

Because the inequality in (28) holds if and only if there exists a feasible pair $(\boldsymbol{\xi}, \boldsymbol{\rho})$ satisfying (28), dropping the max operator, we obtain that the constraint in (3) is equivalent to

$$l(\mathbf{x}) + \sum_{i=1}^{M+1} \lambda_i R_i(\mathbf{x}) + \varphi^*(\boldsymbol{\xi}) + \sum_{i=1}^{M+1} \lambda_i R_i^*(\frac{\boldsymbol{\rho}_i}{\lambda_i}) + \boldsymbol{\xi}^T\mathbf{b}_t \leq 0,$$

$$A_t\boldsymbol{\xi} + \sum_{i=1}^{M+1} \boldsymbol{\rho}_i = \mathbf{0}.$$

We complete the proof. $\qquad\square$

## B.2 EXPLANATIONS FOR LEMMA 2.1

**(a) Explanations for equivalence:**

In Lemma 2.1, the phrase equivalent form denotes equivalence both in the set of minimizers and in the optimal objective value. We explain the reasons as follows.

Under Slater's condition, strong duality holds between the LL problem in (3) and its Fenchel dual, guaranteeing that their optimal values coincide. Our notion of equivalence derives precisely from this fact. Concretely, in the proof of Lemma 2.1, we clarify that the LL problem in (3) is equivalent to the relation

$$l(\mathbf{x}) + \sum_{i=1}^{M+1} \lambda_i R_i(\mathbf{x}) \leq \min_{\mathbf{x}}\{l(\mathbf{x}) + \sum_{i=1}^{M+1} \lambda_i R_i(\mathbf{x})\}.$$

The above inequality appears in (27). We reformulate is by replacing the right part with its Fenchel dual in (27)-(28). Therefore, the constraint enforces recovery of the LL solution and its dual multipliers introduced for (25).

As a result, if $(\mathbf{x}, \boldsymbol{\lambda}, \boldsymbol{\rho}, \boldsymbol{\xi})$ is the minimizer of (5), then $\mathbf{x}$ is the minimizer of (3). Moreover, since the objective $L(\mathbf{x})$ remains unchanged, the optimal value is also preserved.

**(b) Explanations for specific forms of the LL problems:**

We stress that our work is firmly centered on bilevel hyperparameter optimization, where regularization in the form of a sum of norms naturally arises in real applications. This is not an artificial construct but a practical necessity.

We emphasize that our reformulation strategy extends beyond sums of norms. Whenever the lower-level problem takes the form

$$\mathbf{y} \in \arg\min_{\mathbf{y}}\{g(\mathbf{x}, \mathbf{y}) + \sum_i g_i(\mathbf{x}, \mathbf{y})\},\ i \geq 1.$$

it can be reformulated via lower-level duality [59, Lemma 2.1], making both the reformulation (6) and penalty framework are both applicable.

### B.3 PROOF OF THEOREM 2.5

*Proof.* We adopt the convention $A(\mathbf{z}) = \frac{1}{2}\|A_t\boldsymbol{\xi} + \sum_{i=1}^{M+1}\boldsymbol{\rho}_i\|^2$. It is straightforward that $A(\mathbf{z}) \geq 0$. Let $\bar{\mathbf{z}}$ be any limit point of the sequence $\{\mathbf{z}^k\}$ and $\{\mathbf{z}^{k_j}\} \subset \{\mathbf{z}^k\}$ be the subsequence such that $\mathbf{z}^{k_j} \to \bar{\mathbf{z}}$.

Assume that $\mathbf{z}^*$ is a solution of the reformulation (6). Then it holds that $L(\mathbf{x}^*) \leq L(\mathbf{x})$ for all $\mathbf{z} = (\mathbf{x}, \boldsymbol{\lambda}, \boldsymbol{\rho}, \mathbf{r}, \boldsymbol{\xi}, s)$ feasible to (6). Note that the constraints in (6) subsume those in (8), so any point $\mathbf{z}$ feasible to (6) is also feasible to (8).

Since $\mathbf{z}^{k+1}$ is the minimizer of the problem (8) with $\beta_k$, it follows that

$$L(\mathbf{x}^{k+1}) + \beta_k(p(\mathbf{z}^{k+1}) + A(\mathbf{z}^{k+1})) \overset{(a)}{\leq} L(\mathbf{x}^*) + \beta_k(p(\mathbf{z}^*) + A(\mathbf{z}^*)) \overset{(b)}{\leq} L(\mathbf{x}^*). \tag{29}$$

Here, $(a)$ follows from the feasibility of $\mathbf{z}^{k+1}$ and $\mathbf{z}^*$ for the penalized problem (8) and the optimality of $\mathbf{z}^{k+1}$. Since $\mathbf{z}^*$ is feasible to (6), we have $p(\mathbf{z}^*) \leq 0$ and $A(\mathbf{z}^*) = 0$, and thus $(b)$ holds. Rearranging (29), we deduce that

$$p(\mathbf{z}^{k+1}) + A(\mathbf{z}^{k+1}) \leq \frac{1}{\beta_k}(L(\mathbf{x}^*) - L(\mathbf{x}^{k+1})). \tag{30}$$

**Proof for $L(\mathbf{x}^*) \leq L(\bar{\mathbf{x}})$:**

- Since the functions $p$ is lower semi-continuous and $A$ is continuous in $\mathbf{z}$, letting $k = k_j$ and taking $k_j \to \infty$, we obtain that
$$p(\bar{\mathbf{z}}) + A(\bar{\mathbf{z}}) \leq \lim_{k_j \to \infty} p(\mathbf{z}^{k_j}) + A(\mathbf{z}^{k_j}).$$

- From $\beta_k \to \infty$ as $k \to \infty$, we have
$$\frac{1}{\beta_k}(L(\mathbf{x}^*) - L(\mathbf{x}^{k+1})) \to 0, \text{ as } k \to \infty.$$

Combining these facts, (30) gives that

$$p(\bar{\mathbf{z}}) + A(\bar{\mathbf{z}}) \overset{(c)}{\leq} \lim_{k_j \to \infty} p(\mathbf{z}^{k_j}) + A(\mathbf{z}^{k_j}) \leq \lim_{k_j \to \infty} \frac{1}{\beta_{k_j}}(L(\mathbf{x}^*) - L(\mathbf{x}^{k_j})) = 0,$$

where $(c)$ is derived from Definition B.2. Therefore, we obtain that $p(\bar{\mathbf{z}}) + A(\bar{\mathbf{z}}) \leq 0$. Since the assumptions of Theorem 2.5 are consistent with those of Lemma 2.1, we obtain the following relation from the formulation of $p$ and (27)

$$p(\mathbf{z}) = l(\mathbf{x}) + \sum_{i=1}^{M+1} \lambda_i R_i(\mathbf{x}) - \min_{\mathbf{x}}\{l(\mathbf{x}) + \sum_{i=1}^{M+1} \lambda_i R_i(\mathbf{x})\},$$

which directly implies that $p(\mathbf{z}) \geq 0$ for all $\mathbf{z}$. Combined with $A(\mathbf{z}) \geq 0$ for all $\mathbf{z}$, we further deduce that

$$p(\bar{\mathbf{z}}) = 0, \ A(\bar{\mathbf{z}}) = 0. \tag{31}$$

Therefore, $\bar{\mathbf{z}}$ is feasible for (6). Since $\mathbf{z}^*$ is optimal for (6), it holds that $L(\mathbf{x}^*) \leq L(\bar{\mathbf{x}})$.

**Proof for $L(\bar{\mathbf{x}}) \leq L(\mathbf{x}^*)$:**

From the non-negativity of $p(\mathbf{z})$ and $A(\mathbf{z})$ for all $\mathbf{z}$, inequality (29) yields

$$L(\mathbf{x}^{k+1}) \leq L(\mathbf{x}^{k+1}) + \beta_k(p(\mathbf{z}^{k+1}) + A(\mathbf{z}^{k+1})) \leq L(\mathbf{x}^*),$$

which implies that $L(\mathbf{z}^{k+1}) \leq L(\mathbf{x}^*)$. Since $L$ is lower semicontinuous, letting $k = k_j$ and taking the limit as $j \to \infty$ in the inequality above, we obtain

$$L(\bar{\mathbf{z}}) \overset{(d)}{\leq} \lim_{k_j \to \infty} L(\mathbf{x}^{k_j}) \leq L(\mathbf{x}^*),$$

where $(d)$ is also derived from Definition B.2. Therefore, we have $L(\bar{\mathbf{x}}) \leq L(\mathbf{x}^*)$.

In summary, we deduce that $L(\bar{\mathbf{x}}) = L(\mathbf{x}^*)$ and $\bar{\mathbf{z}}$ is an optimal solution of (6). This completes the proof. □

### B.4 CONJUGATE FUNCTIONS FOR PROBLEMS LISTED IN TABLE 1

we calculate the closed-form expression of the conjugate functions of $\varphi$ in problems as follows:

For **least squares loss**, $\varphi^*(v) = \frac{1}{2}v^2$.

For **smoothed hinge loss**, $\varphi^*(v) = \frac{1}{2}v^2 + v$ if $-1 < v < 0$ and $\varphi^*(v) = \infty$ otherwise.

For **Huber loss**, $\varphi^*(v) = \frac{1}{2}v^2$ if $|v| \leq \delta$ and $\varphi^*(v) = \delta|v| - \frac{1}{2}\delta^2$ if $|v| < \delta$.

## C  EPIGRAPHICAL PROJECTIONS

In this section, we discuss the projection onto the cones in Algorithms 1 and 2. According to different cases detailed in Section 3.1 and 3.2, we discuss the projections when involving different norm regularizers. We summarize the computation cost of these projections in Appendix C.3.

### C.1  PROJECTIONS INVOLVING VECTOR NORMS

When $R_i$ represents different norm terms, the explicit forms of $\mathcal{K}_i$ and $\mathcal{K}_i^d$ defined in (9) are expressed as follows.

- $R_i(x) = \|\mathbf{x}\|_1$: $\mathcal{K}_i = \{(\mathbf{x}, r_i) \mid \|\mathbf{x}\|_1 \leq r_i\}, \mathcal{K}_i^d = \{(\boldsymbol{\rho}_i, \lambda_i) \mid \|\boldsymbol{\rho}_i\|_\infty \leq \lambda_i\}$.
- $R_i(x) = \|\mathbf{x}\|_2$: $\mathcal{K}_i = \{(\mathbf{x}, r_i) \mid \|\mathbf{x}\|_2 \leq r_i\}, \mathcal{K}_i^d = \{(\boldsymbol{\rho}_i, \lambda_i) \mid \|\boldsymbol{\rho}_i\|_2 \leq \lambda_i\}$.
- $R_i(x) = \|\mathbf{x}\|_\infty$: $\mathcal{K}_i = \{(\mathbf{x}, r_i) \mid \|\mathbf{x}\|_\infty \leq r_i\}, \mathcal{K}_i^d = \{(\boldsymbol{\rho}_i, \lambda_i) \mid \|\boldsymbol{\rho}_i\|_1 \leq \lambda_i\}$.

- $R_i(\mathbf{x}) = \frac{1}{2}\|\mathbf{x}\|_2^2$: $\mathcal{K}_i = \{(\mathbf{x}, r_i) \mid \|\mathbf{x}\|_2^2 \leq 2r_i\}$, $\mathcal{K}_i^d = \{(\boldsymbol{\rho}_i, \lambda_i, s) \mid \|\boldsymbol{\rho}_i\|_2^2 \leq 2\lambda_i s\}$.

**(1) Projection onto the epigraph of $\ell_2$ norm:**

**Proposition C.1.** *[101, Example 6.37] Let $L_2^n = \{(\mathbf{x}, t) \mid \|\mathbf{x}\|_2 \leq t\}$, for any $(\mathbf{x}, t) \in \mathbb{R}^{d_x} \times \mathbb{R}$, we have*

$$\mathrm{proj}_{L_2^n}((\mathbf{x}, t)) = \begin{cases} (\frac{\|\mathbf{x}\|_2 + t}{2\|\mathbf{x}\|_2}\mathbf{x}, \frac{\|\mathbf{x}\|_2 + t}{2}), & \|\mathbf{x}\|_2 \geq |t|, \\ (\mathbf{0}, 0), & t < \|\mathbf{x}\|_2 < -t, \\ (\mathbf{x}, t), & \|\mathbf{x}\|_2 \leq t. \end{cases}$$

We next turn to the epigraphical projection for other norms. To this end, we recall a general result on projections onto epigraphs of convex functions.

**Theorem C.2.** *[101, Theorem 6.36] Let $C = \mathrm{epi}(g) = \{(\mathbf{x}, t) \mid g(\mathbf{x}) \leq t\}$ where $g$ is convex. Then for any $(\mathbf{x}, t) \in \mathbb{R}^{d_x} \times \mathbb{R}$, it holds that*

$$\mathrm{proj}_C((\mathbf{x}, t)) = \begin{cases} (\mathbf{x}, t), & g(\mathbf{x}) \leq t, \\ (\mathrm{prox}_{\lambda^* g}(\mathbf{x}), t + \gamma^*), & g(\mathbf{x}) > t, \end{cases}$$

*where $\gamma^*$ is any positive root of the function*

$$\psi(\gamma) = g(\mathrm{prox}_{\gamma g}(\mathbf{x}) - \gamma - t).$$

*In addition, $\psi$ is nonincreasing.*

**(2) Projection onto the epigraph of $\ell_1$ norm:**

**Proposition C.3.** *[101, Example 6.38] Let $L_1^n = \{(\mathbf{x}, t) \mid \|\mathbf{x}\|_1 \leq t\}$, for any $(\mathbf{x}, t) \in \mathbb{R}^{d_x} \times \mathbb{R}$, we have*

$$\mathrm{proj}_{L_1^n}((\mathbf{x}, t)) = \begin{cases} (\mathbf{x}, t), & \|\mathbf{x}\|_1 \leq t, \\ (\mathcal{T}_{\gamma^*}(\mathbf{x}), t + \gamma^*), & \|\mathbf{x}\|_1 > t, \end{cases}$$

where $\mathcal{T}_\gamma = \mathrm{prox}_{\gamma\|\cdot\|_1}$ denotes the proximal of $\ell_1$-norm, defined as

$$\mathcal{T}_\gamma(y) = [|y| - \gamma]_+ \,\mathrm{sgn}(y) = \begin{cases} y - \gamma, & y \geq \gamma \\ 0, & |y| < \gamma, \\ y + \gamma, & y \leq -\gamma. \end{cases}$$

Here, $\lambda^*$ is any positive root of the nonincreasing function $\psi(\gamma) = \|\mathcal{T}_\gamma(\mathbf{x})\|_1 - \gamma - s$. In practice, the $\ell_1$ norm epigraphical projection can be computed in linear time using the quick-select algorithm proposed by [102].

**(3) Projection onto the epigraph of $\ell_\infty$ norm:**

It can be computed directly via the Moreau decomposition. Let $L_\infty^n = \{(\mathbf{x}, t) \mid \|\mathbf{x}\|_\infty \leq t\}$, then the projection is given by
$$\mathrm{proj}_{L_\infty^n}(\mathbf{x}, t) = (\mathbf{x}, t) - \mathrm{proj}_{L_1^n}(\mathbf{x}, t).$$

**(4) Projection onto the epigraph of squared $\ell_2$ norm:**

According to Theorem C.2, for any $(\mathbf{x}, t) \in \mathbb{R}^{d_x} \times \mathbb{R}$, we have

$$\mathrm{proj}_{\mathcal{K}_{M+1}}(\mathbf{x}, t) = \begin{cases} (\mathbf{x}, t), & \|\mathbf{x}\|_2^2 \leq 2t, \\ (\frac{\mathbf{x}}{1+\gamma^*}, t + \gamma^*), & \|\mathbf{x}\|_2^2 > 2t, \end{cases}$$

where $\gamma^*$ is any positive root of the nonincreasing function $\psi(\gamma) = (\frac{1}{2}\gamma + t)(1 + 2\gamma^2) - \|x\|_2^2$. Similar to $\ell_1$-norm epigraphic projection, it can also be effectively solved in linear time with quick-select algorithm proposed by [102].

**(5) Projection onto rotated second-order cones:**

For the rotated second-order cone $\mathcal{K}_{M+1}^d = \{(\boldsymbol{\rho}_i, \lambda_i, s) \mid \|\boldsymbol{\rho}_i\|_2^2 \leq 2\lambda_i s\}$ where $\boldsymbol{\rho} \in \mathbb{R}^{d_x}$, an equivalent representation is given by $\{(\boldsymbol{\rho}_i, \lambda_i, s) \mid \|(\boldsymbol{\rho}_i, \lambda_i, s)\|_2 \leq \lambda_i + s\}$. We introduce auxiliary variables $\mathbf{w} = (\boldsymbol{\rho}_i, \lambda_i, s) \in \mathbb{R}^{d_x + 2}$ and $t = \lambda_i + s \in \mathbb{R}$. In this way, the projection onto $\mathcal{K}_{M+1}^d$ for given $(\bar{\boldsymbol{\rho}}_i, \bar{\lambda}_i, \bar{s})$ is equivalent to the following optimization problem with $(\bar{\mathbf{w}}, \bar{t})$:

$$\min_{\mathbf{w}, t} \frac{1}{2}\|\mathbf{w} - \bar{\mathbf{w}}\|^2 + \frac{1}{2}(t - \bar{t})^2 \text{ s.t. } \|\mathbf{w}\|_2 \leq t, \mathbf{w}^T \mathbf{c}_0 = t,$$

where $\mathbf{c}_0 = (0, ..., 0, 1, 1) \in \mathbb{R}^{d_x + 2}$. The problem can be solved directly using the analytic solution provided in [103, Proposition 6.4].

**(6) Projections for block-wise regularization:**

When the regularization involves a group component-wise regularizers, i.e., $R_i(\mathbf{x}) = \|\mathbf{x}^{(i)}\|_{(t)}$, we observe that projection onto the set $\mathcal{K}_i$ and $\mathcal{K}_i^d$ corresponds to the $\ell_1$, $\ell_2$ or $\ell_\infty$-norm. The same projection applies to the vector $\boldsymbol{\rho} = (\boldsymbol{\rho}^{(1)}, ..., \boldsymbol{\rho}^{(M)})$. Specifically, we project each group independently and then assemble the full vector.

## C.2 PROJECTIONS INVOLVING MATRIX NORMS

Now we study the projection onto the epigraphs of nuclear norm $\|\cdot\|_*$ and spectral norm $\|\cdot\|_{op}$. Since our reformulation relies on conjugate functions and the conjugate of a norm is its dual norm, we need to take both into consideration.

For a matrix $X \in \mathbb{R}^{m \times n}$, the nuclear norm is defined as $\|X\|_* = \sum_{i=1}^{\min\{m,n\}} \sigma_i(X)$ and the spectral norm is defined as $\|X\|_{op} = \max_i \sigma_i(X)$, where $\sigma_i(X)$ is singular values for $X$. In this case, the explicit of $\mathcal{K}_i$ and $\mathcal{K}_i^d$ is given by

- $R_i(X) = \|X\|_*$: $\mathcal{K}_i = \{(X, r_i) \mid \|X\|_* \leq r_i\}$, $\mathcal{K}_i^d = \{(\boldsymbol{\rho}_i, \lambda_i) \mid \|\boldsymbol{\rho}_i\|_{op} \leq \lambda_i\}$.

**(1) Projection onto the epigraph of nuclear norm:**

Given a matrix $A \in \mathbb{R}^{m \times n}$ and a scalar $t$, the projection onto the epigraph of the nuclear norm $\{X \in \mathbb{R}^{m \times n}, \tau \geq 0 \mid \|X\|_* \leq \tau\}$ involves solving the following optimization problem

$$\min_{X, \tau \geq 0} \frac{1}{2}\|X - A\|_F^2 + \frac{1}{2}\|t - \tau\|^2 \text{ s.t. } \|X\|_* \leq \tau,$$

where $\|\cdot\|_F$ denotes Frobenius norm of a matrix.

- If $\|A\|_* \leq t$, the point $(A, t)$ already lies in the epigraph and the projection is simply $(X, \tau) = (A, t)$.
- If $\|A\|_* > t$, we first compute the singular value decomposition of $A$ as $A = U\Sigma V$, where $\Sigma = \text{diag}\{\sigma_1, \sigma_2, ..., \sigma_r\}$ is the single value matrix of $A$ and $U \in \mathbb{R}^{m \times r}$, $V \in \mathbb{R}^{n \times r}$. According to [1,Theorem 6.36], the projected matrix is obtained by soft-thresholding the singular values:

$$\bar{\sigma}_i = \max(\sigma_i - \gamma^*, 0), i = 1, 2, ..., r,$$

where $\gamma^*$ is determined by the equation $\sum_{i=1}^{r} \max(\sigma_i - \gamma, 0) = t + \gamma$. This equation is typically solved efficiently via a bisection search. Subsequently, we obtain the solution $\tau^* = t + \gamma^*$ and reconstruct the projected matrix as $X^* = U\bar{\Sigma}V^T$ where $\bar{\Sigma} = \text{diag}\{\bar{\sigma}_1, \bar{\sigma}_2, ..., \bar{\sigma}_r\}$. The projected pair $(X^*, \tau^*)$ is the closest point to $(A, t)$ in the epigraph of the nuclear norm.

**(2) Projection onto the epigraph of spectral norm:**

Given a matrix $A \in \mathbb{R}^{m \times n}$ and a scalar $t$, now we consider projection onto the epigraph of the nuclear norm $\{X \in \mathbb{R}^{m \times n}, \tau \geq 0 \mid \|X\|_{op} \leq \tau\}$

- If $\|A\|_{op} \leq t$, the point $(A, t)$ already lies in the epigraph and the projection is simply $(X, \tau) = (A, t)$.

- If $\|A\|_{op} > t$, we first compute the singular value decomposition of $A$ as $A = U\Sigma V$, where $\Sigma = \text{diag}\{\sigma_1, \sigma_2, ..., \sigma_r\}$ is the single value matrix of $A$ and $U \in \mathbb{R}^{m \times r}$, $V \in \mathbb{R}^{n \times r}$.

  Since the epigraph of the spectral norm is defined by the constraint $\|X\|_{op} = \max_i \sigma_i(X) \leq \tau$, we need to adjust the singular values so that the largest does not exceed the new scalar $\tau^*$ as

  $$\tilde{\sigma}_i = \min\{\sigma_i, \tau^*\} \quad \text{for } i = 1, 2, \ldots, r.$$

  To determine $\tau^*$, we solve the one-dimensional optimization problem

  $$\min_{\tau \geq 0} \frac{1}{2} \sum_{i:\, \sigma_i > \tau} (\sigma_i - \tau)^2 + \frac{1}{2}(\tau - t)^2.$$

  In practice, the optimal $\tau^*$ can be efficiently computed using a bisection search. Subsequently, we reconstruct the projected matrix as $X^* = U\tilde{\Sigma}V^T$ where $\tilde{\Sigma} = \text{diag}\{\tilde{\sigma}_1, \tilde{\sigma}_2, \ldots, \tilde{\sigma}_r\}$.

  The projected pair $(X^*, \tau^*)$ is the closest point to $(A, t)$ in the epigraph of the spectral norm.

From the above discussions, it is evident that the projections can be computed efficiently.

## C.3 COMPUTATION COST

In this subsection, we denote the dimension of vector input as $\mathbf{x} \in \mathbb{R}^{d_x} = \mathbb{R}^n$ and matrix input as $X \in \mathbb{R}^{d_x} = \mathbb{R}^{m \times n}$.

The projections onto the $\ell_2$ norm cones and rotated second-order cones have closed-form solutions, whose cost is $\mathcal{O}(n)$. For the other norms, which do not admit explicit epigraphic projection formulas, the projection can be computed by finding the root of a nonincreasing scalar function $\psi(\gamma)$. These procedures leverage efficient quick-select routines to ensure fast computation. In summary, we deduce that

- For epigraphic projection for vector inputs, the overall runtime is $\tilde{\mathcal{O}}(n)$, where the tilde hides logarithmic factors.

- For epigraphic projection for matrix inputs, the dominant cost arises from computing the SVD, which takes $\tilde{\mathcal{O}}(mn \min\{m, n\})$, followed by a root-finding step of complexity $\tilde{\mathcal{O}}(r)$ with $r = \text{rank}(X)$. Moreover, for nuclear-norm or spectral-norm projections, only the nonzero singular components are needed, so an economy-size SVD is not only sufficient but standard and computationally preferable [102].

- For epigraphic projection of group norms, the total cost is $\mathcal{O}(n)$.

Table 4: Computation cost of epigraphical projections for vector $\mathbf{x} \in \mathbb{R}^n$ or matrix $X \in \mathbb{R}^{m \times n}$ with $r = \text{rank}(X)$.

| Projection Type | Complexity |
|---|---|
| $\ell_2$ norm | $O(n)$ |
| $\ell_1$ norm | $O(n)$ (quick-select) |
| $\ell_\infty$ norm | $O(n)$ (quick-select) |
| Squared $\ell_2$ norm | $O(n)$ |
| Nuclear norm | $O(mnr)$ |
| Spectral norm | $O(mnr)$ |
| Rotated SOC | $O(n)$ |

More detailed results are provided in the following table. Since each projection uses the same low-dimensional routine, the overall computation remains efficient.

Based on Table 4, we present a comparison of the per-iteration computational costs of our method and other single-loop Hessian-free methods for BLO with nonsmooth LL problems. When the LL variable is $x \in R^n$ with inputs $A \in R^{d \times n}$ and $b \in R^d$ or a matrix $x \in R^{m \times n}$ with rank of $r$, the corresponding computational costs are summarized in the table below.

Table NEW1: per-iteration computation cost. Here, GD stands for gradient descent.

| Methods | Vector Variable | | Matrix variable | |
|---|---|---|---|---|
| | Cost(Nonsmooth Terms) | Cost(GD) | Cost(Nonsmooth Terms) | Cost(GD) |
| VF-iDCA[57] | off-the-shelf solvers | | off-the-shelf solvers | |
| LDMMA[59] | off-the-shelf solvers | | off-the-shelf solvers | |
| MEHA[63] | $\mathcal{O}(dn)$ | $\mathcal{O}(d^2 n)$ | $\mathcal{O}(mn \min\{m, n\})$ | $\mathcal{O}(mn \min\{m, n\})$ |
| LV-HBA[64] | $\mathcal{O}(dn)$ | $\mathcal{O}(d^2 n)$ | $\mathcal{O}(mn \min\{m, n\})$ | $\mathcal{O}(mn \min\{m, n\})$ |
| BiC-GAFFA[65] | $\mathcal{O}(dn)$ | $\mathcal{O}(d^2 n)$ | $\mathcal{O}(mn \min\{m, n\})$ | $\mathcal{O}(mn \min\{m, n\})$ |
| LDPM(Ours) | $\mathcal{O}(n)$ | $\mathcal{O}(dn)$ | $\mathcal{O}(mnr)$ | $\mathcal{O}(mn \min\{m, n\})$ |

Because LDPM handles nonsmooth regularizers via explicit epigraphical projection, its gradient computation involves only one first-order update of the penalty objective. In contrast, Moreau-envelope-based and gap-function-based methods compute the proximal operators of the nonsmooth termspenalty terms, which requires Jacobian operations or full SVDs. Therefore, LDPM achieves the lowest computational cost among existing single-loop Hessian-free methods for BLO with a nonsmooth LL problem.

# D  PROOFS AND EXPLANATIONS FOR SECTION 3

In this section, we provide additional explanations and the proofs for the convergence results of our proposed algorithms in Section 3.

## D.1  INITIALIZATION OF ALGORITHMS 1 AND 2

We initialize the starting point by following the algorithms for BLO proposed in [57; 59; 64]. For Algorithm 1, given the input $\boldsymbol{\lambda}^0, \boldsymbol{\xi}^0$, we initialize $\mathbf{x}^0$ by solving the LL problem of (3). The remaining initial variables are set as $r_i^0 = R_i(\mathbf{x})$, $\boldsymbol{\rho}^0 = -\nabla l(\mathbf{x}^0)$ and $s^0 = \|\boldsymbol{\rho}^0\|^2 / 2\lambda_1^0$. For Algorithm 2, given the input $\boldsymbol{\lambda}^0, \boldsymbol{\xi}^0$, we

also initialize $\mathbf{x}^0$ with solving the LL problem of (3). The other initial variables are set as $r_i^0 = R_i(\mathbf{x}^0)$, $\boldsymbol{\rho}_i^0 = -\frac{1}{M+1} A_t \boldsymbol{\xi}^0$ and $s^0 = \|\boldsymbol{\rho}_{M+1}^0\|^2 / 2\lambda_{M+1}^0$.

This initialization strategy ensures a feasible starting point for the corresponding reformulation of original BLO, thereby facilitating convergence and enhancing the overall efficiency of the optimization process.

## D.2 EXPLANATIONS FOR MERIT FUNCTIONS

To initiate the proof of the convergence results, we establish the rationale for selecting $\phi_{res}^k$ and $\phi_{fea}$ as the merit measures. Note that $\phi_{res}^k$ and $\phi_{fea}$ in Section 3.1 and 3.2 are both defined based on the penalized formulation (8) within a unified framework as follows:

$$\phi_{res}^k(\mathbf{z}) := \text{dist}\left(0, \nabla_{\mathbf{z}} F_k(\mathbf{z}) + \mathcal{N}_\mathcal{K}(\mathbf{z})\right), \tag{32}$$

$$\phi_{fea}(\mathbf{z}) := \max\{p(\mathbf{x}, \boldsymbol{\lambda}, \mathbf{r}, \boldsymbol{\xi}, s), \|A_t \boldsymbol{\xi} + \sum_{i=1}^{M+1} \boldsymbol{\rho}_i\|\}, \tag{33}$$

where $\mathcal{K} = (\mathcal{K}_1 \cap \cdots \cap \mathcal{K}_{M+1}) \times \mathcal{K}_1^d \times \cdots \mathcal{K}_{M+1}^d$. For the case of single-round global regularization discussed in Section 3.1, the set $\mathcal{K}$ reduces to $\mathcal{K} = \mathcal{K}_1 \times \mathcal{K}_1^d$ and $(\boldsymbol{\rho}_1, ..., \boldsymbol{\rho}_{M+1})$ is replaced by a single $\boldsymbol{\rho}$.

From Lemma 2.1, we know that (5) is a direct reformulation of (3). For convenience, we simplify the left hand of the first constraint as:

$$F(\mathbf{x}, \boldsymbol{\lambda}, \boldsymbol{\rho}, \boldsymbol{\xi}) = l(\mathbf{x}) + \sum_{i=1}^{M+1} \lambda_i R_i(\mathbf{x}) + \varphi^*(\boldsymbol{\xi}) + \sum_{i=1}^{M+1} \lambda_i R_i^*(\frac{\boldsymbol{\rho_i}}{\lambda_i}) + \boldsymbol{\xi}^T \mathbf{b}_t.$$

Similar to (8), we construct the penalized formulation for (5) as follows,

$$\min_{\mathbf{z}} \ L(\mathbf{x}) + \beta_k F(\mathbf{x}, \boldsymbol{\lambda}, \boldsymbol{\rho}, \boldsymbol{\xi}) + \frac{\beta_k}{2} \|A_t \boldsymbol{\xi} + \sum_{i=1}^{M+1} \boldsymbol{\rho}_i\|^2, \tag{34}$$

where $\beta_k$ serves as the penalty parameter.

**Proposition D.1.** *If $\phi_{fea}(\mathbf{z}) = 0$, then $(\mathbf{x}, \boldsymbol{\lambda}, \boldsymbol{\rho}, \boldsymbol{\xi})$ is a feasible point to (5). Moreover, if $\phi_{fea}(\mathbf{z}) = 0$ and $\phi_{res}(\mathbf{z}) = 0$ both hold, then $(\mathbf{x}, \boldsymbol{\lambda}, \boldsymbol{\rho}, \boldsymbol{\xi})$ is a stationary point of (34).*

*Proof.* **(a) When $\phi_{fea} = 0$ holds:**

From the non-negativity of the function $p$ and $\| \cdot \|^2$, if $\phi_{fea}(\mathbf{z}) = 0$, it holds that $p(\mathbf{x}, \boldsymbol{\lambda}, \mathbf{r}, \boldsymbol{\xi}, s) = 0$ and $A_t \boldsymbol{\xi} + \sum_{i=1}^{M+1} \boldsymbol{\rho}_i = \mathbf{0}$

According to the constraints of (8), we know that

$$R_i(\mathbf{x}) \leq r_i, \ i = 1, ..., M+1,$$
$$R_i^*(\frac{\boldsymbol{\rho}_i}{\lambda_i}) = 0, \ i = 1, ..., M.$$

Additionally, we restore $\lambda_{M+1} R^*_{M+1}(\frac{\boldsymbol{\rho}_{M+1}}{\lambda_{M+1}})$ with the inequality $\frac{\|\boldsymbol{\rho}_{M+1}\|_2^2}{2\lambda_{M+1}} \leq s$. Consequently, we observe that

$$
\begin{aligned}
F(\mathbf{x}, \boldsymbol{\lambda}, \boldsymbol{\rho}, \boldsymbol{\xi}) &= l(\mathbf{x}) + \sum_{i=1}^{M+1} \lambda_i R_i(\mathbf{x}) + \varphi^*(\boldsymbol{\xi}) + \sum_{i=1}^{M+1} \lambda_i R_i^*(\frac{\boldsymbol{\rho_i}}{\lambda_i}) + \boldsymbol{\xi}^T \mathbf{b}_t \\
&= l(\mathbf{x}) + \sum_{i=1}^{M+1} \lambda_i R_i(\mathbf{x}) + \varphi^*(\boldsymbol{\xi}) + \lambda_{M+1} R_{M+1}^*(\frac{\boldsymbol{\rho}_{M+1}}{\lambda_{M+1}}) + \boldsymbol{\xi}^T \mathbf{b}_t \\
&\leq l(\mathbf{x}) + \sum_{i=1}^{M+1} \lambda_i r_i + \varphi^*(\boldsymbol{\xi}) + \boldsymbol{\xi}^T \mathbf{b}_t + s \\
&= p(\mathbf{x}, \boldsymbol{\lambda}, \mathbf{r}, \boldsymbol{\xi}, s) = 0,
\end{aligned}
$$

which implies that $(\mathbf{x}, \boldsymbol{\lambda}, \boldsymbol{\rho}, \boldsymbol{\xi})$ is feasible to (5).

**(b) When $\phi_{res}^k(\mathbf{z}) = 0$ and $\phi_{fea}(\mathbf{z}) = 0$ both hold:**

In this part, we use Moreau-Rockafellar theorem [104, Theorem 23.8] to calculate the sum rule of subdifferentials. If $f_1$ and $f_2$ are convex and lower continuous at $x$ and $f_2$ is differentiable at $x \in \text{int}(\text{dom}(f_1)) \cap \text{int}(\text{dom}(f_2))$, then it holds that

$$
\partial(f_1 + f_2)(x) \subset \partial f_1(x) + \partial f_2(x).
$$

We analyze $\phi_{res}^k(\mathbf{z}) = 0$ for each component of $\mathbf{z}$.

- For $\mathbf{x}$ and $\mathbf{r}$, we have

$$
-(\nabla L(\mathbf{x}) + \beta_k \nabla l(\mathbf{x}), \beta_k \boldsymbol{\lambda}) \in \mathcal{N}_{\mathcal{K}_1 \cap \cdots \cap \mathcal{K}_{M+1}}(\mathbf{x}, \mathbf{r}), \tag{35}
$$

  where $\mathcal{K}_i = \{(\mathbf{x}, \mathbf{r}) \mid R_i(\mathbf{x}) \leq r_i\}$. Let $\partial R_i$ denote the limiting subdifferential of the function $R_i$ [69]. According to the definition of the normal cone of inequality constraints [105; 106] and the definition of $\mathcal{K}_i$ in (9), we know that

$$
\begin{aligned}
\mathcal{N}_{\mathcal{K}_1 \cap \cdots \cap \mathcal{K}_{M+1}}(\mathbf{x}, \mathbf{r}) &= \text{cone}\{(\partial R_i(\mathbf{x}), -1), i = 1, ..., M+1\} \\
&= \{\sum_{i=1}^{M+1} t_i(\partial R_i(\mathbf{x}), -1) \mid t_i \geq 0\},
\end{aligned}
$$

  where cone denotes the conic hull of a set. Combining with (35), we obtain

$$
0 \in \nabla L(\mathbf{x}) + \beta_k \nabla l(\mathbf{x}) + \beta_k \sum_{i=1}^{M+1} \lambda_i \partial R_i(\mathbf{x}). \tag{36}
$$

- For $\boldsymbol{\xi}$, we have

$$
\nabla \varphi^*(\boldsymbol{\xi}) + \mathbf{b}_t + A_t^T(A_t \boldsymbol{\xi} + \sum_{i=1}^{M+1} \boldsymbol{\rho}_i) = 0. \tag{37}
$$

- For $(\boldsymbol{\rho}_i, \lambda_i), i = 1, ..., M$, we have

$$
-(A_t \boldsymbol{\xi} + \sum_{i=1}^{M+1} \boldsymbol{\rho}_i, r_i) \in \mathcal{N}_{\mathcal{K}_i^d}(\boldsymbol{\rho}_i, \lambda_i), i = 1, ..., M,
$$

where $\mathcal{K}_i^d = \{(\boldsymbol{\rho}_i, \lambda_i) \mid \|\boldsymbol{\rho}_i\|_{*(i)} \leq \lambda_i\}$. From (28) and the definition of $p$, we know that $F(\mathbf{x}, \boldsymbol{\lambda}, \boldsymbol{\rho}, \boldsymbol{\xi}) \geq 0$ for all $(\mathbf{x}, \boldsymbol{\lambda}, \boldsymbol{\rho}, \boldsymbol{\xi})$. If $\phi_{fea}(\mathbf{z}) = 0$, the following chain of inequalities holds:

$$0 \leq F(\mathbf{x}, \boldsymbol{\lambda}, \boldsymbol{\rho}, \boldsymbol{\xi}) \leq p(\mathbf{x}, \boldsymbol{\lambda}, \mathbf{r}, \boldsymbol{\xi}, s) \leq 0,$$

which naturally reduces to equalities. Consequently, we have $F(\mathbf{x}, \boldsymbol{\lambda}, \boldsymbol{\rho}, \boldsymbol{\xi}) = p(\mathbf{x}, \boldsymbol{\lambda}, \mathbf{r}, \boldsymbol{\xi}, s)$, implying that $R_i(\mathbf{x}) = r_i, i = 1, ..., M$. Therefore, we obtain that

$$-(A_t \boldsymbol{\xi} + \sum_{i=1}^{M+1} \boldsymbol{\rho}_i, R_i(\mathbf{x})) \in \mathcal{N}_{\mathcal{K}_i^d}(\boldsymbol{\rho}_i, \lambda_i), i = 1, ..., M,$$

Meanwhile, we note that for $i = 1, ..., M$, $R_i^*$ is the indicator function of the set $\{\|\mathbf{y}\|_{*(i)} \leq 1\}$. Combining with the fact that the normal cone is equivalent to the subdifferential of indicator function, for the variables $\boldsymbol{\rho}_i$ and $\lambda_i$, the above formulation implies that

$$-(A_t \boldsymbol{\xi} + \sum_{i=1}^{M+1} \boldsymbol{\rho}_i) \in \partial_{\boldsymbol{\rho}_i} I_{\{\|\boldsymbol{\rho}\|_{*(i)} \leq \lambda_i\}} = \partial_{\boldsymbol{\rho}_i} I_{\{\|\boldsymbol{\rho}\|_{*(i)}/\lambda_i \leq 1\}} \overset{(*)}{=} \partial_{\boldsymbol{\rho}_i} \left[\lambda_i R_i^*(\frac{\boldsymbol{\rho}_i}{\lambda_i})\right]. \quad (38)$$

$$\begin{aligned} -R_i(\mathbf{x}) &\in \partial_{\lambda_i} I_{\{\|\boldsymbol{\rho}\|_{*(i)} \leq \lambda_i\}} \overset{(a)}{=} \partial_{\lambda_i} I_{\{\|\boldsymbol{\rho}\|_{*(i)} \leq \lambda_i\}} + I_{\{\|\boldsymbol{\rho}\|_{*(i)} \leq \lambda_i\}} \\ &= \partial_{\lambda_i} I_{\{\|\boldsymbol{\rho}\|_{*(i)} \leq \lambda_i\}} + R_i^*(\frac{\boldsymbol{\rho}_i}{\lambda_i}) \overset{(*)}{=} \partial_{\lambda_i} \left[\lambda_i R_i^*(\frac{\boldsymbol{\rho}_i}{\lambda_i})\right], \end{aligned} \quad (39)$$

where $(a)$ follows the fact $\|\boldsymbol{\rho}\|_{*(i)} \leq \lambda_i$ and $(*)$ holds from the direct calculation of the subdifferential.

- For $(\boldsymbol{\rho}_{M+1}, \lambda_{M+1}, s)$, we have

$$-(A_t \boldsymbol{\xi} + \sum_{i=1}^{M+1} \boldsymbol{\rho}_i, r_{M+1}, 1) \in \mathcal{N}_{\mathcal{K}_{M+1}^d}(\boldsymbol{\rho}_{M+1}, \lambda_{M+1}, s),$$

where $\mathcal{K}_{M+1}^d = \{(\boldsymbol{\rho}_{M+1}, \lambda_{M+1}, s) \mid \|\boldsymbol{\rho_{M+1}}\|_2^2 \leq 2\lambda_{M+1}s\}$. Similar to the deduction for $(\boldsymbol{\rho}_i, \lambda_i)$ in (38) and (39), we can obtain

$$\begin{aligned} -(A_t \boldsymbol{\xi} + \sum_{i=1}^{M+1} \boldsymbol{\rho}_i) &\in \partial_{\boldsymbol{\rho}_{M+1}} \left[\lambda_{M+1} R_{M+1}^*(\frac{\boldsymbol{\rho}_{M+1}}{\lambda_{M+1}})\right], \\ -R_{M+1}(\mathbf{x}) &\in \partial_{\lambda_{M+1}} \left[\lambda_{M+1} R_{M+1}^*(\frac{\boldsymbol{\rho}_{M+1}}{\lambda_{M+1}})\right]. \end{aligned} \quad (40)$$

In summary, we find that the equations (36), (37), (38), (39) and (40) coincide with the stationary conditions of (34). Therefore, we conclude that $(\mathbf{x}, \boldsymbol{\lambda}, \boldsymbol{\xi}, \boldsymbol{\rho})$ is a stationary point of (34).

$\square$

From deduction (27) and (28), we conclude that $\phi_{fea}(\mathbf{z}) = 0$ implies

$$l(\mathbf{x}) + \sum_{i=1}^{M+1} \lambda_i R_i(\mathbf{x}) = \min_{\mathbf{x}}\{l(\mathbf{x}) + \sum_{i=1}^{M+1} \lambda_i R_i(\mathbf{x})\}.$$

Following the reasoning in Theorem 2.5, we conclude that as $\beta_k \to \infty$, any limit point of the sequence of optimal solutions to (34) with $\beta_k$ is an optimal solution of (5). According to (36), we further obtain that

$$\text{dist}(0, \nabla l(\mathbf{x}) + \sum_{i=1}^{M+1} \lambda_i \partial R_i(\mathbf{x})) \leq \frac{1}{\beta_k} \|\nabla L(\mathbf{x})\| \to 0,$$

as $\beta_k \to \infty$. These results demonstrate that $\phi_{res}^k$ and $\phi_{fea}$ can effectively character the optimality condition of the LL problem in (3). In summary, the selection of $\phi_{res}^k$ and $\phi_{fea}$ is reasonable.

We provide the proofs for non-asymptotic convergence of Algorithm 1 and 2 in the subsequent sections.

### D.3 PROOF OF THEOREM 3.5

We first recall the update for the variables of $\mathbf{z}$ in Algorithm 1 as follows. We calculate the update directions of $\mathbf{z}$ as $\mathbf{d}_{\mathbf{z}}^k = (\mathbf{d}_{\mathbf{x}}^k, \mathbf{d}_{\boldsymbol{\lambda}}^k, \mathbf{d}_{\boldsymbol{\rho}}^k, \mathbf{d}_{\mathbf{r}}^k, \mathbf{d}_{\boldsymbol{\xi}}^k, \mathbf{d}_s^k)$, where

$$
\begin{aligned}
\mathbf{d}_{\mathbf{x}}^k &= \tfrac{1}{\beta_k}\nabla L(\mathbf{x}^k) + \nabla l(\mathbf{x}^k), \\
\mathbf{d}_{\boldsymbol{\xi}}^k &= \nabla\varphi^*(\boldsymbol{\xi}^k) + \mathbf{b}_t + A_t^T(A_t\boldsymbol{\xi}^k + \boldsymbol{\rho}^k), \\
\mathbf{d}_{\boldsymbol{\lambda}}^k &= \mathbf{r}^k, \ \mathbf{d}_{\mathbf{r}}^k = \boldsymbol{\lambda}^k, \ d_s^k = 1, \\
\mathbf{d}_{\boldsymbol{\rho}}^k &= A_t\boldsymbol{\xi}^{k+1} + \boldsymbol{\rho}^k.
\end{aligned}
\tag{41}
$$

With these directions, the gradient descent step is performed as

$$
\bar{\mathbf{z}}^{k+1} = \mathbf{z}^k - e_k\mathbf{d}_{\mathbf{z}}^k.
$$

For $\bar{\mathbf{z}}^{k+1} = (\bar{\mathbf{x}}^{k+1}, \bar{\boldsymbol{\lambda}}^{k+1}, \bar{\boldsymbol{\rho}}^{k+1}, \bar{\mathbf{r}}^{k+1}, \bar{\boldsymbol{\xi}}^{k+1}, \bar{s}^{k+1})$, we subsequently apply the projection

$$
\mathbf{z}^{k+1} = \mathrm{proj}_{\mathcal{K}}(\bar{\mathbf{z}}^{k+1}).
\tag{42}
$$

Note that the variable $\boldsymbol{\xi}$ is not involved in the projection step and thus it is evolved directly as $\boldsymbol{\xi}^{k+1} = \bar{\boldsymbol{\xi}}^{k+1}$.

Next, we discuss the sufficient decrease property for Algorithm 1.

**Lemma D.2.** *Suppose Assumptions 3.1, 3.2 hold. For $k \in \mathbb{N}$, let $\{\mathbf{z}^k\}$ be generated from Algorithm 1. Under Assumptions 3.1, we let $\underline{L} := \inf_{\mathbf{x}} L(\mathbf{x}) > -\infty$. Define $V_k = \frac{1}{\beta_k}(F_k(\mathbf{z}^k) - \underline{L})$, then the following inequality holds:*

$$
\begin{aligned}
V_{k+1} - V_k \ \leq \ & \left( \tfrac{\alpha_L + \beta_k\|A_t\|_2^2\alpha_p}{2\beta_k} - \tfrac{1}{e_k} \right) \|\mathbf{x}^{k+1} - \mathbf{x}^k\|^2 + \left( \tfrac{1}{2} - \tfrac{1}{e_k} \right) \|\boldsymbol{\rho}^{k+1} - \boldsymbol{\rho}^k\|^2 \\
& - \tfrac{1}{e_k}\|s^{k+1} - s^k\|^2 + \left( \tfrac{1}{2} - \tfrac{1}{e_k} \right)(\|\boldsymbol{\lambda}^{k+1} - \boldsymbol{\lambda}^k\|^2 + \|\mathbf{r}^{k+1} - \mathbf{r}^k\|^2) \\
& + \left( \tfrac{\alpha_d + \|A_t\|_2^2}{2} - \tfrac{1}{e_k} \right) \|\boldsymbol{\xi}^{k+1} - \boldsymbol{\xi}^k\|^2.
\end{aligned}
\tag{43}
$$

*Proof.* Given Assumption 3.2 that $\varphi$ is $\alpha_p$-smooth, we know that $l$ is $\|A_t\|_2^2\alpha_p$-smooth. By applying the sufficient decrease lemma [101, Lemma 5.7], we obtain that

$$
\begin{aligned}
\tfrac{1}{\beta_k}L(\mathbf{x}^{k+1}) + l(\mathbf{x}^{k+1}) \leq & \tfrac{1}{\beta_k}L(\mathbf{x}^k) + l(\mathbf{x}^k) + \langle \tfrac{1}{\beta_k}\nabla L(\mathbf{x}^k) + \nabla l(\mathbf{x}^k), \mathbf{x}^{k+1} - \mathbf{x}^k \rangle \\
& + \tfrac{1}{2}(\tfrac{1}{\beta_k}\alpha_L + \|A_t\|_2^2\alpha_p)\|\mathbf{x}^{k+1} - \mathbf{x}^k\|^2.
\end{aligned}
$$

Based on the convexity of the cones and the second projection theorem [101, Theorem 6.41], we have

$$
\langle (\bar{\mathbf{x}}^{k+1}, \bar{\mathbf{r}}^{k+1}) - (\mathbf{x}^{k+1}, \mathbf{r}^{k+1}), (\mathbf{x}^k, \mathbf{r}^k) - (\mathbf{x}^{k+1}, \mathbf{r}^{k+1}) \rangle \leq 0.
$$

We combine the above inequalities and the same derivation for $\mathbf{r}$, it holds that

$$
\begin{aligned}
& \tfrac{1}{\beta_k}L(\mathbf{x}^{k+1}) + l(\mathbf{x}^{k+1}) + \langle \boldsymbol{\lambda}^{k+1}, \mathbf{r}^{k+1} - \mathbf{r}^k \rangle \\
\leq \ & \tfrac{1}{\beta_k}L(\mathbf{x}^k) + l(\mathbf{x}^k) + \left( \tfrac{\alpha_L + \beta_k\|A_t\|_2^2\alpha_p}{2\beta_k} - \tfrac{1}{e_k} \right)\|\mathbf{x}^{k+1} - \mathbf{x}^k\|^2 + \left( \tfrac{1}{2} - \tfrac{1}{e_k} \right)\|\mathbf{r}^{k+1} - \mathbf{r}^k\|^2.
\end{aligned}
$$

Subtracting $\frac{1}{\beta_k}\underline{L}$ from both sides of the inequality, we obtain

$$\frac{1}{\beta_k}(L(\mathbf{x}^{k+1}) - \underline{L}) + l(\mathbf{x}^{k+1}) + \langle \boldsymbol{\lambda}^{k+1}, \mathbf{r}^{k+1} - \mathbf{r}^k \rangle$$
$$\leq \frac{1}{\beta_k}(L(\mathbf{x}^k)) - \underline{L}) + l(\mathbf{x}^k) + \left( \frac{\alpha_L + \beta_k \|A_t\|_2^2 \alpha_p}{2\beta_k} - \frac{1}{e_k} \right) \|\mathbf{x}^{k+1} - \mathbf{x}^k\|^2 + \left( \frac{1}{2} - \frac{1}{e_k} \right) \|\mathbf{r}^{k+1} - \mathbf{r}^k\|^2.$$

Given $\beta_k = \underline{\beta}(1 + k)^p$, we have $\frac{1}{\beta_{k+1}} \leq \frac{1}{\beta_k}$. From Assumption 3.1, let $L(\mathbf{x}^K) - \underline{L} \geq 0$ holds for all $k$, so we have $\frac{1}{\beta_{k+1}}(L(\mathbf{x}^{k+1}) - \underline{L}) \leq \frac{1}{\beta_k}(L(\mathbf{x}^{k+1}) - \underline{L})$. Then we can derive that

$$\frac{1}{\beta_{k+1}}(L(\mathbf{x}^{k+1}) - \underline{L}) + l(\mathbf{x}^{k+1}) + \langle \boldsymbol{\lambda}^{k+1}, \mathbf{r}^{k+1} - \mathbf{r}^k \rangle$$
$$\leq \frac{1}{\beta_k}(L(\mathbf{x}^k) - \underline{L}) + l(\mathbf{x}^k) + \left( \frac{\alpha_L + \beta_k \|A_t\|_2^2 \alpha_p}{2\beta_k} - \frac{1}{e_k} \right) \|\mathbf{x}^{k+1} - \mathbf{x}^k\|^2 + \left( \frac{1}{2} - \frac{1}{e_k} \right) \|\mathbf{r}^{k+1} - \mathbf{r}^k\|^2.$$
$$(44)$$

The same derivation process applies to $\boldsymbol{\rho}, \lambda_i, r_i$, leading to the following results:

$$\|A_t \boldsymbol{\xi}^{k+1} + \boldsymbol{\rho}^{k+1}\|^2 + \langle \boldsymbol{\lambda}^{k+1} - \boldsymbol{\lambda}^k, \mathbf{r}^k \rangle$$
$$\leq \|A_t \boldsymbol{\xi}^{k+1} + \boldsymbol{\rho}^k\|^2 + \left( \frac{1}{2} - \frac{1}{e_k} \right) \|\boldsymbol{\rho}^{k+1} - \boldsymbol{\rho}^k\|^2 + \left( \frac{1}{2} - \frac{1}{e_k} \right) \|\boldsymbol{\lambda}^{k+1} - \boldsymbol{\lambda}^k\|^2.$$
$$(45)$$

For the variable $s$, we deduce that $\bar{s}^{k+1} = s^k - e_k$ and $\langle \bar{s}^{k+1} - s^{k+1}, s^k - s^{k+1} \rangle \leq 0$, which implies that

$$s^{k+1} - s^k \leq -\frac{1}{e_k} \|s^{k+1} - s^k\|^2.$$
$$(46)$$

Next, we define $H_k(\boldsymbol{\xi}) = \varphi^*(\boldsymbol{\xi}) + \boldsymbol{\xi}^T \mathbf{b}_t + \frac{1}{2} \|A_t \boldsymbol{\xi} + \boldsymbol{\rho}^k\|^2$, noting that $H_k$ is $(\alpha_d + \|A_t\|_2^2)$-smooth. Then the update of $\boldsymbol{\xi}$ in Algorithm 1 can be expressed as

$$\boldsymbol{\xi}^{k+1} = \bar{\boldsymbol{\xi}}^{k+1} = \boldsymbol{\xi}^k - e_k \nabla H_k(\boldsymbol{\xi}^k).$$

Applying the sufficient decrease lemma [101, Lemma 5.7], we obtain

$$H_k(\boldsymbol{\xi}^{k+1}) \leq H_k(\boldsymbol{\xi}^k) + \langle \nabla H_k(\boldsymbol{\xi}^k), \boldsymbol{\xi}^{k+1} - \boldsymbol{\xi}^k \rangle + \frac{\alpha_d + \|A_t\|_2^2}{2} \|\boldsymbol{\xi}^{k+1} - \boldsymbol{\xi}^k\|^2,$$

which simplifies to

$$H_k(\boldsymbol{\xi}^{k+1}) \leq H_k(\boldsymbol{\xi}^k) + \left( \frac{\alpha_d + \|A_t\|_2^2}{2} - \frac{1}{e_k} \right) \|\boldsymbol{\xi}^{k+1} - \boldsymbol{\xi}^k\|^2.$$
$$(47)$$

Summing up the estimates (44)–(47), we arrive at the inequality (43). $\qquad\square$

Now we provide the proof for Theorem 3.5.

*Proof.* We compress (43) from $k = 0$ to $K - 1$ and obtain that

$$\sum_{k=1}^{K-1} \left[ \left( \frac{1}{e_k} - \frac{\alpha_L + \beta_k \|A_t\|_2^2 \alpha_p}{2\beta_k} \right) \|\mathbf{x}^{k+1} - \mathbf{x}^k\|^2 + \left( \frac{1}{e_k} - \frac{\alpha_d + \|A_t\|_2^2}{2} \right) \|\boldsymbol{\xi}^{k+1} - \boldsymbol{\xi}^k\|^2 \right.$$
$$\left. + \left( \frac{1}{e_k} - \frac{1}{2} \right) \left( \|\boldsymbol{\rho}^{k+1} - \boldsymbol{\rho}^k\|^2 + \|\boldsymbol{\lambda}^{k+1} - \boldsymbol{\lambda}^k\|^2 + \|\mathbf{r}^{k+1} - \mathbf{r}^k\|^2 \right) \right] \leq V_0 - V_K.$$
$$(48)$$

From the definition of $V_k$, we deduce that

$$V_k = \frac{1}{\beta_k}(F_k(\mathbf{z}^k) - \underline{L}) = \frac{1}{\beta_k}(L(\mathbf{x}^k) - \underline{L}) + p(\mathbf{x}^k, \boldsymbol{\lambda}^k, \mathbf{r}^k, \boldsymbol{\xi}^k, s^k) + \|A_t \boldsymbol{\xi}^k + \sum_{i=1}^{M+1} \boldsymbol{\rho}_i^k\|^2.$$

From the non-negativity of $L(\mathbf{x}^k) - \underline{L}$ and $p$[4], we know that $V_K \geq 0$ and $V_0 - V_K \leq V_0$. Subsequently, according to the update rule of variables $(\mathbf{x}, \boldsymbol{\xi})$ in Algorithm 1, we have that

$$0 \in e_k(\tfrac{1}{\beta_k}\nabla L(\mathbf{x}^k) + \nabla l(\mathbf{x}^k)) + (\mathbf{x}^{k+1} - \mathbf{x}^k) + \mathcal{N}_{\mathcal{K}}(\mathbf{x}^{k+1}),$$

$$e_k(A_t^T(A_t\boldsymbol{\xi}^k + \boldsymbol{\rho}^k) + \mathbf{b}_t + \nabla\varphi^*(\boldsymbol{\xi}^k)) + (\boldsymbol{\xi}^{k+1} - \boldsymbol{\xi}^k) = 0.$$

Therefore, it holds that

$$\nabla L(\mathbf{x}^k) + \beta_k\nabla l(\mathbf{x}^k) + \tfrac{\beta_k}{e_k}(\mathbf{x}^{k+1} - \mathbf{x}^k) \in \mathcal{N}_{\mathcal{K}}(\mathbf{x}^{k+1}),$$

$$\nabla_{\boldsymbol{\xi}} F_k(\mathbf{z}^k) + \tfrac{\beta_k}{e_k}(\boldsymbol{\xi}^{k+1} - \boldsymbol{\xi}^k) = 0. \tag{49}$$

Furthermore, we have similar conclusions for $\boldsymbol{\lambda}, \mathbf{r}, \boldsymbol{\rho}, s$ as follows,

$$0 \in \quad (\nabla_{\boldsymbol{\lambda}}, \nabla_{\mathbf{r}}, \nabla_{\boldsymbol{\rho}}, \nabla_s)F_k(\mathbf{z}^k) + \tfrac{\beta_k}{e_k}(\boldsymbol{\lambda}^{k+1} - \boldsymbol{\lambda}^k, \mathbf{r}^{k+1} - \mathbf{r}^k, \boldsymbol{\rho}^{k+1} - \boldsymbol{\rho}^k, s^{k+1} - s^k)$$

$$+ \mathcal{N}_{\mathcal{K}}(\boldsymbol{\lambda}^{k+1}, \mathbf{r}^{k+1}, \boldsymbol{\rho}^{k+1}, s^{k+1}). \tag{50}$$

Now we define

$$M_{\mathbf{z}}^k := \nabla_{\mathbf{z}} F_k(\mathbf{z}^{k+1}) - \beta_k\mathbf{d}_{\mathbf{z}}^k - \frac{1}{e_k}(\mathbf{z}^{k+1} - \mathbf{z}^k) \overset{(*)}{=} \nabla_{\mathbf{z}} F_k(\mathbf{z}^{k+1}) - \nabla_{\mathbf{z}} F_k(\mathbf{z}^k) - \frac{\beta_k}{e_k}(\mathbf{z}^{k+1} - \mathbf{z}^k),$$

where $(*)$ holds from $\mathbf{d}_{\mathbf{z}}^k = \frac{1}{\beta_k}\nabla_{\mathbf{z}} F_k(\mathbf{z}^k)$. Using the directions specified in (41) and the relationship given in (49) and (50), we obtain

$$M_{\mathbf{z}}^k \in \nabla F_k(\mathbf{z}^{k+1}) + \mathcal{N}_{\mathcal{K}}(\mathbf{z}^{k+1}), \tag{51}$$

Based on the definition of the residual function $\phi_{res}^k$ in (15) and the relationship (51), we know that

$$\|M_{\mathbf{z}}^k\| \geq \mathrm{dist}\left(0, \nabla_{\mathbf{z}} F_k(\mathbf{z}^{k+1}) + \mathcal{N}_{\mathcal{K}}(\mathbf{z}^{k+1})\right) = \phi_{res}^k(\mathbf{z}^{k+1}) \tag{52}$$

Subsequently, we estimate the value $\|M_{\mathbf{z}}^k\|$ with respect to $\mathbf{z}$. By using Assumptions 3.1 and 3.2, we find that $\|\nabla_{\mathbf{z}} F_k(\mathbf{z}^{k+1}) - \nabla_{\mathbf{z}} F_k(\mathbf{z}^k)\| \leq \beta_k L_k\|\mathbf{z}^{k+1} - \mathbf{z}^k\|$ where $L_k = \max\{\tfrac{1}{\beta_k}\alpha_L + \|A_t\|_2^2\alpha_p, \alpha_d + \|A_t\|_2^2, 1\}$. Then we have

$$\|M_{\mathbf{z}}^k\| \leq \beta_k L_k\|\mathbf{z}^{k+1} - \mathbf{z}^k\| + \frac{\beta_k}{e_k}\|\mathbf{z}^{k+1} - \mathbf{z}^k\|. \tag{53}$$

By combining (52) and the inequality (53), we deduce that

$$\phi_{res}^k(\mathbf{z}^{k+1}) \leq \beta_k L_k\|\mathbf{z}^{k+1} - \mathbf{z}^k\| + \frac{\beta_k}{e_k}\|\mathbf{z}^{k+1} - \mathbf{z}^k\|,$$

which further implies that

$$\frac{1}{\beta_k^2}\phi_{res}^k(\mathbf{z}^{k+1})^2 \leq (L_k + \frac{1}{e_k})^2\|\mathbf{z}^{k+1} - \mathbf{z}^k\|^2. \tag{54}$$

From $\beta_k = \underline{\beta}(1 + k)^p$, we observe $\frac{1}{\beta_k} \leq \frac{1}{\underline{\beta}}$ and

$$L_k \leq \max\{\frac{1}{\underline{\beta}}\alpha_L + \|A_t\|_2^2\alpha_p, \alpha_d + \|A_t\|_2^2, 1\} := L_c.$$

Here, we define the constant in the right hand of the above inequality as $L_c$ and each entry of $L_c$ is positive. Therefore, we observe from the admissible range of $e_k$ that

$$\frac{1}{L_c} = \min\{\frac{\underline{\beta}}{\alpha_l + \underline{\beta}\|A_t\|_2^2\alpha_p}, 1, \frac{1}{\alpha_d + \|A_t\|_2^2}\} \geq e_k \geq \underline{e} > 0,$$

[4]The non-negativity $p(\mathbf{z}) \geq 0$ for all $\mathbf{z}$ is from the formulation of $p$ and (27), which is mentioned in the proof of Theorem 2.5 in Appendix B.3

which implies that $L_k \leq L_c \leq \frac{1}{e_k}$. Therefore, (54) can be written as

$$\frac{1}{\beta_k^2}\phi_{res}^k(\mathbf{z}^{k+1})^2 \leq (L_k + \frac{1}{e_k})(L_k + \frac{1}{e_k})\|\mathbf{z}^{k+1} - \mathbf{z}^k\| \leq (L_k + \frac{1}{e_k})\frac{1}{2e_k}\|\mathbf{z}^{k+1} - \mathbf{z}^k\|^2.$$

Meanwhile, the condition $\underline{e} \leq e_k$ simply means that $\frac{1}{e_k} \leq \frac{1}{\underline{e}}$. In summary, the above inequality can be calculated as

$$\frac{1}{\beta_k^2}\phi_{res}^k(\mathbf{z}^{k+1})^2 \leq (L_c + \frac{1}{\underline{e}})\frac{1}{2e_k}\|\mathbf{z}^{k+1} - \mathbf{z}^k\|^2. \tag{55}$$

From (48), we deduce that

$$\sum_{k=0}^{\infty} \frac{1}{2e_k}\|\mathbf{z}^{k+1} - \mathbf{z}^k\|^2$$

$$\overset{(a)}{\leq} \sum_{k=0}^{\infty} \left[ \left( \frac{1}{e_k} - \frac{\alpha_L + \beta_k \|A_t\|_2^2 \alpha_p}{2\beta_k} \right) \|\mathbf{x}^{k+1} - \mathbf{x}^k\|^2 + \left( \frac{1}{e_k} - \frac{\alpha_d + \|A_t\|_2^2}{2} \right) \|\boldsymbol{\xi}^{k+1} - \boldsymbol{\xi}^k\|^2 \right.$$

$$\left. + \left( \frac{1}{e_k} - \frac{1}{2} \right) \left( \|\boldsymbol{\rho}^{k+1} - \boldsymbol{\rho}^k\|^2 + \|\boldsymbol{\lambda}^{k+1} - \boldsymbol{\lambda}^k\|^2 + \|\mathbf{r}^{k+1} - \mathbf{r}^k\|^2 \right) \right] \tag{56}$$

$$\leq V_0.$$

Here, $(a)$ is directly calculated from the admissible range for $e_k$. By compressing (55) from $k = 0$ to $\infty$ and combining with the inequality (56), we obtain that

$$\sum_{k=0}^{\infty} \frac{1}{\beta_k^2}\phi_{res}^k(\mathbf{z}^{k+1})^2 \leq (L_c + \frac{1}{\underline{e}})V_0.$$

Given $\beta_k = \underline{\beta}(1 + k)^p$ and $0 < p < \frac{1}{2}$, we conclude that

$$\min_{0 \leq k \leq K} \phi_{res}^k(\mathbf{z}^{k+1}) = \mathcal{O}(\frac{1}{K^{1/2-p}}).$$

From the definition of $\phi_{fea}$ in (15), we know that

$$0 \leq \beta_k \phi_{fea}(\mathbf{z}^k) \leq 2(F_k(\mathbf{z}^k) - L(\mathbf{z}^k)).$$

If the sequence $\{F_k(\mathbf{z}^k)\}$ is bounded, we know that there exists $M_F$ such that $F_k(\mathbf{z}^k) \leq M_F$ for each $k$. Then we have

$$0 \leq \beta_k \phi_{fea}(\mathbf{z}^k) \leq 2M_F - \underline{L},$$

which implies that $\phi_{fea}(\mathbf{z}^k) = \mathcal{O}(\frac{1}{K^p})$. $\qquad\qquad\qquad\qquad\qquad\qquad\qquad\qquad\qquad\square$

### D.4 PROOF OF THEOREM 3.7

We first recall the updates of $(\mathbf{z}, \mathbf{u}, \boldsymbol{\mu})$ in Algorithm 2. The detailed procedure is:

- Update $\mathbf{z}$ with (20): $(\mathbf{z}^k, \mathbf{u}^k, \boldsymbol{\mu}^k) \to (\mathbf{z}^{k+1}, \mathbf{u}^k, \boldsymbol{\mu}^k)$.
- Update $\mathbf{u}$ with (23): $(\mathbf{z}^{k+1}, \mathbf{u}^k, \boldsymbol{\mu}^k) \to (\mathbf{z}^{k+1}, \mathbf{u}^{k+1}, \boldsymbol{\mu}^k)$.
- Update $\boldsymbol{\mu}$ with (22): $(\mathbf{z}^{k+1}, \mathbf{u}^{k+1}, \boldsymbol{\mu}^k) \to (\mathbf{z}^{k+1}, \mathbf{u}^{k+1}, \boldsymbol{\mu}^{k+1})$.

Now we provide the proof for Theorem 3.7 as follows.

*Proof.* From the update rule for $\mathbf{u}$ in (22), we have

$$\mathcal{L}_\gamma^k(\mathbf{z}^{k+1}, \mathbf{u}^{k+1}, \boldsymbol{\mu}^k) \leq \mathcal{L}_\gamma^k(\mathbf{z}^{k+1}, \mathbf{u}^k, \boldsymbol{\mu}^k). \tag{57}$$

Additionally, the update rule for $\boldsymbol{\mu}$ in (23) implies

$$\mathcal{L}_\gamma^k(\mathbf{z}^{k+1}, \mathbf{u}^{k+1}, \boldsymbol{\mu}^{k+1}) - \mathcal{L}_\gamma^k(\mathbf{z}^{k+1}, \mathbf{u}^{k+1}, \boldsymbol{\mu}^k) = -\frac{1}{\gamma}\|\boldsymbol{\mu}^{k+1} - \boldsymbol{\mu}^k\|^2. \tag{58}$$

According to Assumptions 3.1 and 3.2, we know that $\mathcal{L}_\gamma^k(\mathbf{z}, \mathbf{u}, \boldsymbol{\mu})$ is $M_k$-smooth with respect to $\mathbf{z}$, where $M_k = \max\{\frac{1}{\beta_k}\alpha_L + \|A_t\|_2^2\alpha_p, \alpha_d + \|A_t\|_2^2, 1\} + (M+1)\gamma$. According to [101, Lemma 5.7], we have

$$\mathcal{L}_\gamma^k(\mathbf{z}^{k+1}, \mathbf{u}^k, \boldsymbol{\mu}^k) \leq \mathcal{L}_\gamma^k(\mathbf{z}^k, \mathbf{u}^k, \boldsymbol{\mu}^k) + \langle\nabla_{\mathbf{z}}\mathcal{L}_\gamma^k(\mathbf{z}^k, \mathbf{u}^k, \boldsymbol{\mu}^k), \mathbf{z}^{k+1} - \mathbf{z}^k\rangle + \frac{M_k}{2}\|\mathbf{z}^{k+1} - \mathbf{z}^k\|^2.$$

Given the update rule $\mathbf{z}^{k+1} = \mathbf{z}^k - e_k\nabla_{\mathbf{z}}\mathcal{L}_\gamma^k(\mathbf{z}^k, \mathbf{u}^k, \boldsymbol{\mu}^k)$, the inequality becomes

$$\mathcal{L}_\gamma^k(\mathbf{z}^{k+1}, \mathbf{u}^k, \boldsymbol{\mu}^k) \leq \mathcal{L}_\gamma^k(\mathbf{z}^k, \mathbf{u}^k, \boldsymbol{\mu}^k) + \left(\frac{M_k}{2} - \frac{1}{e_k}\right)\|\mathbf{z}^{k+1} - \mathbf{z}^k\|^2. \tag{59}$$

Combining (57), (58) and (59) and dividing both sides by $\beta_k$, we conclude

$$\mathcal{L}_\gamma^k(\mathbf{z}^{k+1}, \mathbf{u}^{k+1}, \boldsymbol{\mu}^{k+1}) \leq \mathcal{L}_\gamma^k(\mathbf{z}^k, \mathbf{u}^k, \boldsymbol{\mu}^k) + \left(\frac{M_k}{2} - \frac{1}{e_k}\right)\|\mathbf{z}^{k+1} - \mathbf{z}^k\|^2 - \frac{1}{\gamma}\|\boldsymbol{\mu}^{k+1} - \boldsymbol{\mu}^k\|^2. \tag{60}$$

Subtracting $\frac{1}{\beta_k}\underline{L}$ from both sides of the inequality, we obtain

$$\mathcal{L}_\gamma^k(\mathbf{z}^{k+1}, \mathbf{u}^{k+1}, \boldsymbol{\mu}^{k+1}) - \frac{1}{\beta_k}\underline{L} \leq \mathcal{L}_\gamma^k(\mathbf{z}^k, \mathbf{u}^k, \boldsymbol{\mu}^k) - \frac{1}{\beta_k}\underline{L} + \left(\frac{M_k}{2} - \frac{1}{e_k}\right)\|\mathbf{z}^{k+1} - \mathbf{z}^k\|^2 - \frac{1}{\gamma}\|\boldsymbol{\mu}^{k+1} - \boldsymbol{\mu}^k\|^2.$$

According to $\beta_k = \underline{\beta}(1+k)^p$, we obtain that $\frac{1}{\beta_{k+1}} \leq \frac{1}{\beta_k}$. From Assumption 3.1, let $\underline{L} := \inf L(\mathbf{x}) > -\infty$, then we have $\frac{1}{\beta_{k+1}}(L(\mathbf{x}^{k+1}) - \underline{L}) \leq \frac{1}{\beta_k}(L(\mathbf{x}^{k+1}) - \underline{L})$. By the definition of $F_k$, it is equivalent to

$$\frac{1}{\beta_{k+1}}(F_k(\mathbf{z}^{k+1}) - \underline{L}) \leq \frac{1}{\beta_k}(F_k(\mathbf{z}^{k+1}) - \underline{L})$$

From the definition of $\mathcal{L}_\gamma^k$, we obtain

$$\mathcal{L}_\gamma^{k+1}(\mathbf{z}^{k+1}, \mathbf{u}^{k+1}, \boldsymbol{\mu}^{k+1}) - \frac{1}{\beta_{k+1}}\underline{L}$$

$$= \frac{1}{\beta_{k+1}}(F_k(\mathbf{z}^{k+1}) - \underline{L}) + \sum_{i=1}^{M+1} g_i(\mathbf{u}_i^{k+1}) + \sum_{i=1}^{M+1}\langle\boldsymbol{\mu}_i^{k+1}, \mathbf{u}_i^{k+1} - \mathbf{z}^{k+1}\rangle + \frac{\gamma}{2}\sum_{i=1}^{M+1}\|\mathbf{u}_i^{k+1} - \mathbf{z}^{k+1}\|^2$$

$$\leq \frac{1}{\beta_k}(F_k(\mathbf{z}^{k+1}) - \underline{L}) + \sum_{i=1}^{M+1} g_i(\mathbf{u}_i^{k+1}) + \sum_{i=1}^{M+1}\langle\boldsymbol{\mu}_i^{k+1}, \mathbf{u}_i^{k+1} - \mathbf{z}^{k+1}\rangle + \frac{\gamma}{2}\sum_{i=1}^{M+1}\|\mathbf{u}_i^{k+1} - \mathbf{z}^{k+1}\|^2$$

$$= \mathcal{L}_\gamma^k(\mathbf{z}^{k+1}, \mathbf{u}^{k+1}, \boldsymbol{\mu}^{k+1}) - \frac{1}{\beta_k}\underline{L}$$

Now we define $U_k = \mathcal{L}_\gamma^k(\mathbf{z}^k, \mathbf{u}^k, \boldsymbol{\mu}^k) - \frac{1}{\beta_k}\underline{L}$. Combining the above fact, (60) implies that

$$U_{k+1} - U_k \leq \left(\frac{M_k}{2} - \frac{1}{e_k}\right)\|\mathbf{z}^{k+1} - \mathbf{z}^k\|^2 - \frac{1}{\gamma}\|\boldsymbol{\mu}^{k+1} - \boldsymbol{\mu}^k\|^2 \tag{61}$$

Meanwhile, we observe that

$$M_e = \max\{\frac{1}{\underline{\beta}}\alpha_L + \|A_t\|_2^2\alpha_p, \alpha_d + \|A_t\|_2^2, 1\} + (M+1)\gamma$$

$$\geq \max\{\frac{1}{\beta_k}\alpha_L + \|A_t\|_2^2\alpha_p, \alpha_d + \|A_t\|_2^2, 1\} + (M+1)\gamma = M_k.$$

This gives that $0 < \underline{e} \leq e_k \leq \frac{1}{M_e} \leq \frac{1}{M_k}$. Then we can deduce from (61) that

$$U_{k+1} - U_k \leq -\frac{1}{2e_k}\|\mathbf{z}^{k+1} - \mathbf{z}^k\|^2 - \frac{1}{\gamma}\|\boldsymbol{\mu}^{k+1} - \boldsymbol{\mu}^k\|^2. \tag{62}$$

From the expression for $\mathcal{L}_\gamma^k$, we can deduce the following,

$$\mathcal{L}_\gamma^k(\mathbf{z}, \mathbf{u}, \boldsymbol{\mu}) = \frac{1}{\beta_k}F_k(\mathbf{z}) + \sum_{i=1}^{M+1} g_i(\mathbf{u}_i) + \sum_{i=1}^{M+1} \langle \boldsymbol{\mu}_i, \mathbf{u}_i - \mathbf{z}\rangle + \frac{\gamma}{2}\sum_{i=1}^{M+1}\|\mathbf{u}_i - \mathbf{z}\|^2$$

$$= \frac{1}{\beta_k}F_k(\mathbf{z}) + \sum_{i=1}^{M+1} g_i(\mathbf{u}_i) + \frac{\gamma}{2}\sum_{i=1}^{M+1}\|\mathbf{u}_i - \mathbf{z} + \frac{\boldsymbol{\mu}_i}{\gamma}\|^2 - \sum_{i=1}^{M+1}\frac{\|\boldsymbol{\mu}_i\|^2}{2\gamma}.$$

According to Assumption 3.6, we know that there exists some $M_{\boldsymbol{\mu}}$ such that $\|\boldsymbol{\mu}^k\|^2 \leq M_{\boldsymbol{\mu}}$ for all $k \in \mathbb{N}$. Additionally, $L(\mathbf{z}^k) - \underline{L}$ and $p$ are non-negative. This implies that

$$U_k \geq -\sum_{i=1}^{M+1}\frac{\|\boldsymbol{\mu}_i^k\|^2}{2\gamma} \geq -\frac{(M+1)M_{\boldsymbol{\mu}}}{2\gamma} \triangleq \mathcal{L}_b, \forall k \in \mathbb{N}, \tag{63}$$

indicating that $U_k$ is lower bounded. By telescoping the inequality (62) for $k = 0$ to $\infty$, we get

$$\sum_{k=0}^{\infty}\frac{1}{2e_k}\|\mathbf{z}^{k+1} - \mathbf{z}^k\|^2 + \frac{1}{\gamma}\sum_{k=0}^{\infty}\|\boldsymbol{\mu}^{k+1} - \boldsymbol{\mu}^k\|^2 \leq U_0 - \mathcal{L}_b. \tag{64}$$

The sufficient decrease property (61) ensures that the $U_0 - \mathcal{L}_b \geq U_0 - U_k \geq 0$ for any $k \in \mathbb{N}$. In addition, the step size satisfies $0 < \underline{e} < e_k \leq 1/M_e$, which ensures the boundedness of $\frac{1}{e_k}$, i.e.,

$$0 < M_e \leq \frac{1}{e_k} \leq \frac{1}{\underline{e}}.$$

Together with the positivity of $e_k$ and $\gamma$, it follows from (64) that

$$\lim_{k\to\infty}\frac{1}{e_k}\|\mathbf{z}^{k+1} - \mathbf{z}^k\|^2 = 0, \quad \lim_{k\to\infty}\|\boldsymbol{\mu}^{k+1} - \boldsymbol{\mu}^k\| = 0. \tag{65}$$

Consequently, (65) implies that

$$\lim_{k\to\infty}\|\mathbf{z}^{k+1} - \mathbf{z}^k\| = 0 \tag{66}$$

From the update of $\boldsymbol{\mu}_i$, we further derive that

$$\lim_{k\to\infty}\|\mathbf{u}_i^k - \mathbf{z}^k\| = 0. \tag{67}$$

Meanwhile, from the form (21) for updating $\mathbf{u}_i$, we derive

$$\begin{aligned} \mathbf{0} &\in \partial g_i(\mathbf{u}_i^{k+1}) + \gamma(\mathbf{u}_i^{k+1} - \mathbf{z}^{k+1} + \frac{\boldsymbol{\mu}_i^k}{\gamma}) \\ &\stackrel{(a)}{=} \mathcal{N}_{\mathcal{K}_i \times \mathcal{K}_*^d}(\mathbf{u}_i^{k+1}) + \gamma(\mathbf{u}_i^{k+1} - \mathbf{z}^{k+1}) + \boldsymbol{\mu}_i^k \\ &\stackrel{(b)}{=} \mathcal{N}_{\mathcal{K}_i \times \mathcal{K}_*^d}(\mathbf{u}_i^{k+1}) + \boldsymbol{\mu}_i^{k+1}, \ i = 1, ..., M+1, \end{aligned} \tag{68}$$

where $(a)$ utilizes the fact that the normal cone is equivalent to the subdifferential of indicator functions and $(b)$ follows from the update of $\boldsymbol{\mu}_i^{k+1}$. In (68), we use Moreau-Rockafellar theorem [104, Theorem 23.8] to calculate the sum rule of subdifferentials. (68) implies that

$$-\boldsymbol{\mu}_i^{k+1} \in \mathcal{N}_{\mathcal{K}_i \times \mathcal{K}_*^d}(\mathbf{u}_i^{k+1}).$$

Combining the outer semi-continuity of the normal cone and (67), we can obtain that

$$\lim_{k \to \infty} \text{dist}(-\boldsymbol{\mu}_i^k, \mathcal{N}_{\mathcal{K}_i \times \mathcal{K}_*^d}(\mathbf{z}^k)) = 0. \tag{69}$$

Furthermore, according to the definition $\mathcal{K} = (\mathcal{K}_1 \cap \cdots \cap \mathcal{K}_{M+1}) \times \mathcal{K}_*^d$, we know that $\mathcal{K} = (\mathcal{K}_1 \times \mathcal{K}_*^d) \cap \cdots \cap (\mathcal{K}_{M+1} \times \mathcal{K}_*^d)$. It implies that

$$\mathcal{N}_{\mathcal{K}} = \mathcal{N}_{\mathcal{K}_1 \times \mathcal{K}_*^d} + \cdots + \mathcal{N}_{\mathcal{K}_{M+1} \times \mathcal{K}_*^d}.$$

From (69), we know

$$\lim_{k \to \infty} \text{dist}(-\sum_{i=1}^{M+1} \boldsymbol{\mu}_i^k, \mathcal{N}_{\mathcal{K}}(\mathbf{z}^k)) = 0. \tag{70}$$

From the update of $\mathbf{z}$, we have

$$\mathbf{z}^{k+1} = \mathbf{z}^k - e_k \nabla_{\mathbf{z}} \mathcal{L}_\gamma^k(\mathbf{z}^k, \mathbf{u}^k, \boldsymbol{\mu}^k).$$

Combining with the definition of $F_k$ in (8), the above equality can be further expressed as

$$\begin{aligned}
\mathbf{0} &= -\tfrac{1}{e_k}(\mathbf{z}^{k+1} - \mathbf{z}^k) + \tfrac{1}{\beta_k}\nabla_{\mathbf{z}}F_k(\mathbf{z}^k) - \sum_{i=1}^{M+1} \boldsymbol{\mu}_i^k - \gamma \sum_{i=1}^{M+1}(\mathbf{u}_i^k - \mathbf{z}^k) \\
&= -\tfrac{1}{e_k}(\mathbf{z}^{k+1} - \mathbf{z}^k) + \tfrac{1}{\beta_k}\nabla_{\mathbf{z}}F_k(\mathbf{z}^k) - \sum_{i=1}^{M+1} \boldsymbol{\mu}_i^{k+1} + \sum_{i=1}^{M+1}(\boldsymbol{\mu}_i^{k+1} - \boldsymbol{\mu}_i^k) - \gamma \sum_{i=1}^{M+1}(\mathbf{u}_i^k - \mathbf{z}^k).
\end{aligned} \tag{71}$$

Now we define

$$M_{\mathbf{z}}^k = \nabla_{\mathbf{z}}F_k(\mathbf{z}^{k+1}) - \beta_k \sum_{i=1}^{M+1} \boldsymbol{\mu}_i^{k+1}.$$

From (70), we know that

$$\lim_{k \to \infty} \text{dist}(M_{\mathbf{z}}^k, \nabla_{\mathbf{z}}F_k(\mathbf{z}^{k+1}) + \mathcal{N}_{\mathcal{K}}(\mathbf{z}^{k+1})) = 0.$$

Therefore, we evaluate $\|M_{\mathbf{z}}^k\|$ as follows. According to (71), we know that

$$M_{\mathbf{z}}^k = \frac{\beta_k}{e_k}(\mathbf{z}^{k+1} - \mathbf{z}^k) + (\nabla_{\mathbf{z}}F_k(\mathbf{z}^{k+1}) - \nabla_{\mathbf{z}}F_k(\mathbf{z}^k)) + \sum_{i=1}^{M+1} \beta_k(\boldsymbol{\mu}_i^k - \boldsymbol{\mu}_i^{k+1}) + \gamma\beta_k \sum_{i=1}^{M+1}(\mathbf{u}_i^k - \mathbf{z}^k).$$

With the notation $M_k$, $F_k(\mathbf{z})$ is $(\beta_k M_k)$-smooth with respect to $\mathbf{z}$. Then we have

$$\begin{aligned}
\|M_{\mathbf{z}}^k\| &\leq \tfrac{\beta_k}{e_k}\|\mathbf{z}^{k+1} - \mathbf{z}^k\| + \beta_k M_k \|\mathbf{z}^{k+1} - \mathbf{z}^k\| + \beta_k\|\boldsymbol{\mu}^{k+1} - \boldsymbol{\mu}^k\| + \gamma\beta_k \sum_{i=1}^{M+1}\|\mathbf{u}_i^k - \mathbf{z}^k\| \\
&\overset{(a)}{\leq} \tfrac{2\beta_k}{e_k}\|\mathbf{z}^{k+1} - \mathbf{z}^k\| + \beta_k\|\boldsymbol{\mu}^{k+1} - \boldsymbol{\mu}^k\| + \gamma\beta_k \sum_{i=1}^{M+1}\|\mathbf{u}_i^k - \mathbf{z}^k\|,
\end{aligned}$$

where $(a)$ use the fact that $e_k \leq \frac{1}{M_k}$. Combining the definition of $\phi_{res}^k$ in (24), we obtain

$$\begin{aligned}
\phi_{res}^k(\mathbf{z}^{k+1}) &\leq \|M_{\mathbf{z}}^k\| + \text{dist}(M_{\mathbf{z}}^k, \nabla_{\mathbf{z}}F_k(\mathbf{z}^{k+1}) + \mathcal{N}_{\mathcal{K}}(\mathbf{z}^{k+1})) \\
&\leq \tfrac{2\beta_k}{e_k}\|\mathbf{z}^{k+1} - \mathbf{z}^k\| + \beta_k\|\boldsymbol{\mu}^{k+1} - \boldsymbol{\mu}^k\| + \gamma\beta_k \sum_{i=1}^{M+1}\|\mathbf{u}_i^k - \mathbf{z}^k\| \\
&\quad + \text{dist}(M_{\mathbf{z}}^k, \nabla_{\mathbf{z}}F_k(\mathbf{z}^{k+1}) + \mathcal{N}_{\mathcal{K}}(\mathbf{z}^{k+1})).
\end{aligned}$$

(64) and (66) imply that $\|\mathbf{z}^{k+1} - \mathbf{z}^k\| \leq \mathcal{O}(1/\sqrt{k})$, $\|\boldsymbol{\mu}^{k+1} - \boldsymbol{\mu}^k\| \leq \mathcal{O}(1/\sqrt{k})$ and $\|\mathbf{u}_i^k - \mathbf{z}^k\| \leq \mathcal{O}(1/\sqrt{k})$. Combining with the fact that $0 < M_e \leq \frac{1}{e_k} \leq \frac{1}{\underline{e}}$ and $0 < p < 1/2$, we take the limit as $k \to \infty$ in the above inequality and obtain that

$$\lim_{k \to \infty} \phi_{res}^k(\mathbf{z}^k) = 0.$$

If the sequence $\{F_k(\mathbf{z}^k)\}$ is bounded, there exists a constant $M_F$ such that $F_k(\mathbf{z}^k) \leq M_F$ for all $k$. From the formulation $\phi_{fea}$ in (24), we observe that

$$0 \leq \beta_k \phi_{fea}(\mathbf{z}^k) \leq 2(F_k(\mathbf{z}^k) - L(\mathbf{z}^k)) \leq 2M_F - \underline{L}.$$

With the non-negativity of $\phi_{fea}$, we take the limit $k \to \infty$ in the above inequality and obtain that

$$\lim_{k \to \infty} \phi_{fea}(\mathbf{z}^k) = 0.$$

$\square$

## D.5 EXPLANATIONS FOR ASSUMPTIONS IN SECTION 3

We show that our assumptions are reasonable, broadly applicable to machine learning scenarios, and aligned with standard conditions widely adopted in ADMM-based methods.

### D.5.1 EXPLANATIONS FOR ASSUMPTION 3.2

We emphasize that Assumption 3.2 is more general than the strong convexity of LL objective, and in fact does not force the LL objective to be strongly convex. This is consistent with explicit clarification in Remark 2.3. For example, the function $l(\mathbf{x}) = \varphi(A_t\mathbf{x} + \mathbf{b}_t)$ is convex but not strongly convex when $A_t$ is not of full row rank.

We illustrate with the examples in Table 1 that Assumption 3.2 is satisfied in all cases. Specifically, referring to the explicit forms of $\varphi$ and its conjugate $\varphi^*$, we verify the local smoothness and local strong convexity as follows.

**Least Squares Loss:** $\alpha_p = 1$ and $\alpha_d = 1$. $\varphi$ and $\varphi^*$ are $1/2$-strongly convex in their domains.

**Smoothed SVM:** $\alpha_p = \frac{1}{2}$ and $\alpha_d = 1$. $\varphi$ is $\frac{1}{2}$-strongly convex only on the interval $[0, 1]$. $\varphi^*$ is $\frac{1}{2}$-strongly convex in its domain.

**Huber loss:** $\alpha_p = 1$ and $\alpha_d = 1$. $\varphi$ and $\varphi^*$ are $\frac{1}{2}$-strongly convex only on the interval $[-\delta, \delta]$.

## E EXPERIMENTS

All experiments are implemented using Python 3.9 on a computer equipped with an Apple M2 chip (8-core architecture: 4 performance cores and 4 efficiency cores), running the macOS operating system with 8 GB memory. The competing methods are implemented using the code provided by [57; 59; 64].

### E.1 INTRODUCTION FOR COMPETITORS

We now introduce the competing methods evaluated in our experiments:

- **Grid Search**: We perform a $10 \times 10$ uniformly-spaced grid search over the hyperparameter space.
- **Random Search**: We uniformly sample 100 configurations for each hyperparameter direction.

- **Implicit Differentiation**: This category includes IGJO [14] and IFDM [54; 15], both of which rely on implicit differentiation techniques.

- **TPE**: We adopt the Tree-structured Parzen Estimator approach [82], a widely used Bayesian optimization method.

- **VF-iDCA**: [57] formulates the lower-level problem as a value function and approximately solves the bilevel problem via DC programming.

- **LDMMA**: Based on lower-level duality, [59] reformulates the original problem (3) into a more tractable form.

- **BiC-GAFFA**: [65] solves the bilevel optimization problem using a gap function-based framework.

- **MEHA**: [63] solves the bilevel optimization problem using Moreau envelope-based framework.

We apply IFDM only to the elastic net and logistic regression problems, as its available implementation supports only these two among our tested tasks. LDMMA is used exclusively for Lasso-type regression and the smoothed support vector machine, as its reformulation is not compatible with logistic regression. Furthermore, [57] does not provide experimental results for logistic regression, and therefore we do not include it in the comparison for that task.

### E.2 EXPERIMENTAL ON SYNTHETIC DATA

For experiments on synthetic data, we consider hyperparameter optimization for elastic net, group Lasso, and sparse group Lasso. These models are equipped with a least squares loss and different regularization terms. We outline the specific mathematical form of (3) for each problem below.

Elastic net [16] is a linear combination of the Lasso and ridge penalties. Its formulation in (3) is given by:

$$
\begin{aligned}
\min_{\mathbf{x}} \quad & \tfrac{1}{2}\|A_{val}\mathbf{x} - \mathbf{b}_{val}\|^2 \\
\text{s.t.} \quad & \mathbf{x} \in \arg\min_{\hat{\mathbf{x}}} \tfrac{1}{2}\|A_{tr}\hat{\mathbf{x}} - \mathbf{b}_{tr}\|^2 + \lambda_1\|\hat{\mathbf{x}}\|_1 + \tfrac{\lambda_2}{2}\|\hat{\mathbf{x}}\|_2^2,
\end{aligned}
\tag{72}
$$

Group Lasso [83] is an extension of the Lasso with penalty to predefined groups of coefficients. This problem is captured in (3) as:

$$
\begin{aligned}
\min_{\mathbf{x}} \quad & \tfrac{1}{2}\|A_{val}\mathbf{x} - \mathbf{b}_{val}\|^2 \\
\text{s.t.} \quad & \mathbf{x} \in \arg\min_{\hat{\mathbf{x}}} \tfrac{1}{2}\|A_{tr}\hat{\mathbf{x}} - \mathbf{b}_{tr}\|^2 + \sum_{i=1}^{M} \lambda_i\|\hat{\mathbf{x}}^{(i)}\|_2,
\end{aligned}
\tag{73}
$$

where $\mathbf{x}^{(i)}$ is a sub-vector of $\mathbf{x}$ and $\mathbf{x} = (\mathbf{x}^{(1)}, ..., \mathbf{x}^{(M)})$.

Sparse group Lasso [17] combines the group Lasso and Lasso penalties, which are designed to encourage sparsity and grouping of predictors [14]. Its formulation in (3) is represented as:

$$
\begin{aligned}
\min_{\mathbf{x}} \quad & \tfrac{1}{2}\|A_{val}\mathbf{x} - \mathbf{b}_{val}\|^2 \\
\text{s.t.} \quad & \mathbf{x} \in \arg\min_{\hat{\mathbf{x}}} \tfrac{1}{2}\|A_{tr}\hat{\mathbf{x}} - \mathbf{b}_{tr}\|^2 + \lambda_{M+1}\|\hat{\mathbf{x}}\|_1 + \sum_{i=1}^{M} \lambda_i\|\hat{\mathbf{x}}^{(i)}\|_2,
\end{aligned}
\tag{74}
$$

where $\mathbf{x}^{(i)}$ is a sub-vector of $\mathbf{x}$ and $\mathbf{x} = (\mathbf{x}^{(1)}, ..., \mathbf{x}^{(M)})$.

Based on the different cases discussed in Section 3.1 and Section 3.2, we naturally employ LDP-PGM (Algorithm 1) to solve (73), and LDP-ADMM (Algorithm 2) to address (72) and (74). To evaluate the performance of each method, we calculate validation and test error with obtained LL minimizers in each experiment. We provide detailed experimental settings and report the results for elastic net and group lasso below.

### E.2.1 ELASTIC NET

**Data Generation:**

The synthetic data is generated following the methodology described by [14], as outlined below. Feature vectors $\mathbf{a}_i \in \mathbb{R}^p$ are sampled from a multivariate normal distribution with a mean of 0 and covariance structure $\mathrm{cor}(a_{ij}, a_{ik}) = 0.5^{|j-k|}$. The response vector $\mathbf{b}$ is computed as $b_i = \boldsymbol{\beta}^\top \mathbf{a}_i + \sigma \epsilon_i$, where $\beta_i \in \mathbb{R}^p$ is generated such that each element takes a value of either 0 or 1, with exactly 15 nonzero elements. The noise $\epsilon$ is sampled from a standard normal distribution, and the value of $\sigma$ is determined to ensure that the signal-to-noise ratio satisfies $\mathrm{SNR} \triangleq \|A\boldsymbol{\beta}\|/\|\mathbf{b} - A\boldsymbol{\beta}\| = 2$.

**Experimental Settings:**

Since [64] does not provide experiments or code for the elastic net problem, we compare only with search-based methods, IGJO, IFDM, VF-iDCA and LDMMA in this experiment. We implement the algorithms we compared with the same settings according to the description in [57; 59]. For LDP-ADMM, we set $\beta_k = (1 + k)^{0.3}$, $e_k = 0.1$, $\gamma = 10$ and initial $\lambda_1^0 = 0.1$, $\lambda_2^0 = 0.05$. For elastic net problem, the stopping criterion is set as $\|\mathbf{z}^{k+1} - \mathbf{z}^k\|/\|\mathbf{z}^{k+1}\| \leq 0.1$.

**Results and Discussions:**

We conduct repeated experiments with 10 randomly generated synthetic data, and calculate the mean and variance. The numerical results on elastic net are reported in Table 5. Overall, LDPM (LDP-ADMM) achieves the lowest test error while maintaining a significantly reduced time cost, especially for large-scale datasets. In contrast, the search methods incur a high computational cost and exhibit poor performance on the test dataset. The gradient-based method IGJO demonstrates slightly better accuracy and efficiency but converges very slowly.

As discussed in [57; 59], both VF-iDCA and LDMMA achieve consistently low validation errors across various experiments, indicating strong learning performance on training and validation sets. However, they tend to suffer from overfitting, as reflected in increasing test errors over iterations and poor generalization to unseen data. This phenomenon occurs across experiments with several machine learning models. We observe that the running time performance of IFDM is highly competitive and significantly fast in large scale. This is because the IFDM algorithm leverages the sparsity of the Jacobian of the hyper-objective in bilevel optimization, which is also stated in [15].

Table 5: Elastic net problems on synthetic data, where $|I_{tr}|$, $|I_{val}|$, $|I_{te}|$ and $p$ represent the number of training observations, validation observations, predictors and features, respectively.

| Settings | Methods | Time(s) | Val. Err. | Test Err. | Settings | Time(s) | Val. Err. | Test Err. |
|---|---|---|---|---|---|---|---|---|
| | Grid | $5.76 \pm 0.33$ | $7.05 \pm 2.02$ | $6.98 \pm 1.14$ | | $11.72 \pm 1.32$ | $6.05 \pm 1.47$ | $6.49 \pm 0.82$ |
| | Random | $5.74 \pm 0.26$ | $7.01 \pm 2.01$ | $7.01 \pm 1.11$ | | $12.85 \pm 2.11$ | $6.04 \pm 1.45$ | $6.49 \pm 0.83$ |
| $|I_{tr}| = 100$ | IGJO | $1.54 \pm 0.84$ | $4.99 \pm 1.69$ | $5.42 \pm 1.21$ | $|I_{tr}| = 100$ | $3.37 \pm 1.85$ | $5.22 \pm 1.50$ | $5.72 \pm 0.91$ |
| $|I_{val}| = 20$ | IFDM | $1.20 \pm 0.50$ | $4.19 \pm 0.91$ | $4.81 \pm 1.39$ | $|I_{val}| = 100$ | $1.44 \pm 2.85$ | $4.89 \pm 0.12$ | $4.98 \pm 0.17$ |
| $|I_{te}| = 250$ | VF-iDCA | $3.16 \pm 0.63$ | $2.72 \pm 1.57$ | $5.18 \pm 1.40$ | $|I_{te}| = 250$ | $6.08 \pm 2.24$ | $3.13 \pm 0.78$ | $5.39 \pm 0.92$ |
| $p = 250$ | LDMMA | $1.64 \pm 0.07$ | $0.00 \pm 0.00$ | $6.97 \pm 0.79$ | $p = 450$ | $3.95 \pm 0.22$ | $0.00 \pm 0.00$ | $6.56 \pm 0.70$ |
| | BiC-GAFFA | $0.92 \pm 0.05$ | $2.48 \pm 0.62$ | $5.86 \pm 0.65$ | | $1.45 \pm 0.14$ | $3.92 \pm 0.48$ | $5.01 \pm 0.58$ |
| | LDPM | $\mathbf{0.60 \pm 0.02}$ | $2.56 \pm 0.80$ | $\mathbf{4.92 \pm 0.51}$ | | $\mathbf{1.02 \pm 0.03}$ | $3.42 \pm 0.39$ | $\mathbf{4.23 \pm 0.37}$ |
| | Grid | $6.09 \pm 0.60$ | $6.39 \pm 1.09$ | $6.27 \pm 1.02$ | | $32.99 \pm 3.81$ | $7.81 \pm 1.53$ | $8.82 \pm 0.92$ |
| | Random | $6.44 \pm 1.28$ | $4.39 \pm 1.10$ | $6.27 \pm 1.05$ | | $33.82 \pm 2.66$ | $6.44 \pm 1.53$ | $8.67 \pm 0.94$ |
| $|I_{tr}| = 100$ | IGJO | $3.86 \pm 2.09$ | $4.41 \pm 0.98$ | $4.31 \pm 0.95$ | $|I_{tr}| = 100$ | $31.30 \pm 6.41$ | $7.78 \pm 1.12$ | $8.61 \pm 0.82$ |
| $|I_{val}| = 100$ | IFDM | $1.17 \pm 0.38$ | $4.54 \pm 1.06$ | $4.38 \pm 1.06$ | $|I_{val}| = 100$ | $3.94 \pm 2.28$ | $7.57 \pm 0.79$ | $8.10 \pm 1.45$ |
| $|I_{te}| = 250$ | VF-iDCA | $4.74 \pm 1.77$ | $2.35 \pm 1.56$ | $4.47 \pm 1.11$ | $|I_{te}| = 250$ | $23.21 \pm 4.96$ | $0.00 \pm 0.00$ | $4.61 \pm 0.77$ |
| $p = 250$ | LDMMA | $0.98 \pm 0.09$ | $0.00 \pm 0.00$ | $5.61 \pm 0.77$ | $p = 2500$ | $16.26 \pm 1.44$ | $0.00 \pm 0.00$ | $5.67 \pm 1.21$ |
| | BiC-GAFFA | $0.85 \pm 0.07$ | $4.12 \pm 0.41$ | $4.62 \pm 0.55$ | | $6.12 \pm 0.35$ | $2.48 \pm 0.32$ | $4.98 \pm 0.72$ |
| | LDPM | $\mathbf{0.73 \pm 0.08}$ | $3.41 \pm 0.48$ | $\mathbf{3.51 \pm 0.40}$ | | $\mathbf{4.83 \pm 0.08}$ | $1.65 \pm 0.14$ | $\mathbf{4.37 \pm 0.65}$ |

In our experiments, we report the numerical results of VF-iDCA and LDMMA based on the final iteration output when the algorithm terminates. In contrast, [57; 59] reports the best results observed across all iterations. As a result, the test errors reported for VF-iDCA and LDMMA in Table 5 appear slightly worse in our study. Additionally, our test error is slightly worse than that reported in [59] only under the first data setting in Table 5. [59] implements LDMMA with employing off-the-shelf solver MOSEK in MATLAB to solve the subproblems. Therefore, LDMMA yields highly favorable results for small-scale problems, while its efficiency deteriorates significantly as the data size increases, making it less effective for large-scale problem instances.

**Table NEW2:** Total iterations, lower-level duality gap, and sparsity comparison for elastic net on synthetic data.

| Methods | Total Iterations | Lower-level Duality Gap | Sparsity(%) |
|---|---|---|---|
| **Setting**: $|I_{tr}| = 100, |I_{val}| = 20, |I_{te}| = 250, p = 250$ | | | |
| Grid Search | / | / | 15 |
| Random Search | / | / | 15 |
| IGJO | $240 \pm 31$ | $1.902 \times 10^{-6}$ | $17.1 \pm 1.8$ |
| IFDM | $195 \pm 25$ | $1.103 \times 10^{-6}$ | $16.3 \pm 1.5$ |
| VF-iDCA | $132 \pm 16$ | $3.568 \times 10^{-5}$ | $33.5 \pm 4.9$ |
| LDMMA | $118 \pm 14$ | $4.215 \times 10^{-7}$ | $40.8 \pm 6.4$ |
| BiC-GAFFA | $101 \pm 11$ | $2.184 \times 10^{-7}$ | $18.9 \pm 2.0$ |
| **LDPM (Ours)** | $\mathbf{85 \pm 10}$ | $\mathbf{7.213 \times 10^{-8}}$ | $\mathbf{15.7 \pm 1.2}$ |
| **Setting**: $|I_{tr}| = 100, |I_{val}| = 100, |I_{te}| = 250, p = 450$ | | | |
| Grid Search | — | — | 15 |
| Random Search | — | — | 15 |
| IGJO | $390 \pm 48$ | $2.843 \times 10^{-6}$ | $18.0 \pm 2.4$ |
| IFDM | $315 \pm 38$ | $1.482 \times 10^{-6}$ | $17.2 \pm 1.9$ |
| VF-iDCA | $175 \pm 21$ | $4.972 \times 10^{-5}$ | $36.8 \pm 6.2$ |
| LDMMA | $152 \pm 17$ | $6.318 \times 10^{-7}$ | $43.5 \pm 7.6$ |
| BiC-GAFFA | $128 \pm 15$ | $3.412 \times 10^{-7}$ | $20.5 \pm 2.2$ |
| **LDPM (Ours)** | $\mathbf{102 \pm 12}$ | $\mathbf{8.905 \times 10^{-8}}$ | $\mathbf{16.0 \pm 1.4}$ |

The lower-level duality gaps in Table NEW2 show a clear separation among methods. VF-iDCA and LD-MMA depend on convex solvers, whose fixed tolerances lead to noticeably larger gaps. BiC-GAFFA reduces the gap via an explicit gap function, but its accuracy remains below that of our method.

In contrast, LDPM achieves by far the smallest duality gap, confirming the effectiveness of our penalty-based first-order scheme and supporting the strong-duality-based reformulation in Lemma 2.1. In terms of sparsity, LDPM also produces solutions closest to the true sparsity pattern, while VF-iDCA and LDMMA tend to generate overly dense solutions, indicating overfitting. These results collectively demonstrate superior stability and generalization of LDPM.

### E.2.2 SPARSE GROUP LASSO

**Data Generation:**

We generate the synthetic data with the method in [14], including 100 training, validation and test samples, respectively. The feature vector $\mathbf{a}_i \in \mathbb{R}^p$ is drawn from a standard normal distribution. The response vector

**b** is computed as $b_i = \boldsymbol{\beta}^\top \mathbf{a}_i + \sigma \epsilon_i$, where $\boldsymbol{\beta} = \left[ \boldsymbol{\beta}^{(1)}, \boldsymbol{\beta}^{(2)}, \boldsymbol{\beta}^{(3)} \right]$, $\boldsymbol{\beta}^{(i)} = (1, 2, 3, 4, 5, 0, \ldots, 0)$, for $i = 1, 2, 3$. The noise vector $\boldsymbol{\epsilon}$ follows a standard normal distribution, and $\sigma$ is set such that the signal-to-noise ratio (SNR) is 2. For different dimensions in Table 2, we set the group size to 30 for $p = 600$ and $p = 1200$, and to 300 for $p = 2400$ and $p = 4800$. Notably, compared to [57; 59], our feature vector dimensions are larger, while the number of samples is evidently smaller.

**Experimental Settings:**

We compare our method with search methods, IGJO, VF-iDCA, LDMMA and BiC-GAFFA in this experiment. For the compared method BiC-GAFFA, we follow the recommended procedure outlined in [64]. For the other comparison methods, we adopt the exact settings from [57; 59]. For LDP-ADMM, we set $\beta_k = (1 + k)^{0.3}$, $\gamma = 10$ and the step size $e_k = 0.001$. The hyperparameters are initialized as $\lambda_i^0 = 0.1, i = 1, 2, ..., M$ and $\lambda_{M+1}^0 = 0.05$. For sparse group Lasso problem, the stopping criterion is set as $\|\mathbf{z}^{k+1} - \mathbf{z}^k\| / \|\mathbf{z}^{k+1}\| \leq 0.2$.

**Results and Discussions:**

From Table 2, we observe that LDPM (LDP-ADMM) achieves lowest test error and outperforms other algorithms in terms of time cost. As the scale of data increases, LDPM (LDP-ADMM) consistently finds the best hyperparameters and model solutions. In comparison, search methods become extremely unstable when facing dozens of hyperparameters. IGJO converges slowly and requires huge amount of computation. Similar to the experiments on the elastic net problem, LDMMA and VF-iDCA still exhibit a certain degree of overfitting. Both LDPM and BiC-GAFFA belong to the class of single-loop Hessian-free algorithms. Since LDPM (LDP-ADMM) employs projection to handle nonsmooth constraints, it achieves slightly better performance and efficiency compared to BiC-GAFFA.

### E.2.3 GROUP LASSO

Compared to the sparse group Lasso problem, this experiment removes the $\ell_1$-norm regularization term, leading to a reduction in the complexity of the LL problem. However, this omission also results in weaker control over the sparsity of **x**, potentially affecting the structure and interpretability of the solution. While the lower computational complexity may improve efficiency, the trade-off is a less strictly enforced sparsity constraint, which could affect the ability to capture key features in high-dimensional settings.

**Experimental Settings:**

The synthetic data is generated following the same procedure as described in Appendix E.2.2. For this experiment, we adopt the same settings for other compared algorithms as those used in the experiment for the sparse group Lasso problem in Appendix E.2.2. We conduct LDP-PGM with $\beta_k = (1+k)^{0.3}$, $e_k = 0.01$ and initial $\lambda_i^0 = (0.1, 0.1, ..., 0.1), i = 1, 2, ..., M$.

**Results and Discussions:**

We conduct experiments with different data scales and report numerical results over 10 repetitions in Table 6. The overall comparison results in Table 6 are similar to those in Table 2. In this case, LDPM (LDP-PGM) only requires projected gradient descent, leading to a significant improvement in efficiency.

To better evaluate scalability under the most challenging conditions, we report the total iterations and lower-level duality gaps on the largest-scale setting ($p = 4800$) for both Group Lasso and Sparse Group Lasso. Since these two tasks share similar bilevel structures, summarizing their lower-level optimality in a single table provides a clear comparison of efficiency across methods. As shown in Table NEW3, LDPM achieves the fewest iterations and the smallest duality gap, demonstrating superior convergence behavior in large-scale nonsmooth bilevel optimization.

Table 6: Group Lasso problems on the synthetic data, where $p$ represents the number of features.

| Settings | $p = 600$ | | | $p = 1200$ | | |
|---|---|---|---|---|---|---|
| | Time(s) | Val. Err. | Test Err. | Time(s) | Val. Err. | Test Err. |
| Grid | $5.72 \pm 1.69$ | $93.20 \pm 5.82$ | $96.07 \pm 17.50$ | $12.31 \pm 2.24$ | $93.15 \pm 4.74$ | $94.60 \pm 20.27$ |
| Random | $5.42 \pm 1.81$ | $148.69 \pm 6.55$ | $162.17 \pm 28.09$ | $11.38 \pm 2.56$ | $151.66 \pm 15.63$ | $160.88 \pm 17.07$ |
| IGJO | $1.42 \pm 0.25$ | $112.12 \pm 4.48$ | $105.99 \pm 15.09$ | $6.62 \pm 1.31$ | $143.62 \pm 15.42$ | $117.37 \pm 4.41$ |
| VF-iDCA | $0.50 \pm 0.14$ | $62.66 \pm 6.14$ | $84.52 \pm 12.46$ | $7.77 \pm 2.62$ | $95.02 \pm 7.04$ | $96.34 \pm 9.79$ |
| LDMMA | $0.51 \pm 0.12$ | $90.97 \pm 5.53$ | $79.68 \pm 16.19$ | $4.25 \pm 1.94$ | $92.32 \pm 8.05$ | $92.43 \pm 9.99$ |
| MEHA | $0.41 \pm 0.03$ | $78.82 \pm 6.91$ | $78.04 \pm 11.52$ | $3.11 \pm 0.26$ | $91.44 \pm 6.01$ | $89.36 \pm 8.20$ |
| BiC-GAFFA | $0.35 \pm 0.02$ | $74.16 \pm 6.91$ | $78.60 \pm 11.81$ | $2.27 \pm 0.26$ | $90.43 \pm 5.53$ | $87.79 \pm 8.43$ |
| LDPM | $\mathbf{0.29 \pm 0.03}$ | $\mathbf{70.44 \pm 6.85}$ | $\mathbf{70.92 \pm 9.71}$ | $\mathbf{1.81 \pm 0.12}$ | $\mathbf{88.92 \pm 6.41}$ | $\mathbf{82.76 \pm 6.51}$ |

| Settings | $p = 2400$ | | | $p = 4800$ | | |
|---|---|---|---|---|---|---|
| | Time(s) | Val. Err. | Test Err. | Time(s) | Val. Err. | Test Err. |
| Grid | $21.81 \pm 3.65$ | $105.19 \pm 15.54$ | $93.35 \pm 16.60$ | $42.38 \pm 5.71$ | $141.83 \pm 26.52$ | $126.95 \pm 19.38$ |
| Random | $19.95 \pm 6.17$ | $132.04 \pm 16.90$ | $161.45 \pm 18.37$ | $41.67 \pm 5.01$ | $109.35 \pm 18.21$ | $134.74 \pm 21.41$ |
| IGJO | $10.03 \pm 6.69$ | $100.75 \pm 16.47$ | $127.58 \pm 16.43$ | $26.78 \pm 8.50$ | $109.73 \pm 16.66$ | $117.14 \pm 8.23$ |
| VF-iDCA | $12.88 \pm 1.31$ | $69.53 \pm 5.90$ | $90.11 \pm 11.59$ | $40.61 \pm 2.79$ | $81.03 \pm 11.58$ | $105.70 \pm 10.05$ |
| LDMMA | $6.75 \pm 0.19$ | $72.85 \pm 8.22$ | $87.00 \pm 15.13$ | $32.53 \pm 3.29$ | $86.47 \pm 13.55$ | $105.39 \pm 10.37$ |
| MEHA | $5.32 \pm 0.16$ | $88.55 \pm 11.72$ | $84.93 \pm 10.38$ | $4.89 \pm 0.49$ | $99.92 \pm 8.88$ | $102.77 \pm 7.70$ |
| BiC-GAFFA | $4.60 \pm 0.09$ | $95.51 \pm 14.88$ | $84.02 \pm 9.46$ | $4.53 \pm 0.57$ | $103.77 \pm 9.01$ | $101.26 \pm 7.84$ |
| LDPM | $\mathbf{4.22 \pm 0.06}$ | $\mathbf{92.94 \pm 6.92}$ | $\mathbf{78.41 \pm 2.98}$ | $\mathbf{3.98 \pm 0.13}$ | $\mathbf{91.28 \pm 6.27}$ | $\mathbf{94.42 \pm 6.01}$ |

**Table NEW3:** Total iterations and lower-level duality gap on the largest-scale setting ($p = 4800$) for both Group Lasso and Sparse Group Lasso.

| Methods | Group Lasso ($p = 4800$) | | Sparse Group Lasso ($p = 4800$) | |
|---|---|---|---|---|
| | Total Iterations | Lower-level Duality Gap | Total Iterations | Lower-level Duality Gap |
| IGJO | $520 \pm 60$ | $1.46 \times 10^{-5}$ | $545 \pm 58$ | $1.52 \times 10^{-5}$ |
| VF-iDCA | $365 \pm 45$ | $6.92 \times 10^{-5}$ | $382 \pm 41$ | $7.15 \times 10^{-5}$ |
| LDMMA | $305 \pm 33$ | $4.21 \times 10^{-7}$ | $318 \pm 35$ | $4.56 \times 10^{-7}$ |
| MEHA | $255 \pm 23$ | $3.08 \times 10^{-6}$ | $268 \pm 24$ | $3.31 \times 10^{-6}$ |
| BiC-GAFFA | $220 \pm 20$ | $1.92 \times 10^{-7}$ | $233 \pm 19$ | $2.04 \times 10^{-7}$ |
| **LDPM (Ours)** | $\mathbf{172 \pm 16}$ | $\mathbf{6.11 \times 10^{-8}}$ | $\mathbf{185 \pm 18}$ | $\mathbf{5.94 \times 10^{-8}}$ |

### E.2.4 LOW-RANK MATRIX COMPLETION

We consider low-rank matrix completion problem on synthetic data. The formulation in (3) of the low-rank matrix completion is given as:

$$
\begin{aligned}
\min_{\boldsymbol{\theta},\boldsymbol{\beta},\Gamma} \quad & \sum_{(i,j)\in\Omega_{val}} |M_{ij} - \mathbf{x}_i\boldsymbol{\theta} - \mathbf{z}_j\boldsymbol{\beta} - \Gamma_{ij}|^2 \\
\text{s.t.} \quad & (\boldsymbol{\theta},\boldsymbol{\beta},\Gamma) \in \arg\min_{\boldsymbol{\theta},\boldsymbol{\beta},\Gamma} \left\{ \sum_{(i,j)\in\Omega_{tr}} |M_{ij} - \mathbf{x}_i\boldsymbol{\theta} - \mathbf{z}_j\boldsymbol{\beta} - \Gamma_{ij}|^2 \right. \\
& \left. + \lambda_0\|\Gamma\|_* + \sum_{g=1}^{G} \lambda_g\|\boldsymbol{\theta}^{(g)}\|_2 + \sum_{g=1}^{G} \lambda_{g+G}\|\boldsymbol{\beta}^{(g)}\|_2 \right\}
\end{aligned}
\tag{75}
$$

**Data Generation:**

The data generation procedure follows the approach in [14; 57]. Specifically, two entries per row and column are selected as the training set $\Omega_{tr}$, and one entry per row and column is selected as the validation set $\Omega_{val}$. The remaining entries form the test set $\Omega_{test}$. The row and column features are each grouped into 12 groups, with 3 covariates per group, resulting in $p = 36$ and $G = 12$.

The true coefficients are set as $\boldsymbol{\alpha}^{(g)} = g\mathbf{1}_3$ for $g = 1,\ldots,4$ and $\boldsymbol{\beta}^{(g)} = g\mathbf{1}_3$ for $g = 1,2$, with all other group coefficients set to zero. The low-rank effect matrix $\Gamma$ is generated as a rank-one matrix $\Gamma = \mathbf{u}\mathbf{v}^{\top}$, where $\mathbf{u}$ and $\mathbf{v}$ are sampled from the standard normal distribution.

The row features $X$ and column features $Z$ are also sampled from a standard normal distribution and then scaled so that the Frobenius norm of $X\boldsymbol{\alpha}\mathbf{1}^\top + (Z\boldsymbol{\beta}\mathbf{1}^\top)^\top$ matches that of $\Gamma$. Finally, the matrix observations are generated as

$$M_{ij} = \mathbf{x}_i^\top \boldsymbol{\alpha} + \mathbf{z}_j^\top \boldsymbol{\beta} + \Gamma_{ij} + \sigma \epsilon_{ij},$$

where $\epsilon_{ij}$ is standard Gaussian noise, and the noise level $\sigma$ is chosen such that the signal-to-noise ratio (SNR) equals 2.

**Experimental Settings:**

In this experiment, since multiple regularizers are involved, we employ LDP-ADMM. we compare LDP-ADMM with grid serach, random search, TPE, IGJO, VF-iDCA. For grid search, we explore two hyper-parameters $\mu_1$ and $\mu_2$ with the regularization parameters defined as $\lambda_0 = 10^{\mu_1}$ and $\lambda_g = 10^{\mu_2}$ for each $g = 1, \ldots, 2G$. A $10 \times 10$ grid uniformly spaced over the range $[-3.5, -1] \times [-3.5, -1]$ is employed, consistent with the approach of [14]. For both the random search and TPE methods, the optimization is conducted over transformed variables $u_g = \log_{10}(\lambda_m)$ for $m = 0, 1, 2, \ldots, 2G$, where each $u_g$ is drawn from a uniform distribution on the interval $[-3.5, -1]$. For IGJO, the initial values for the regularization vector $\boldsymbol{\lambda}$ are set to $[0.005, 0.005, \ldots, 0.005]$. For VF-iDCA, the initial guess for the auxiliary parameter $\mathbf{r}$ is chosen as $[1, 0.1, 0.1, \ldots, 0.1]$. The algorithm is terminated when the stopping criterion $(\|\mathbf{z}^{k+1} - \mathbf{z}^k\|)/\|\mathbf{z}^k\| \leq 0.1$ is satisfied. For LDP-ADMM, we set $\beta_k = (1 + k)^{0.3}$, $\gamma = 10$, step size $e_k = 0.025$ and initial $\lambda_i = 0.05, i = 0, 1, 2, ..., 2G$.

**Results and Discussions:**

Throughout all experiments, feature grouping is performed sequentially as follows, every three consecutive features are assigned to the same group, starting from the first feature onward.

We present the statistical results in repeated experiments in Table 7. VF-iDCA and LDPM (LDP-ADMM) incur longer runtimes than search methods because they perform more intensive iterative updates. VF-iDCA leverages inexact DC-programming steps to more faithfully enforce the low-rank and group-sparsity penalties. This additional computational effort yields tighter approximation of the underlying low-rank factors, resulting in substantially lower validation and test errors. LDPM (LDP-ADMM) repeatedly perform costly matrix projections as discussed in Appendix C.2 to enforce the rank constraints accurately. These intensive projection steps allow them to recover the underlying low-rank structure more precisely, which translates into substantially lower validation and test errors.

Table 7: Low-rank matrix completion problems on synthetic data

| Methods | Time(s) | Val. Acc. | Test Acc. |
|---------|---------|-----------|-----------|
| Grid | $21.02 \pm 0.95$ | $0.71 \pm 0.21$ | $0.76 \pm 0.20$ |
| Random | $33.12 \pm 2.10$ | $0.72 \pm 0.22$ | $0.79 \pm 0.19$ |
| TPE | $36.80 \pm 9.45$ | $0.69 \pm 0.20$ | $0.75 \pm 0.18$ |
| IGJO | $1205.0 \pm 312.5$ | $0.67 \pm 0.20$ | $0.71 \pm 0.17$ |
| VF-iDCA | $55.20 \pm 12.05$ | $0.65 \pm 0.18$ | $0.69 \pm 0.15$ |
| LDPM | $62.10 \pm 15.31$ | $\mathbf{0.58 \pm 0.14}$ | $\mathbf{0.66 \pm 0.13}$ |

### E.2.5 ROBUST REGRESSION

**Experimental Settings:**

Robust Regression is captured in (3) as:

$$\begin{aligned} \min_{\mathbf{x}} \quad & l_\delta(A_{val}\mathbf{x} - \mathbf{b}_{val}) \\ \text{s.t.} \quad & \mathbf{x} \in \arg\min_{\hat{\mathbf{x}}} \ l_\delta(A_{tr}\hat{\mathbf{x}} - \mathbf{b}_{tr}) + \lambda_1 \|\hat{\mathbf{x}}\|_1 + \frac{\lambda_2}{2}\|\hat{\mathbf{x}}\|_2^2, \end{aligned} \tag{76}$$

where $l_\delta$ is defined in Table 1 and we select $\delta = 1.345$. The synthetic data are generated following the same methodology as in the elastic net experiments, detailed in Appendix E.2.1. In this setting, the regression loss is replaced by the Huber loss to enhance robustness against outliers and we also employ LDP-ADMM due to the presence of multiple regularizers. Since [64] does not provide experiments or code for the robust regression problem, we compare only with search-based methods, IGJO, IFDM, VF-iDCA and LDMMA in this experiment. All algorithms are implemented under the same settings as those described in Appendix E.2.1. For LDP-ADMM, we set $\beta_k = (1 + k)^{0.3}$, $e_k = 0.1$, $\gamma = 10$ and initial $\lambda_1^0 = 0.1, \lambda_2^0 = 0.05$. For the robust regression problem, the stopping criterion is set as $\|\mathbf{z}^{k+1} - \mathbf{z}^k\|/\|\mathbf{z}^{k+1}\| \leq 0.1$.

**Results and Discussions:**

We conduct repeated experiments with 10 randomly generated synthetic data, and calculate the mean and variance. The numerical results on robust regression are reported in Table 8. Overall, LDPM (LDP-ADMM) achieves the lowest test error while maintaining a significantly reduced time cost, especially for large-scale datasets. In contrast, the search methods incur a high computational cost and exhibit poor performance on the test dataset. The gradient-based method IGJO demonstrates slightly better accuracy and efficiency but converges very slowly.

Table 8: Robust regression problems with Huber loss on synthetic data, where $|I_{tr}|$, $|I_{val}|$, $|I_{te}|$ and $p$ represent the number of training observations, validation observations, predictors and features, respectively.

| Settings | Methods | Time(s) | Val. Err. | Test Err. | Settings | Time(s) | Val. Err. | Test Err. |
|---|---|---|---|---|---|---|---|---|
| | Grid | $6.22 \pm 0.55$ | $6.28 \pm 1.05$ | $6.12 \pm 1.00$ | | $33.21 \pm 3.74$ | $7.62 \pm 1.42$ | $8.59 \pm 0.88$ |
| | Random | $6.37 \pm 1.12$ | $4.25 \pm 1.06$ | $6.19 \pm 1.01$ | | $34.07 \pm 2.59$ | $6.32 \pm 1.46$ | $8.48 \pm 0.92$ |
| $|I_{tr}| = 100$ | IGJO | $3.74 \pm 1.98$ | $4.35 \pm 0.94$ | $4.25 \pm 0.91$ | $|I_{tr}| = 100$ | $30.82 \pm 6.18$ | $7.59 \pm 1.09$ | $8.45 \pm 0.79$ |
| $|I_{val}| = 100$ | IFDM | $1.21 \pm 0.35$ | $4.40 \pm 1.01$ | $4.26 \pm 1.02$ | $|I_{val}| = 100$ | $3.88 \pm 2.15$ | $7.45 \pm 0.73$ | $8.02 \pm 1.32$ |
| $|I_{te}| = 250$ | VF-iDCA | $4.63 \pm 1.62$ | $2.28 \pm 1.42$ | $4.39 \pm 1.05$ | $|I_{te}| = 100$ | $22.98 \pm 4.75$ | $2.15 \pm 0.88$ | $4.50 \pm 0.73$ |
| $p = 250$ | LDMMA | $0.95 \pm 0.08$ | $2.34 \pm 1.01$ | $5.52 \pm 0.74$ | $p = 2500$ | $15.89 \pm 1.39$ | $2.41 \pm 0.92$ | $5.61 \pm 1.12$ |
| | MEHA | $0.82 \pm 0.06$ | $1.87 \pm 0.55$ | $4.08 \pm 0.66$ | | $8.12 \pm 0.52$ | $1.98 \pm 0.40$ | $4.92 \pm 0.58$ |
| | BiC-GAFFA | $0.75 \pm 0.05$ | $1.65 \pm 0.48$ | $3.76 \pm 0.54$ | | $6.25 \pm 0.47$ | $1.80 \pm 0.36$ | $4.63 \pm 0.50$ |
| | LDPM | $\mathbf{0.60 \pm 0.07}$ | $1.29 \pm 0.42$ | $\mathbf{3.02 \pm 0.38}$ | | $\mathbf{4.65 \pm 0.09}$ | $1.58 \pm 0.13$ | $\mathbf{3.78 \pm 0.60}$ |

### E.3 SENSITIVITY OF PARAMETERS

In this part, we conduct experiments to analyze the sensitivity of our methods to different parameter combinations. We evaluate both LDP-PGM (Algorithm 1) and LDP-ADMM (Algorithm 2). To investigate the parameter sensitivity of LDP-PGM, we carry out supplementary experiments on the group Lasso problem with a problem dimension of 1200. In each trial, we vary one parameter while keeping the others fixed. The corresponding convergence times and projected gradient descent (PGD) iteration counts are summarized in Table 9a. A similar analysis is also performed for LDP-ADMM on the sparse group Lasso instance, also with a dimension of 1200. The convergence performance, including time and steps, is likewise reported in Table 9b.

In LDP-ADMM, larger $\gamma$ enforces the constraint more aggressively, so the primal residual in $\mathbf{z}$-subproblem drops quickly. Smaller $\gamma$ makes $\mathbf{z}$-update more flexible, but the residual decays more slowly, so it end up needing more iterations and longer overall runtime. As presented in Table 9, the algorithm consistently achieves convergence and exhibits strong robustness across a broad spectrum of parameter configurations, highlighting its stability and reliability under varying conditions.

### E.4 EXPERIMENTAL ON REAL-WORLD DATASETS

This section of the experiments aims to demonstrate the numerical performance of our method on real-world datasets.

| Strategy | $e_k$ | $\underline{\beta}$ | $p$ | Steps | Time(s) |
|---|---|---|---|---|---|
| Original | 0.01 | 1 | 0.3 | 29 | 2.04 |
| $e_k$ | 0.005 | 1 | 0.3 | 42 | 3.75 |
| | 0.05 | 1 | 0.3 | 18 | 1.67 |
| | 0.08 | 1 | 0.3 | 14 | 1.42 |
| $\underline{\beta}$ | 0.01 | 2 | 0.3 | 40 | 3.60 |
| | 0.01 | 10 | 0.3 | 44 | 3.89 |
| | 0.01 | 40 | 0.3 | 38 | 3.95 |
| $p$ | 0.01 | 1 | 0.05 | 95 | 11.72 |
| | 0.01 | 10 | 0.15 | 56 | 4.85 |
| | 0.01 | 40 | 0.5 | 31 | 2.93 |

(a) Parameter Sensitivity for LDP-PGM

| Strategy | $e_k$ | $\underline{\beta}$ | $p$ | $\gamma$ | Steps | Time(s) |
|---|---|---|---|---|---|---|
| Original | 0.01 | 1 | 0.3 | 10 | 36 | 2.30 |
| $e_k$ | 0.005 | 1 | 0.3 | 10 | 49 | 4.97 |
| | 0.05 | 1 | 0.3 | 10 | 21 | 1.89 |
| | 0.08 | 1 | 0.3 | 10 | 17 | 1.54 |
| $\underline{\beta}$ | 0.01 | 2 | 0.3 | 10 | 48 | 4.16 |
| | 0.01 | 10 | 0.3 | 10 | 56 | 4.35 |
| | 0.01 | 40 | 0.3 | 10 | 52 | 5.15 |
| $p$ | 0.01 | 1 | 0.05 | 10 | 129 | 16.57 |
| | 0.01 | 10 | 0.15 | 10 | 58 | 6.12 |
| | 0.01 | 40 | 0.5 | 10 | 72 | 8.83 |
| $\gamma$ | 0.01 | 1 | 0.3 | 5 | 62 | 5.12 |
| | 0.01 | 1 | 0.3 | 20 | 39 | 2.48 |

(b) Parameter Sensitivity for LDP-ADMM

Table 9: Parameter Sensitivity Analysis for LDP-PGM and LDP-ADMM

### E.4.1 ELASTIC NET

**Data Introduction:**

We consider elastic net problem on high dimendional datasets gisette and sensit. The mathmatical formulation follows (72). The datasets have a large number of features, which are suitable for evaluating the performance of regularization techniques like the elastic net. Following the approach in [57], we partition the datasets as follows: 50 and 25 examples are extracted as the training set, respectively; 50 and 25 examples are used as the validation set, respectively; and the remaining data was reserved for testing.

**Experimental Settings:**

For the same reasons as in Appendix E.2.1, we also compare LDP-ADMM with search method, IGJO, IFDM, VF-iDCA and LDMMA in this experiment. We conduct compared algorithms with the same settings as [57; 59]. For LDP-ADMM, we adopt the same settings as those used in Appendix E.2.1. The stopping criterion in this experiment is also set as $\|\mathbf{z}^{k+1} - \mathbf{z}^k\|/\|\mathbf{z}^{k+1}\| \leq 0.1$.

**Results and Discussions:**

We report the experimental results in Figure 1 and summarize them in Table 10 as auxiliary experimental results. These demonstrate that LDPM (LDP-ADMM) consistently achieves competitive performance while maintaining fast computational speeds on real-world datasets for elastic net problems.

Table 10: Elastic net problem on datasets gisette and sensit, where $|I_{tr}|$, $|I_{val}|$, $|I_{te}|$ and $p$ represent the number of training samples, validation samples, test samples and features, respectively.

| Dataset | Methods | Time(s) | Val. Err. | Test Err. | Dataset | Time(s) | Val. Err. | Test Err. |
|---|---|---|---|---|---|---|---|---|
| gisette | Grid | $37.21 \pm 4.80$ | $0.24 \pm 0.02$ | $0.24 \pm 0.02$ | sensit | $1.62 \pm 0.19$ | $1.41 \pm 0.75$ | $1.33 \pm 0.47$ |
| | Random | $56.67 \pm 9.55$ | $0.22 \pm 0.05$ | $0.26 \pm 0.02$ | | $1.46 \pm 0.12$ | $1.52 \pm 0.58$ | $1.48 \pm 0.43$ |
| | IGJO | $18.24 \pm 3.17$ | $0.24 \pm 0.02$ | $0.23 \pm 0.03$ | | $0.57 \pm 0.14$ | $0.52 \pm 0.18$ | $0.61 \pm 0.14$ |
| | IFDM | $35.40 \pm 0.74$ | $0.22 \pm 0.02$ | $0.23 \pm 0.03$ | | $6.35 \pm 0.04$ | $0.37 \pm 0.10$ | $0.41 \pm 0.23$ |
| | VF-iDCA | $10.75 \pm 2.72$ | $0.01 \pm 0.00$ | $0.22 \pm 0.01$ | | $0.47 \pm 0.06$ | $0.27 \pm 0.03$ | $0.52 \pm 0.06$ |
| | LDMMA | $9.45 \pm 2.98$ | $0.01 \pm 0.00$ | $0.21 \pm 0.01$ | | $0.41 \pm 0.05$ | $0.25 \pm 0.04$ | $0.50 \pm 0.04$ |
| | LDPM | $4.85 \pm 0.23$ | $0.09 \pm 0.05$ | $0.14 \pm 0.03$ | | $0.28 \pm 0.02$ | $0.08 \pm 0.01$ | $0.34 \pm 0.05$ |

As described in [57; 59], the implementation of VF-iDCA and LDMMA relies heavily on optimization solvers. In particular, the subproblems of LDMMA are entirely dependent on the commercial solver MOSEK, while the subproblems of VF-iDCA also rely on the CVXPY package, utilizing ECOS or CSC as solvers. For large-scale datasets, frequent solver calls can become a major computational bottleneck, limiting the scalability of these methods in high-dimensional or complex problem settings. Furthermore, the conic programming reformulation proposed in [59] introduces second-order cone constraints, making LDMMA inherently a second-order algorithm. Consequently, its efficiency deteriorates significantly when applied to large-scale problems.

In this experiment, we omit the validation/test error-vs-time curves in Figure 1 for both the grid/random search methods and IFDM because their numerical instability leads to highly erratic traces. As discussed in [14; 54], implicit differentiation methods can suffer from numerical instability when applied to problems with sparse regularization like elastic net. In such cases, the inner optimization problems often have poor conditioning, causing oscillatory behavior during convergence.

Similar to the experiments on synthetic data, We report the total iterations, lower-level duality gaps, and sparsity levels for all methods on the real elastic net datasets in Table NEW4. Across both datasets, LDPM achieves the smallest duality gaps and the fewest iterations, while also producing solutions whose sparsity is closest to the ground truth, reflecting both superior convergence efficiency and better generalization performance.

**Table NEW4:** Total iterations, lower-level duality gap, and sparsity comparison for elastic net on real datasets (gisette and sensit).

| Methods | gisette | | | sensit | | |
|---|---|---|---|---|---|---|
| | **Total Iter.** | **LL Gap** | **Sparsity(%)** | **Total Iter.** | **LL Gap** | **Sparsity(%)** |
| Grid Search | / | / | 10.0 | / | / | 7.0 |
| Random Search | / | / | 10.0 | / | / | 7.0 |
| IGJO | $520 \pm 60$ | $1.6 \times 10^{-5}$ | $11.8 \pm 1.9$ | $430 \pm 51$ | $1.8 \times 10^{-5}$ | $8.4 \pm 1.3$ |
| IFDM | $415 \pm 48$ | $9.4 \times 10^{-4}$ | $11.2 \pm 1.6$ | $365 \pm 45$ | $1.1 \times 10^{-4}$ | $7.9 \pm 1.2$ |
| VF-iDCA | $285 \pm 32$ | $3.8 \times 10^{-6}$ | $26.3 \pm 4.0$ | $240 \pm 30$ | $4.1 \times 10^{-6}$ | $19.8 \pm 3.7$ |
| LDMMA | $242 \pm 27$ | $5.7 \times 10^{-7}$ | $28.1 \pm 4.9$ | $210 \pm 25$ | $6.5 \times 10^{-7}$ | $22.0 \pm 4.1$ |
| **LDPM (Ours)** | $\mathbf{132 \pm 14}$ | $\mathbf{7.5 \times 10^{-8}}$ | $\mathbf{10.7 \pm 1.3}$ | $\mathbf{118 \pm 12}$ | $\mathbf{8.9 \times 10^{-8}}$ | $\mathbf{7.6 \pm 1.0}$ |

### E.4.2 SMOOTHED SUPPORT VECTOR MACHINE

The smoothed support vector machine incorporates smoothed hinge loss function and squared $\ell_2$-norm regularization. The formulation in (3) of the smoothed support vector machine is given as:

$$
\begin{aligned}
\min_{\mathbf{x}, \lambda} \quad & \sum_{i \in I_{val}} l_h(-b_i \mathbf{a}_i^T \mathbf{x}) \\
\text{s.t.} \quad & \mathbf{x} \in \arg\min_{\hat{\mathbf{x}}} \sum_{i \in I_{tr}} l_h(b_i \mathbf{a}_i^T \hat{\mathbf{x}}) + \frac{\lambda}{2} \|\hat{\mathbf{x}}\|_2^2,
\end{aligned}
\tag{77}
$$

where $l_h$ denotes the smoothed hinge loss function detailed in Table 1. Since there is only one regularization term in (77), we conduct LDP-PGM due to the single regularizer.

**Data Introduction:**

We use the LIBSVM toolbox [107] [5] to load the datasets and extract the corresponding observation matrix and label vector for each dataset. Each dataset is divided into two separate parts: a cross-validation training set $\Omega$ consisting of $3\lfloor N/6 \rfloor$ samples, and a test set $\Omega_{\text{test}}$ containing the remaining samples. Within this division, the training set is further partitioned into multiple equal parts, and we iteratively use one part as

---

[5]https://www.csie.ntu.edu.tw/ cjlin/libsvmtools/datasets/

the validation set while utilizing the remaining parts as the training set to solve the SVM problem. For the experiments, we conducted 6-fold cross-validation on the training and validation sets across all three datasets to optimize the hyperparameters.

**Experimental Settings:**

During the process of solving the smoothed support vector machine problem with $K$-fold cross-validation, the loss function on the validation set is defined as follows:

$$\Theta_{val}(\mathbf{x}^1, \mathbf{x}^2, \ldots, \mathbf{x}^K, \mathbf{c}) := \frac{1}{K} \sum_{k=1}^{K} \frac{1}{|\Omega_{val}^k|} \sum_{j \in \Omega_{val}^k} l_h(b_j \mathbf{a}_j^T \mathbf{x}^k), \tag{78}$$

Following the approach used for support vector machine [23], we reformulate the primal problem into the following bilevel optimization model for the smoothed support vector machine:

$$\begin{aligned} \min_{\mathbf{x}, c} \quad & \Theta_{val}(\mathbf{x}^1, \mathbf{x}^2, \ldots, \mathbf{x}^K, \mathbf{c}) \\ \text{s.t.} \quad & \lambda > 0, \bar{\mathbf{x}}_{lb} \leq \bar{\mathbf{x}} \leq \bar{\mathbf{x}}_{ub} \\ & \mathbf{x}^k \in \operatorname*{arg\,min}_{-\bar{\mathbf{x}} \leq \mathbf{x} \leq \bar{\mathbf{x}}} \left\{ \sum_{j \in \Omega_{tr}^k} l_h(b_j \mathbf{a}_j^T \mathbf{x}) + \frac{\lambda}{2} \|\mathbf{x}\|_2^2 \right\}, k = 1, 2, \ldots, K, \end{aligned} \tag{79}$$

where $\mathbf{x}^1, \mathbf{x}^2, \ldots, \mathbf{x}^K$ are $K$ parallel copies of $c$ and $\mathbf{x}$. $\bar{\mathbf{x}}_{ub}$ and $\bar{\mathbf{x}}_{lb}$ are the upper and lower bounds of $\bar{\mathbf{x}}$. Similarly, we define the loss function on the training set in a manner analogous to (78):

$$\Theta_{tr}(\mathbf{x}^1, \mathbf{x}^2, \ldots, \mathbf{x}^K, \mathbf{c}) := \frac{1}{K} \sum_{k=1}^{K} \frac{1}{|\Omega_{tr}^k|} \sum_{j \in \Omega_{tr}^k} l_h(b_j \mathbf{a}_j^T \mathbf{x}^k). \tag{80}$$

We also implement other competitive methods following the effective practice in [57; 59]. For LDP-PGM, the penalty parameter is configured as $\beta_k = (1+k)^{0.3}$ and the step size in each iteration is fixed at $e_k = 0.1$. For the hyperparameter, we set initial value as $\lambda^0 = 0.1$ for LDP-PGM.

**Results and Discussions:**

We plot the convergence curves of each algorithm for validation and test error in Figure 2. From Figure 3, we observe that LDPM (LDP-PGM) consistently achieves the lowest validation and test errors across all datasets (diabetes, sonar, a1a). In particular, its convergence curves drop rapidly at the early stage and remain stable afterwards, while the competing methods converge more slowly or plateau at higher error levels. This demonstrates that LDPM not only converges faster but also generalizes better, highlighting its superiority in both efficiency and accuracy.

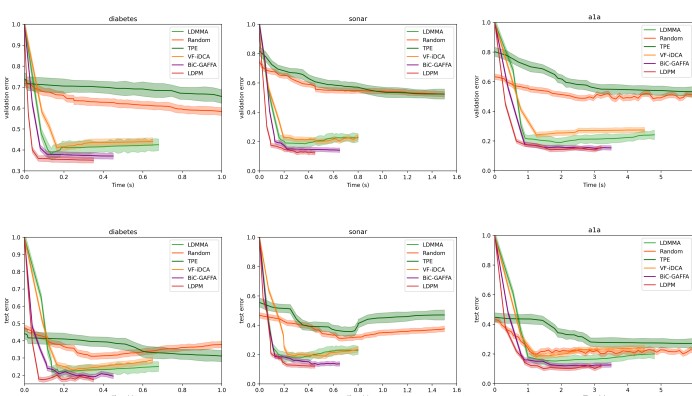

Figure 2: Comparison of the algorithms for SSVM problem on real-world datasets.

### E.4.3 SPARSE LOGISTIC REGRESSION

The sparse logistic regression [24] is equipped with logistic loss function and $\ell_1$-norm regularization. Its formulation in (3) is

$$
\begin{aligned}
\min_{\mathbf{x},\lambda} \quad & \sum_{i \in I_{val}} \log(1 + e^{-b_i \mathbf{a}_i^T \mathbf{x}}) \\
\text{s.t.} \quad & \mathbf{x} \in \arg\min_{\hat{\mathbf{x}}} \sum_{i \in I_{tr}} \log(1 + e^{-b_i \mathbf{a}_i^T \hat{\mathbf{x}}}) + \lambda \|\hat{\mathbf{x}}\|_1.
\end{aligned}
\tag{81}
$$

According to the definition of $\varphi$, we observe that the logistic loss can be abstracted with $\varphi(t) = \log(1+e^{-t})$ and $A_t\mathbf{x} - \mathbf{b}_t = ((\mathbf{b}_{tr}A_{tr})\mathbf{x})$. Correspondingly, the conjugate is calculated as $\varphi^*(v) = -v\log(v) - (1-v)\log(1-v)$ if $0 < v < 1$ and $\varphi^*(v) = \infty$ otherwise.

**Data Introduction:**

Following the experimental setup in [15], we conduct our evaluations on large-scale real-world datasets. Specifically, we use the same datasets as [15], namely news20, rcv1, real-sim and webspam, all of which can be downloaded from LIBSVM website. Table 11 provides a brief introduction to the basic characteristics of these three datasets.

Table 11: Dataset Overview

| Datasets | Samples | Features | Sparsity | Ratio |
|---|---|---|---|---|
| news20.binary | $19,996$ | $1,355,191$ | $0.034\%$ | $0.5236$ |
| rcv1.binary | $20,242$ | $47,236$ | $0.155\%$ | $0.46948$ |
| real-sim | $72,309$ | $20,958$ | $0.245\%$ | $0.33113$ |
| webspam | $350,000$ | $16,609,143$ | $0.024\%$ | $0.6657$ |

**Experimental Settings:**

Due to the single regularizer, we also apply LDP-PGM in this experiment. This experiment is initially conducted in [15]. Since VF-iDCA and LDMMA are not suitable for solving large-scale problems, and the reformulation of LDMMA is not applicable to the logistic loss function, we do not compare these algorithms in this experiment. We compare our method with search methods, IFDM, and BiC-GAFFA. Random search uniformly samples 50 hyperparameter values in the interval $[\lambda_{\max} - 4\log(10), \lambda_{\max}]$. The algorithm settings for IFDM follow the configurations in [15] for each real dataset without modification. For BiC-GAFFA, we

use $\gamma_1 = 10, \gamma_2 = 0.01, \eta_k = 0.01, r = 5, \alpha_k = 0.01, \rho = 0.3$, with a maximum iteration limit of 1000. For LDP-PGM, we set $\beta_k = (1 + k)^{0.3}, e_k = 0.05$ and initial $\lambda^0 = 0.5$. In addition, we consider Assumption 3.2, $\varphi(t)$ is $\frac{1}{4}$-smooth and satisfies it. In contrast, $\varphi^*$ is only gradient Lipschitz on any compact set of $(0, 1)$. Therefore, in our implementation, we enforce a simple numerical safeguard by truncating the dual variable $\boldsymbol{\xi}$ onto $\min(\max(\xi_i, \epsilon), 1 - \epsilon)$ with $\epsilon = 10^{-6}$. This ensures that all iterates remain within a compact domain, thereby guaranteeing that Assumption 3.2 is satisfied in our experiments. Moreover, in practice we observe that the iterates never approach the boundary 0 or 1, so the safeguard is never activated but provides theoretical soundness.

**Results and Discussions:**

In this experiment, we implement the code provided in [15]. Each experiment is repeated 10 times to compute the average and variance of runtime, validation error, validation accuracy, test error, and test accuracy. The convergence curves of each algorithm with respect to validation and test error are illustrated in Figure 3. Additionally, we calculate the corresponding accuracy and report them in Table 12.

Overall, we observe from Figure 3 and Table 12 that LDPM (LDP-PGM) achieves the lowest time cost and test error in the experiment on sparse logistic regression. The comprehensive experimental results provide strong evidence of the efficiency and practicality of our algorithm in addressing bilevel hyperparameter optimization. These results highlight its effectiveness in real-world applications, demonstrating its ability to achieve superior performance while maintaining computational efficiency.

Table 12: Accuracy of sparse logistic regression problem on real-world datasets.

| Dataset | Methods | Time(s) | Val. Acc. | Test Acc. |
|---------|---------|---------|-----------|-----------|
| news20.binary | Random | $654.63 \pm 33.26$ | $81.49 \pm 1.10$ | $80.89 \pm 1.24$ |
| | IFDM | $41.16 \pm 6.81$ | $86.87 \pm 1.14$ | $84.07 \pm 1.09$ |
| | MEHA | $35.42 \pm 4.92$ | $89.85 \pm 1.08$ | $89.41 \pm 0.92$ |
| | BiC-GAFFA | $32.64 \pm 4.48$ | $90.98 \pm 1.03$ | $90.17 \pm 0.81$ |
| | LDPM | $30.85 \pm 3.29$ | $90.59 \pm 1.15$ | $92.94 \pm 0.73$ |
| rcv1.binary | Random | $214.46 \pm 67.15$ | $96.51 \pm 1.19$ | $94.24 \pm 2.39$ |
| | IFDM | $21.08 \pm 5.47$ | $97.95 \pm 0.26$ | $96.12 \pm 1.29$ |
| | MEHA | $17.82 \pm 1.26$ | $98.41 \pm 0.22$ | $96.21 \pm 1.11$ |
| | BiC-GAFFA | $15.92 \pm 0.94$ | $98.72 \pm 0.25$ | $96.50 \pm 1.21$ |
| | LDPM | $14.13 \pm 1.43$ | $98.70 \pm 0.33$ | $97.92 \pm 1.29$ |
| real-sim | Random | $624.45 \pm 38.03$ | $68.30 \pm 1.10$ | $67.65 \pm 1.23$ |
| | IFDM | $25.86 \pm 1.57$ | $91.23 \pm 2.18$ | $91.10 \pm 1.31$ |
| | MEHA | $20.93 \pm 0.88$ | $92.75 \pm 1.64$ | $91.46 \pm 1.88$ |
| | BiC-GAFFA | $18.08 \pm 0.71$ | $93.28 \pm 1.48$ | $91.68 \pm 2.42$ |
| | LDPM | $17.93 \pm 0.68$ | $95.10 \pm 1.13$ | $94.19 \pm 1.57$ |
| webspam | Random | $712.34 \pm 41.28$ | $92.15 \pm 0.74$ | $91.68 \pm 0.82$ |
| | IFDM | $38.92 \pm 2.17$ | $96.84 \pm 0.38$ | $96.57 \pm 0.41$ |
| | MEHA | $28.53 \pm 1.12$ | $97.23 \pm 0.35$ | $96.88 \pm 0.33$ |
| | BiC-GAFFA | $26.47 \pm 1.03$ | $97.52 \pm 0.31$ | $97.28 \pm 0.29$ |
| | LDPM | $22.63 \pm 0.88$ | $97.93 \pm 0.27$ | $97.64 \pm 0.25$ |

We further report the total iterations, lower-level duality gap, and sparsity for all methods on the real sparse logistic regression tasks in Table NEW5. These metrics provide a more direct evaluation of how accurately and efficiently each algorithm solves the underlying bilevel optimization problem. Across all datasets, LDPM achieves the smallest LL duality gaps and the fewest iterations, indicating a more precise enforcement of the lower-level optimality. Moreover, LDPM consistently yields the sparsest solutions, demonstrating superior structural recovery and generalization compared with existing bilevel algorithms.

## F   FURTHER DISCUSSIONS

LDPM effectively solves bilevel optimization problems of the form (3), as demonstrated by strong empirical results. However, the core of LDPM relies on a projected gradient descent, which currently cannot handle nonsmooth loss functions without dedicated solvers, such as the hinge loss in SVMs. In contrast, [57; 59] circumvent this issue by leveraging existing solvers to deal with such nonsmooth components.

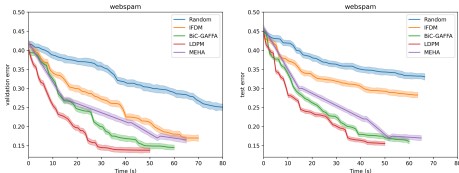

**Figure NEW1**: Comparison of the algorithms for sparse logistic regression on webspam datasets.

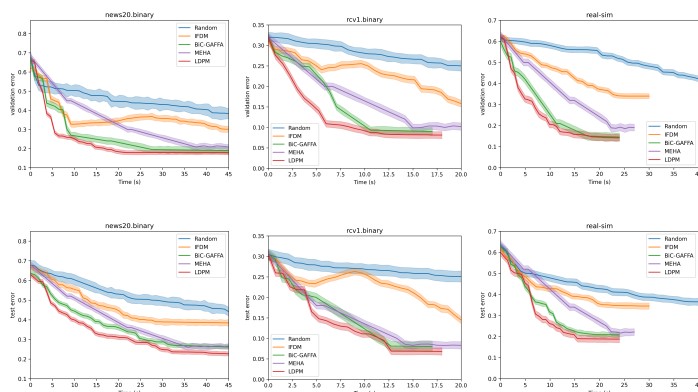

Figure 3: Comparison of the algorithms for sparse logistic regression on real-world datasets.

**Table NEW5:** Total iterations, LL duality gap, and sparsity comparison for sparse logistic regression.

| **news20.binary** | | | | **rcv1.binary** | | | |
|---|---|---|---|---|---|---|---|
| Methods | Total Iter. | LL Duality Gap | Sparsity(%) | Methods | Total Iter. | LL Duality Gap | Sparsity(%) |
| Random | / | / | $4.1 \pm 0.3$ | Random | / | / | $10.3 \pm 0.7$ |
| IFDM | $280 \pm 35$ | $3.203 \times 10^{-6}$ | $6.4 \pm 0.5$ | IFDM | $210 \pm 30$ | $2.814 \times 10^{-6}$ | $5.8 \pm 0.4$ |
| MEHA | $185 \pm 22$ | $2.137 \times 10^{-7}$ | $7.3 \pm 0.6$ | MEHA | $165 \pm 18$ | $1.824 \times 10^{-7}$ | $6.1 \pm 0.5$ |
| BiC-GAFFA | $162 \pm 19$ | $1.487 \times 10^{-7}$ | $7.0 \pm 0.4$ | BiC-GAFFA | $148 \pm 16$ | $1.271 \times 10^{-7}$ | $5.9 \pm 0.4$ |
| **LDPM** | $\mathbf{120 \pm 14}$ | $\mathbf{4.531 \times 10^{-8}}$ | $\mathbf{5.8 \pm 0.3}$ | **LDPM** | $\mathbf{110 \pm 12}$ | $\mathbf{3.873 \times 10^{-8}}$ | $\mathbf{4.6 \pm 0.3}$ |
| **real-sim** | | | | **webspam** | | | |
| Methods | Total Iter. | LL Duality Gap | Sparsity(%) | Methods | Total Iter. | LL Duality Gap | Sparsity(%) |
| Random | / | / | $11.8 \pm 0.8$ | Random | / | / | $14.5 \pm 1.0$ |
| IFDM | $190 \pm 21$ | $3.067 \times 10^{-6}$ | $6.2 \pm 0.5$ | IFDM | $260 \pm 28$ | $4.512 \times 10^{-6}$ | $10.2 \pm 0.7$ |
| MEHA | $150 \pm 15$ | $1.927 \times 10^{-7}$ | $6.7 \pm 0.4$ | MEHA | $180 \pm 20$ | $2.732 \times 10^{-7}$ | $8.8 \pm 0.6$ |
| BiC-GAFFA | $132 \pm 14$ | $1.403 \times 10^{-7}$ | $6.5 \pm 0.4$ | BiC-GAFFA | $163 \pm 17$ | $1.932 \times 10^{-7}$ | $8.5 \pm 0.5$ |
| **LDPM** | $\mathbf{102 \pm 11}$ | $\mathbf{3.184 \times 10^{-8}}$ | $\mathbf{4.4 \pm 0.3}$ | **LDPM** | $\mathbf{115 \pm 13}$ | $\mathbf{5.287 \times 10^{-8}}$ | $\mathbf{8.1 \pm 0.4}$ |

