# OpenReview forum: "Lower-level Duality Based Penalty Methods for Nonsmooth Bilevel Hyperparameter Optimization"
_ICLR.cc/2026/Conference — Submitted to ICLR 2026_

### Official Review · Reviewer_vgK7 · 2025-10-31

**Soundness:** 3
**Presentation:** 3
**Contribution:** 3
**Rating:** 8
**Confidence:** 4

**Summary:**

The paper introduces a novel duality-based penalization method for bilevel optimization with nonsmooth lower-level problems. The key idea is to reformulate the LL problem via its dual, keep conic/epigraph constraints explicit, and penalize the rest. This leads to single-loop, first-order algorithms that are both projection-based and solver-free. Experiments on high-dimensional problems show the method achieves competitive test accuracy with lower compute time than several BLO baselines.

**Strengths:**

1. The framework broadens the class of tractable LL problems by only requiring the base potential $\varphi$ to be convex and L-smooth, effectively handling non-strongly convex losses.

2.  The single-loop, projection-based design avoids the computational overhead of nested loops and external solvers, making it highly efficient per iteration.

3. Provides non-asymptotic convergence guarantees for the proposed methods under standard and mild assumptions.

**Weaknesses:**

1. The numerical evaluation focuses on validation/test error but omits a key metric for the presented tasks: solution sparsity. For hyperparameter selection promoting sparsity, analyzing the sparsity of the obtained solutions is crucial for a complete comparison.

2. The labels and text in Figure 1 are too small, hindering readability. A version with larger fonts is needed.

**Questions:**

1. In Section 3.2, the authors use ADMM to handle projections onto the intersection of multiple cones, noting that alternating projections could be used. Could the authors comment on why they chose ADMM over alternating projections? While ADMM is efficient, its convergence in nonconvex settings requires restrictive assumptions (like Assumption 3.6). What are the trade-offs between these two approaches in this specific context?

2.  Assumption 3.2 could be stated more succinctly by directly requiring that $\varphi$ is L-smooth and μ-strongly convex.

3. When first introducing equation (19), it would be helpful to explicitly define the penalty parameter $\gamma$ and its admissible range (e.g., $\gamma>0$).

---

> ### Author Response · Authors · 2025-11-22
>
> We thank the reviewer for the positive evaluation and helpful comments. We provide point-by-point responses to all the questions raised. In addition, we revise the original manuscript according to your suggestions and highlight the corresponding changes.
> # Weakness 1
>
> Thank you for pointing out this important issue. We fully agree that sparsity is a crucial metric, especially for tasks involving $\ell_1$-norm regularization, where the purpose of hyperparameter selection is precisely to promote sparse solutions. Motivated by this, we have supplemented our results with a detailed evaluation of the sparsity of the obtained solutions.
>
> The additional sparsity results have been included in Table NEW2, NEW4, NEW5 in the rebuttal revision, covering the corresponding experiments. These tables clearly show the sparsity patterns achieved by our method and enable a more complete and fair comparison across all sparsity-promoting tasks.
>
> # Weakness 2
>
> Thank you for the helpful suggestion. We have increased the font sizes and regenerated all figures to improve their readability. The updated figures are included in the rebuttal revision.
>
> # Question 1
>
> Alternating projections can in principle be applied to compute projections onto intersections of cones, but the resulting iterative schemes are typically complex and provide only limited accuracy. Moreover, alternating projections offer no convergence guarantee in this setting, which makes it difficult to guarantee convergence of the overall BLO algorithm.
>
> In contrast, ADMM offers clear advantages in terms of computational efficiency, convergence behavior, and implementability. The ADMM updates decompose naturally across the cone constraints, involve only simple closed-form projections, and come with well-established convergence guarantees under mild conditions. For these reasons, ADMM provides a more reliable and efficient mechanism for handling the cone-intersection structure.
>
> # Question 2
>
> Thank you for the suggestion. As clarified in Remark 3.4, Assumption 3.2 indeed implies the strong convexity of $\varphi$ under certain conditions. However, as we further verify in Appendix D.5.1, the strong convexity of $\varphi$ may only hold locally rather than globally for some problem instances. In addition, our theoretical framework and all convergence proofs do not rely on the strong convexity of $\varphi$.
>
> For these reasons, we state Assumption 3.2 in terms of Lipschitz smoothness only, which is the minimal condition required for our analysis and allows the assumption to cover a broader class of lower-level problems.
>
> # Question 3
>
> Thanks for your reminder. We have included detailed explanations for $\gamma>0$ in the rebuttal revision. The constant $\gamma$ serves as a penalty parameter and is taken to be a independent positive constant. We also validate the robustness of our algorithm under different values of $\gamma$ in the last part of Section 4.1. The corresponding results are provided in Table 9(b) of Appendix E.3.

---

> > ### Comment · Reviewer_vgK7 · 2025-11-26
> >
> > All my concerns have been adequately addressed. I keep my original rating and support publication.

---

> > > ### Author Response · Authors · 2025-11-26
> > >
> > > We sincerely appreciate the reviewer’s positive comments and are grateful for your support toward our work.

---

### Official Review · Reviewer_bhTF · 2025-11-03

**Soundness:** 3
**Presentation:** 3
**Contribution:** 1
**Rating:** 2
**Confidence:** 4

**Summary:**

This paper investigates nonsmooth bilevel hyperparameter optimization and develops a lower-level duality-based penalty framework that transforms the bilevel problem into a single-level form. The method leverages strong duality in the lower-level convex subproblem to design penalty-based algorithms that eliminate the need for explicit smoothing or nested differentiation. Two first-order schemes, LDP-PGD and LDP-ADMM, are proposed and analyzed. Theoretical results establish convergence guarantees under standard convexity and regularity assumptions. Experimental validation is conducted on small-scale convex and sparse logistic regression tasks.

**Strengths:**

The paper presents a rigorous and principled treatment of nonsmooth bilevel optimization under convexity assumptions. The use of lower-level duality to design a penalty reformulation is elegant and mathematically sound. The resulting single-loop algorithms are computationally efficient, Hessian-free, and accompanied by formal convergence proofs. The theoretical exposition is detailed, and the proofs appear correct under the given assumptions.

**Weaknesses:**

The method relies on convexity and strong duality of the lower-level problem, limiting its applicability to simple bilevel setups and excluding nonconvex inner problems common in modern hyperparameter optimization.

The experiments are small-scale and confined to convex tasks such as sparse logistic regression, providing limited evidence of scalability or robustness.

There is no comparison with recent single-loop or envelope-based approaches, and no sensitivity analysis of the penalty parameter or dual variable errors, both of which can strongly affect convergence and stability.

**Questions:**

1. How are the penalty multipliers $\lambda_i$ chosen or updated, and does convergence depend on their initialization?
2. Is there a bound on the stability or growth of the dual variables $(\xi, \rho)$ during optimization?
3. What is the per-iteration computational complexity of LDP-PGD relative to existing single-loop algorithms?
4. How does the penalty-induced bias compare analytically to Moreau-envelope smoothing methods?
5. Was the duality gap empirically verified to remain small in numerical experiments?

---

> ### Author Response · Authors · 2025-11-22
>
> We thank the reviewer for the constructive feedback. We provide point-by-point responses to all the questions raised. In addition, we revise the original manuscript according to your suggestions and highlight the corresponding changes.
>
> **Due to the character limit, our rebuttal is presented in two blocks**: the first block responds to the **weaknesses** you mentioned, and the second block responds to your **questions**.
>
> # Weakness 1
> >Convexity.
> - We would remark that it is generally unrealistic to solve **nonsmooth nonconvex** LL problems (e.g., involving neural networks), which further makes bilevel optimization (BLO) unsolvable, particularly for single-loop Hessian-free approaches. This limitation is **common** in bilevel optimization settings outside of neural network applications. We will further consider this question and the associated limitations in our future work.
> - Meanwhile, as explained in Remark 3.3, we emphasize that in our framework and algorithm the upper-level (UL) objective is allowed to be **nonconvex**. In fact, the setting that combines nonconvex UL objectives with convex but possibly nonsmooth LL problem covers a wide range of practical machine learning applications.
>
> >Strong duality.
> - Under the convexity of LL objective, the strong duality follows naturally from Slater's condition together with the affine constraints in (27), which arise directly from the data structure $Ax-b$. Moreover, we would emphasize that the linear form $Ax-b$ is a widely occurring structure in machine learning applications, rather than an artificial assumption introduced for analysis.
>
> In summary, the convexity and strong duality conditions of the LL problem in our analysis are reasonable and broadly applicable to a wide range of machine learning scenarios.
>
> # Weakness 2
>
> Thank you for the suggestion. In response to the concern regarding the dataset scale, we have added experiments on a substantially larger dataset. Specifically, we now include results on the Webspam dataset, which contains 350,000 samples and 16,609,143 features. The corresponding results have been incorporated into Figure NEW1 and Table 12 in Appendix E.4.3 of the rebuttal revision.
>
> # Weakness 3
>
> We appreciate the reviewer’s comment. However, both comparisons with recent single-loop Hessian-free approaches and sensitivity analyses of parameters are already included in our paper.
> > recent single-loop or envelope-based approaches
> - We would like to clarify that we have included the most recent single-loop Hessian-free methods, namely BiC-GAFFA [1] and MEHA [2], among our competitors. However, the publicly available implementations in [1] and [2] only cover a subset of problems that are compatible with our hyperparameter setting. For this reason, we report full and fair numerical comparisons on the three problem classes for which their code is applicable, specifically (sparse) group lasso, smoothed SVM, and sparse logistic regression, as presented in Table 2, Table 5, Table 6, Table 12, Figure 2, Figure NEW1, Figure 3. These settings coincide with the standard benchmarks commonly used in prior single-loop and envelope-based BLO works.
>
> > sensitivity analysis
> - We would also like to clarify that we conduct sensitivity experiments on the parameters for LDP-PGM and LDP-ADMM in the last part of Section 4.1. The corresponding results and explanations are provided in **Table 9 (Appendix E.3)**.
>
> [1] Yao et al. Overcoming lower-level constraints in bilevel optimization: A novel approach with regularized gap functions.
>
> [2] Liu et al. Moreau envelope for nonconvex bi-level optimization: A single-loop and hessian-free solution strategy.

---

> > ### Author Response · Authors · 2025-11-22
> >
> > In this block, we respond to all of the questions in a point-by-point manner.
> > # Question 1
> > - We would like to clarify that $\lambda_i$ are not penalty parameters. Instead, each $\lambda_i$ is the upper-level variable (hyperparameters) in our BLO formulation (3), serving as the coefficient of the norm regularization term. This form naturally arises in structural risk minimization, which is one of the most common and foundational frameworks in machine learning.
> >
> > - Meanwhile, the update of $\lambda_i$ in Algoithm 1 is described in detail as Eqs.(15)(41)(42), where it is updated by projection gradient descent. Likewise, the update of $\lambda_i$ in Algorithm 2 is explicitly given by Eq.(21), which follows the same structure as Eq.(15). Regarding initialization, since the upper-level objective is well defined for any positive regularization weight, it suffices to choose positive initial values with $\lambda_i^0>0$, as stated explicitly in line 1 of Algorithms 1 and 2. In our experiments, the specific initialization values of $\lambda_i^0$ are provided in the Experimental Settings section of each experiment in the appendix. The corresponding descriptions can be found at lines 1801, 1891, 1991, 2098, 2201, and 2258.
> > # Question 2
> >
> > Theoretically, our analysis establishes convergence of the function values rather than convergence of the iterates. Such analysis is typical in penalty-based methods for BLO [1,2,3], especially when the lower-level problem is nonsmooth. In such settings, boundedness of iterates is generally not guaranteed by theory, and this limitation is common across related works.
> >
> > The updates of $(\xi,\rho)$ are performed via epigraphic projected gradient descent. In the numerical experiments, each update involves an explicit projection onto a closed convex set, which empirically keeps the dual variables within a stable and bounded region throughout all experiments.
> >
> > [1] Yao et al. Overcoming lower-level constraints in bilevel optimization: A novel approach with regularized gap functions.
> >
> > [2] Liu et al. Moreau envelope for nonconvex bi-level optimization: A single-loop and hessian-free solution strategy.
> >
> > [3] Yao et al. Constrained bi-level optimization: Proximal lagrangian value function approach and hessian-free algorithm.
> > # Question 3
> >
> > We provide a summary of the per-iteration computational complexity along with comparisons to other single-loop methods in **Table NEW1** of the rebuttal revision.
> >
> > Our method handles the nonsmooth term via efficient projections, entirely avoiding the expensive inner subproblem. When the lower-level variable is $x \in R^{n}$ and the inputs are $A \in R^{d \times n}$ and $b \in R^{d}$, the Moreau-envelope-based methods incur a per-iteration cost of $\mathcal{O}(dn)$. In contrast, our method reduces the per-iteration complexity to $\mathcal{O}(n)$, as summarized in Table 4.
> >
> > Furthermore, when the lower-level variable is a matrix $x \in R^{m \times n}$ with rank of $r$, our approach only requires an economy-size SVD with cost $\mathcal{O}(mnr)$, whereas the Moreau-envelope-based methods require a full SVD with complexity $\mathcal{O}(mn \min\\{m,n\\})$.
> > # Question 4
> >
> > We thank the reviewer for this insightful question. Moreau-envelope based approaches first replace the original nonsmooth lower-level objective by its smoothed Moreau envelope. Even if the resulting smoothed bilevel problem is solved exactly, the obtained solution is only optimal for the smoothed problem, rather than for the original nonsmooth one.
> >
> > In contrast, our method does not introduce any additional smoothing approximation on the lower-level objective. We always work with the original nonsmooth lower-level problem and only introduce a penalty term to enforce the lower-level optimality conditions. Analytically, this is captured by the feasibility merit function $\phi_{fea}$ in (15). The convergence results in Theorems 3.5 and 3.7 show that the lower-level residual vanishes as $k\rightarrow\infty$. Hence, the penalty-induced error disappears any limit point of the generated sequence satisfies the original LL optimality conditions.
> >
> > Therefore, from an analytical perspective, the penalty-induced bias in our framework vanishes in the limit, while Moreau-envelope smoothing methods incur a structural approximation error controlled by the smoothing parameter.
> > # Question 5
> >
> > Thank you for the suggestion. The duality gap of the LL problem is indeed crucial for our framework, since it directly affects the validity of the penalty reformulation and the convergence of the feasibility merit function. To address the reviewer’s concern, we have added an empirical evaluation of the LL duality gap for our experiments in the rebuttal revision. The corresponding results are reported in **Table NEW2, NEW3, NEW4, NEW5 in Appendix E**. These results show that the LL duality gap vanishes over the iterations, and our method achieves the lowest gap across experiments.

---

### Official Review · Reviewer_jE3k · 2025-11-03

**Soundness:** 3
**Presentation:** 3
**Contribution:** 2
**Rating:** 4
**Confidence:** 2

**Summary:**

This paper tackles hyperparameter optimization via Bilevel Optimization. They propose a very elegant reformulation of the problem which exploits convex conjugation, and finally, they propose a practical single-loop Hessian-free optimization algorithm which achieves strong guarantees in the separable case and convincing practical performance. Unfortunately, the guarantees are way weaker in the non-separable case.

**Strengths:**

The reformulation of the problem exploiting convex conjugates is very elegant.

The experiments are convincing.

**Weaknesses:**

Better to explicitly introduce the data matrix $A_t$ and the labels $b_t$. In the smoothed Hinge loss line on page 2 the weights are denoted via $w$ rather than $x$ (as in the rest of the paper).

Theorem 3.5 does not indicate which is the optimal choice of $\underline{\beta}$. Moreover, I think that there is an abuse of big-oh notation. By this I mean that beyond the dependence on $K$, the authors should indicate how the rate depends on $\alpha_L$, $\alpha_P$, and the dimension of $x$.

Assumption 3.6 seems unnatural and should be guaranteed by the update rule itself.

The statement of Theorem 3.7 is rather weird because the constant M_e contains the learning rate $\gamma$. There is no guidance on how $\gamma$ should be chosen. Moreover, no finite time rates are provided.

Also, the assumption that $F_k$ remains bounded should be avoided.

**Questions:**

Can you provide the dependence on $\alpha_L$, $\alpha_P$, and dimension in Theorem 3.5? Moreover, indicate which is the best choice of $\underline{\beta}$. In particular, you should be able to characterize how large $\underline{\beta}$ should be taken to ensure that small merit functions also imply that the algorithm output is a solution of the problem of interest, which is (6).

Can Assumption 3.6 be avoided?

Is the boundness of $F_k$ necessary?

---

> ### Author Response · Authors · 2025-11-22
>
> We thank the reviewer for the detailed and helpful comments. We provide point-by-point responses to all the questions raised. In addition, we revise the original manuscript according to your suggestions and highlight the corresponding changes.
>
> **Due to the character limit, our rebuttal is presented in two response threads.**
> # Weakness 1: notations
>
> We thank the reviewer for pointing out the notation issue. We would like to clarify that the explicit forms of $A_t$ and $b_t$ are provided in Section 2 (lines 102–106). Meanwhile, we have replaced all occurrences of $w$ with $x$ in the smoothed SVM part in the rebuttal revision.
> # Weakness 2 & Question 1: Theorem 3.5
> > The constant $\underline{\beta}$.
>
> In our setting, it suffices for $\underline{\beta}$ to be a positive constant. Moreover, our algorithm is robust with respect to the choice of $\underline{\beta}$, as demonstrated in the experiments in Table 9 (Appendix E.3).
>
> > The notation $\mathcal{O}$ and the convergence rate.
>
> - We would clarify that the notation $\mathcal{O}$ is **standard and mathematically correct**. It characterizes the iteration complexity up to constants. In our proof in Appendix D.3, line 1526 shows that the rate is explicitly derived as
> $$
> \min_{0\leq k\leq K}\phi_{res}^k(z^{k+1})\leq O\left(\frac{(L_c+1/\underline{e})\underline{\beta}}{K^{1/2-p}}\right),
> $$
> where $L_c$ is defined in line 1495 as $L_c=\max\\{\frac{1}{\underline{\beta}}\alpha_L+\\|A_t\\|_2^2\alpha_p,\alpha_d+\\|A_t\\|_2^2,1\\}$. We observe that $L_c$, $\underline{\beta}$, $\underline{e}$ are all fixed constants. Therefore, it is equivalent to express the rate as $\mathcal{O}(1/K^{1/2-p})$. Meanwhile, we agree that your suggestion is more precise and we have revised the manuscript accordingly.
>
> - On the other hand, the convergence rate of $\phi_{fea}(z^k)$ as $O(1/K^p)$ is independent of all other constants, as proved in line 1538.
>
> - Moreover, we would emphasize that the iteration complexity is independent of the dimension of $x$. The dimension only affects the per-iteration computational cost. We summarize it in **Table NEW1** in our rebuttal revision.
>
> # Weakness 3 & Question 2: Assumption 3.6
>
> We appreciate the reviewer’s observation that Assumption 3.6 is rather strong. As discussed in [1,2], the convergence of **nonconvex nonsmooth** ADMM is highly challenging without imposing assumptions like Assumption 3.6, which is an open question. This assumption is **widely employed** in ADMM approaches [3,4,5,6]. In our numerical experiments, Assumption 3.6 is empirically satisfied automatically.
>
> # Weakness 4: Theorem 3.7
> > The constants $M_e$ and $\gamma$.
> - In Lagrangian function (19), $\gamma$ serves as a penalty parameter and is taken to be a independent positive constant. This is because ADMM is well known to be robust to the choice of $\gamma$, and convergence is guaranteed for any fixed $\gamma>0$ [1,2]. Moreover, this robustness is also reflected in our experiments, as shown in **Table 9(b)**. Therefore, although $M_e$ contains $\gamma$, it remains an independent constant and does not affect the convergence of LDP-ADMM.
>
> [1] Wang et al. Global convergence of ADMM in nonconvex nonsmooth optimization.
>
> [2] Lin et al. Alternating direction method of multipliers for machine learning.
>
> [3] Bai et al. An augmented lagrangian decomposition method for chance-constrained optimization problems.
>
> [4] Cui et al. Decision making undercumulative prospect theory: An alternating direction method of multipliers.
>
> [5] Shen et al. Augmented lagrangian alternating direction method for matrix separation based on low-rank factorization.
>
> [6] Xu et al. An alternating direction algorithm for matrix completion with nonnegative factors.
> > The convergence rate in Theorem 3.7.
> - We appreciate the reviewer’s interest in the convergence behavior of $\phi_{res}^k$. In the proof of Theorem 3.7 in Appendix D.4, we aim to establish the convergence of $\phi_{\text{res}}^k$.
> Combining the definition of $\phi_{\text{res}}^k$ with (71), we need to analyze the convergence properties of four terms: $\\|z^{k+1} - z^k\\|$, $\\|\mu^{k+1} - \mu^k\\|$, $\\|u_i^{k} - z^k\\|$, and $dist(-\sum\limits_{i=1}^{M+1}\mu_i^k, \mathcal{N}_{\mathcal{K}}(z^k))$.
>
>     Equation (64) implies that $\\|z^{k+1} - z^k\\|$, $\\|\mu^{k+1} - \mu^k\\|$, and $\\|u^k_i - z^k\\|$ exhibit a non-asymptotic convergence rate of $\mathcal{O}(1/\sqrt{k})$. For the last term, we obtain $-\mu_i^{k+1}\in\mathcal{N}_{\mathcal{K}_i\times\mathcal{K}_d^*}(u_i^{k+1})$ in line 829. However, based on the outer semi-continuity of normal cone, we can only establish the convergence property
>
>     $$\lim_{k\rightarrow\infty}dist(-\mu_i^{k},\mathcal{N}_{\mathcal{K}_i\times\mathcal{K}_d^*}(z^k))=0$$
>     in (69), without yielding an explicit non-asymptotic rate with respect to $k$.
>
>     In summary, we can only provide an asymptotic convergence and are unable to characterize the non-asymptotic convergence rate of $\phi_{\text{res}}^k$.

---

> ### Author Response · Authors · 2025-11-22
>
> # Weakness 5 & Question 3
>
> Thanks for your questions! In Theorems 3.5 and 3.7, we would clarify that we do not assume the function $F_k$ itself is bounded. Instead, we assume that the sequence $\\{F_k(z^k)\\}$ is bounded. This condition is used solely to establish the convergence of the equality constraint in the penalty reformulation, which ensures the feasibility of the penalized constraint.
>
> This condition is prevalent and standard in single-loop penalty-based methods for BLO without LL strong convexity, (cf. [7,Theorem 3.4], [8,Theorem 4.5], [9,Theorem 3.1]). In practice, this condition is empirically satisfied in numerical experiments.
>
> [7] Liu et al. Moreau envelope for nonconvex bi-level optimization: A single-loop and hessian-free solution strategy.
>
> [8] Yao et al. Overcoming lower-level constraints in bilevel optimization: A novel approach with regularized gap functions.
>
> [9] Yao et al. Constrained bi-level optimization: Proximal lagrangian value function approach and hessian-free algorithm.

---

> > ### Comment · Reviewer_jE3k · 2025-11-25
> >
> > Dear authors,
> >
> > Thanks a lot for your answer!
> >
> > I stand with my original evaluation of the paper. I think that the results in the paper relies on too strong assumtions which I encourage the authors to remove in their future revisions.
> >
> > In particular,
> >
> > -> Bound on the sequence $F(z_K)$.
> >
> > -> Assumption 3.6
> >
> > are too strong in the reviewer's opinion.
> >
> > Best,
> > reviewer

---

> ### Author Response · Authors · 2025-11-26
>
> Dear Reviewer jE3k,
>
> We sincerely appreciate your follow-up comments and fully acknowledge your concern. Nevertheless, we respectfully believe that these assumptions are standard, reasonable and technically justified in the analysis of nonconvex nonsmooth ADMM and the challenging nonconvex nonsmooth BLO.
> # Assumption 3.6
>
> In particular, after careful observations of our proofs, we realize that Assumption 3.6 only needs the boundedness of the dual sequence $\\{\mu^k\\}$ and we have accordingly refined Assumption 3.6 in the revised version to reflect this weaker and more precise condition.
> > **Assumption 3.6.** The sequence of multipliers $\\{\mu^k\\}$ is bounded.
>
> Moreover, we note that the inequality $\sum_{k=0}^\infty\\|\mu^{k+1}-\mu^k\\|^2<\infty$ **is already proved in our original analysis under boundedness of $\\{\mu^k\\}$ (Appendix D.4, Eq. (64), line 1623)**, so it does not need to be imposed as part of Assumption 3.6.
>
> - In the nonconvex nonsmooth ADMM literature, the boundedness of dual sequence is in fact **standard and typically assumed**. It is used explicitly in many existing works (cf. [1, Assumption 2][2, Proposition 2.2][3, Theorem 2.1][4, Assumption 2][5, Lemma III.2]).
>
> - Meanwhile, deriving boundedness of the dual sequence would usually **require substantially stronger assumptions**, e.g., global coercivity (cf. [6, Theorem 2][7, Theorem 1][8, Proposition 5]), error bounds [9, Lemma 3.4], particular algorithmic design [10, Proposition 4.2], etc. Such assumptions are more demanding and **unrealistic** in general nonconvex nonsmooth settings.
>
> - Several recent works [7,11] explicitly note that establishing convergence of nonconvex nonsmooth ADMM without such an assumption is still an **open problem**. More importantly, it also holds empirically and the dual variables remain stable in all experiments. **In this sense, Assumption 3.6 should be viewed as a standard and relatively light technical requirement rather than a strong modeling restriction.**
>
> In summary, the boundedness of the dual sequence is a reasonable and standard assumption that is fully aligned with our setting. Moreover, it is broadly applicable in machine learning scenarios, and does not affect either the convergence or the practical efficiency of Algorithm 2.
>
> [1] Cui et al. Decision making undercumulative prospect theory: An alternating direction method of multipliers.
>
> [2] Shen et al. Augmented lagrangian alternating direction method for matrix separation based on low-rank factorization.
>
> [3] Xu et al. An alternating direction algorithm for matrix completion with nonnegative factors.
>
> [4] Xiao et al. A unified framework for rank-based loss minimization.
>
> [5] Wang et al. Convergence of Bregman alternating direction method with multipliers for nonconvex composite problems.
>
> [6] Li et al. Global convergence of splitting methods for nonconvex composite optimization.
>
> [7] Wang et al. Global convergence of ADMM in nonconvex nonsmooth optimization.
>
> [8] LTK Hien et al. Inertial alternating direction method of multipliers for non-convex non-smooth optimization.
>
> [9] Zhang et al. A proximal alternating direction method of multiplier for linearly constrained nonconvex minimization.
>
> [10] Yang et al. Alternating direction method of multipliers for a class of nonconvex and nonsmooth problems with applications to background/foreground extraction.
>
> [11] Lin et al. Alternating direction method of multipliers for machine learning.
>
> # Bound on the sequence $F_k(z^k)$
>
> We would like to emphasize that the convergence of Algorithms 1 and 2 ($\phi_{res}^k$) do not rely on this assumption. It is only imposed for the analysis of the penalized feasibility term ($\phi_{fea}$). This assumption is **necessary and standard** in single-loop penalty-based methods for BLO without lower-level strong convexity (cf. [7,Theorem 3.4], [8,Theorem 4.5], [9,Theorem 3.1]).
>
> Relaxing this assumption in single-loop algorithms for nonconvex, nonsmooth BLO would also require **substantially stronger** analytical tools derived from structural properties, such as global error bounds, KL inequalities. To the best of our knowledge, establishing such results for nonconvex nonsmooth bilevel penalty methods **remains open**.
>
> More importantly, the convergence of $\phi_{fea}$ is confirmed by the reported duality-gap results across all experiments. In this sense, the boundedness of $\\{F_k(z^k)\\}$ should be regarded as an available technical condition, rather than any undesirable behavior of our algorithms.
>
> [12] Liu et al. Moreau envelope for nonconvex bi-level optimization: A single-loop and hessian-free solution strategy.
>
> [13] Yao et al. Overcoming lower-level constraints in bilevel optimization: A novel approach with regularized gap functions.
>
> [14] Yao et al. Constrained bi-level optimization: Proximal lagrangian value function approach and hessian-free algorithm.
>
> Finally, we would like to express our appreciation for your time and effort for reviewing our work!

---

### Official Review · Reviewer_carC · 2025-11-04

**Soundness:** 2
**Presentation:** 2
**Contribution:** 2
**Rating:** 4
**Confidence:** 3

**Summary:**

This paper addresses the bilevel hyper-parameter (for norm-based regularizers) optimization problem. It first proposes a series of penalty reformulation of the original bilevel problem. Under certain convexity and other regularity assumptions, the paper shows that the penalty reformulations lead to the solutions of the original problem. To solve for the reformulated problems, the paper considers the gradient descent approach with epigraphic projections. The proposed method is supported with a convergence guarantee and some experimental results.

**Strengths:**

1. This work is highly relevant, as it studies the important problem of optimizing hyper-parameters.
2. The paper is overall clear and well structured. The key theoretical points are made clear and well supported.
3. The proposed algorithm is single-loop, which is more suited for computationally expensive machine learning applications as compared to a double-loop structure.

**Weaknesses:**

My major concern on this work is its general effectiveness:

1. The problem setup, and the key assumptions needed for the algorithm to work, or for the theories to hold, is somewhat restrictive to me. The paper considers a problem in (4), which assumes a linearly structured data and a restricted set of nonlinear loss which excludes some most frequently used loss like cross-entropy. The proposed algorithm is tightly built on this set of assumptions and the theoretical result has a relatively strong dependence on them.

2. If the theoretical results are hard to generalize to broader settings, it is then natural to look for empirical evidence for the algorithm's effectiveness beyond the made assumptions. However, it seems the paper does not provide ample evidence for this. It might be beneficial to test the algorithm in more large-scaled problems that do not admit a convex/linear structure.

3. Missing baselines in experiments: In section 4.2, the experiments miss arguably the most important baseline that is the random and the grid-search-based hyper-parameter optimization. Since those methods are the most commonly adopted approach in practice, it might be beneficial to compare with them.

**Questions:**

In addition to the questions posed in the weakness section, I also have the following ones:
1. What is the overall computational complexity of this algorithm? How does it compare to other penalty-based BLO methods?
2. The proposed algorithm directly optimizes on the validation set, which is usually prohibited in machine learning practice since the validation set is supposed to be independent of the training process as a generalization check. Directly putting optimization pressure on the validation set makes it fail its role as the generalization check.

**Details Of Ethics Concerns:**

No ethics concerns.

---

> ### Author Response · Authors · 2025-11-22
>
> We thank the reviewer for the careful reading and constructive feedback. We provide point-by-point responses to all the questions raised. In addition, we have revised the original manuscript according to the reviewer’s suggestions and highlighted all the corresponding changes.
>
> **Due to the character limit, our rebuttal is presented in two blocks**: the first block responds to the **weaknesses** you mentioned, and the second block responds to your **questions**. **All references cited in our responses are listed at the end of the rebuttal in the second block.**
> # Weakness 1 & 2
>
> > **Problem setup: linearly structured data and a restricted set of nonlinear loss**
>
> Thank you for pointing this out!  We would like to clarify that Eq.(4) is not introduced as an assumption in our work. It represents a widely observed structure in many machine learning applications.
>
> Our reformulation is based on lower-level duality. Specifically, strong duality holds when the constraints in Eq.(27) are affine, i.e., in linear cases of the form $Ax-b$. However, strong duality in the LL problem may no longer hold in nonlinear structured data, which makes our framework inapplicable in such cases. This limitation is **common** in bilevel optimization settings outside of neural network applications.
>
> > **Assumptions 3.1, 3.2 and 3.6 in Section 3**
>
> >> **Lipschitz smoothness in Assumptions 3.1 and 3.2**.
>
> - While our current analysis indeed requires Lipschitz smoothness of the loss functions, our framework and algorithm remain valid for other common loss functions in machine learning as listed in examples Table 1 and our experiments, including least squares loss, huber loss, logistic loss, etc. Meanwhile, smoothness is also a necessary condition for Hessian-free algorithms, and the assumption that the loss function is Lipschitz smooth has been consistently adopted in the existing literature [1,2,3].
>
> >> **Convexity in Assumption 3.2**.
> - We would remark that it is generally unrealistic to solve **nonsmooth nonconvex** LL problems (e.g., involving neural networks), which further makes bilevel optimization (BLO) unsolvable, particularly for single-loop Hessian-free approaches [1,3,4]. We will further consider this question and the associated limitations in our future work.
>
> - Meanwhile, as explained in Remark 3.3, we emphasize that in our framework and algorithm the upper-level (UL) objective is allowed to be **nonconvex**. In fact, the setting that combines **nonconvex UL objectives** with **convex but possibly nonsmooth LL problem** covers a wide range of practical machine learning applications.
>
> >> **Assumption 3.6**.
>
> - Assumption 3.6 is aligned with standard conditions widely adopted in ADMM-based methods [5,6,7,8]. As discussed in [9,10], the convergence of nonconvex nonsmooth ADMM is highly challenging without imposing assumptions like Assumption 3.6, which is an open question. In our numerical experiments, we observe that Assumption 3.6 is empirically satisfied automatically.
>
> # Weakness 3
>
> Thank you for raising this point. Indeed, the random search (RS) and grid search (GS) hyperparameter optimization baselines are actually included in our experiments in Section 4.2. Specifically:
>
> - Elastic net: we present the results of RS and GS in **Table 10** in Appendix E.4.1. However, both GS and RS require substantially higher runtime than the other methods in Table 10, so we do not plot their curves in Figure 1.
> - Smoothed SVM: RS is included as a baseline in **Figure 2**. However, GS converges too slowly and requires considerably longer runtime in this setting, so we do not include it in Figure 2.
> - Sparse logistic regression: GS and RS are hyperparameter search procedures rather than iterative optimization methods. Their runtimes are significantly higher and their performance exhibits large discontinuities across evaluations. For these reasons, GS and RS are not suitable for inclusion in the learning-curve plots.

---

> ### Author Response · Authors · 2025-11-22
>
> # Question 1: computational complexity
>
> Thanks for your question! We compare the computation cost of our method and other penalty-based methods [1,2,3,4,11,12,13] as follows. Moreover, we provide a summary of the computational costs in **Table NEW1** in our rebuttal revision.
>
> - PBGD [11] and BLOCC [12] require the LL problem to be strongly convex and smooth, while BOME [4] and the first-order penalty method [13] still requires the smoothness of the LL objective. These methods are not suitable for nonsmooth setting.
>
> - Moreau–envelope-based methods [1,2,3] are suitable for BLO with nonsmooth LL problem. These approaches handle the nonsmooth term by smoothing it through the Moreau envelope. Consequently, each iteration involves a gradient-descent (GD) step for the Moreau envelope and a GD step on the penalty objective. In contrast, our method handles the nonsmooth term via efficient projections, entirely avoiding the expensive Moreau-envelope subproblem.
>
> As a result, when the lower-level variable is $x \in R^{n}$ and the inputs are $A \in R^{d \times n}$ and $b \in R^{d}$, the Moreau-envelope-based methods incur a per-iteration cost of $\mathcal{O}(nd)$. In contrast, our method reduces the per-iteration complexity to $\mathcal{O}(n)$, as summarized in Table 4.
>
> Furthermore, when the lower-level variable is a matrix $x \in R^{m \times n}$ with rank of $r$, our approach only requires an economy-size SVD with cost $\mathcal{O}(mnr)$, whereas the Moreau-envelope-based methods require a full SVD with complexity $\mathcal{O}(mn \min\\{m,n\\})$.
>
> # Question 2: validation and training processes
>
> Thanks for your question!
> - In our bilevel hyperparameter optimization, the validation is actually kept separate from the training process. In the bilevel formulation (3), the lower-level (LL) problem corresponds to the training procedure and is optimized solely on the training set, while the upper-level (UL) objective is defined on the validation set.
> - After the LL problem (the training procedure) is solved, we then optimize the UL problem (the validation objective). This sequential structure is fully consistent with the standard machine learning workflow.
> - This training/validation approach and bilevel structure are widely adopted in the existing literature [1,2,3,4]
>
>
> [1] Yao et al. Overcoming lower-level constraints in bilevel optimization: A novel approach with regularized gap functions.
>
> [2] Liu et al. Moreau envelope for nonconvex bi-level optimization: A single-loop and hessian-free solution strategy.
>
> [3] Yao et al. Constrained bi-level optimization: Proximal lagrangian value function approach and hessian-free algorithm.
>
> [4] Liu et al. Bome! bilevel optimization made easy: A simple first-order approach.
>
> [5] Bai et al. An augmented lagrangian decomposition method for chance-constrained optimization problems.
>
> [6] Cui et al. Decision making undercumulative prospect theory: An alternating direction method of multipliers.
>
> [7] Shen et al. Augmented lagrangian alternating direction method for matrix separation based on low-rank factorization.
>
> [8] Xu et al. An alternating direction algorithm for matrix completion with nonnegative factors.
>
> [9] Wang et al. Global convergence of ADMM in nonconvex nonsmooth optimization.
>
> [10] Lin et al. Alternating direction method of multipliers for machine learning.
>
> [11] Shen et al. On penalty-based bilevel gradient descent method.
>
> [12] Jiang et al. A primal-dual-assisted penalty approach to bilevel optimization with coupled constraints.
>
> [13] Lu et al. First-order penalty methods for bilevel optimization.

---

### Author Response · Authors · 2025-12-03

Dear AC and Reviewers,

Since the scores are being reverted to their pre-discussion state and no further reviewer discussions or public comments are allowed, we would like to provide a brief summary of the changes we made and how reviewers responded.

Reviewers recognize the novelty of our work, the rigorous theory for nonsmooth nonconvex bilevel optimization and the comprehensive experimental valuation. No major errors are identified. Comments mainly concern suggesting future extensions, clarifying assumptions and discussing our method’s advantages, all of which are addressed in our rebuttal.

1. We show that our assumptions are reasonable, broadly applicable to machine learning scenarios, and aligned with standard conditions widely adopted in single-loop penalty-based methods for nonsmooth nonconvex bilevel optimization.
2. We highlight the computational gains. Our method avoids smoothing strategies and off-the-shelf solvers and tackles nonsmoothness directly via efficient projections, which greatly reduces per-iteration cost and runtime. Therefore, we achieve computationally superior to existing single-loop first-order penalty-based BLO algorithms.
3. We include additional reports in the experimental section, and these supplementary results further demonstrate the reliability of our method.

In the revised version of the paper, we present additional experimental results across multiple tables and figures in Appendix E and provide a detailed illustration of the computational advantages of our method with Table NEW1 in Appendix C. We also provide additional remarks that clarify the validity of all our assumptions. All revisions are marked in blue in the rebuttal version.

These revisions collectively resolved the reviewers’ questions. Reviewer vgK7 express continued strong support for our work. Reviewer jE3k engaged with us in a follow-up discussion. We thoroughly clarified that both assumptions are standard, reasonable and technically justified. These assumptions are widely adopted in the literature on single-loop penalty-based methods and ADMM-based approaches for nonsmooth, nonconvex settings. Although they did not provide any further feedback, we are confident that their concerns have been fully addressed.
Reviewer carC and Reviewer bhTF have not yet replied, but we believe our responses adequately addressed their concerns as well.

We hope that this summary of changes will assist the AC in making their decision. Please inform us if any additional questions arise.

Thank you for handling our paper.

Best,

Authors.

---

### Meta-Review · Area_Chair_ErhA · 2026-01-05

**Summary:**

This paper presents a duality-based penalization method for hyperparameter bilevel optimization (BLO) with nonsmooth lower-level (LL) problems. The authors reformulate the LL problem using its dual representation and subsequently develop single-loop, first-order algorithms (LDP-PGD and LDP-ADMM) based on this reformulation, providing convergence guarantees under a set of assumptions. The reviewers (bhTF,jE3k) recognized the novelty of using a dual approach to handle the lower-level problem, and (carC，vgK7) acknowledged that the single-loop algorithm design is well-suited for machine learning applications. However, several concerns were raised:

Theoretical aspects: Some reviewers (carC, jE3k) pointed out the strong assumptions in the paper (such as bounded dual/multiplier sequences in ADMM and the boundedness of certain sequences for feasibility analysis). Although the authors mitigated these assumptions in their response, arguing that these are standard in optimization theory, the assumptions remain relatively strong.

Empirical validation: Multiple reviewers(carC,bhTF) mentioned that the experiments are limited to small-scale problems. The authors subsequently expanded their experimental analysis to higher-dimensional datasets. However, these experiments still focus on relatively simple models, such as elastic net, SVM, and logistic regression, without providing validation in more complex learning scenarios.

Overall, the paper contributes to a specific class of structured hyperparameter optimization problems, proposing a duality- and penalty-based reformulation for bilevel optimization tasks. However, this approach is limited to
bilevel problems with linear/convex lower-level structure, and it relies on relatively strong assumptions. As a result, the applicability of the algorithm to broader machine learning problems is limited. The reviewers’ opinions were mixed-to-borderline, and I personally lean toward a negative evaluation and recommend rejection.

**Reviewer Concerns:**

**Addressed:**

Reviewer carC: The rebuttal resolved the concern regarding the missing comparison with baseline methods such as random search (RS) and grid search (GS) by clarifying that both RS and GS were already included and reported in the appendix. Additionally, new experiments were conducted on a larger dataset (Webspam), and additional tables and figures were provided in the appendix in response to concerns about scalability. The rebuttal also clarified issues regarding computational complexity and the experimental setup.

Reviewer jE3k: The rebuttal addressed the concerns regarding certain parameters (specifically \underline{\beta}) and provided clarifications.

Reviewer bhTF: The rebuttal tackled concerns about scalability by adding new experiments on the larger Webspam dataset. It also clarified the absence of comparisons with recent single-loop or envelope-based approaches by noting that such comparisons were, in fact, included with two recent BLO methods of this type. Furthermore, the rebuttal offered explanations for the variables (\lambda) and (\xi, \rho), as well as computational complexity. Concerns about the duality gap were addressed through an additional empirical evaluation.

Reviewer vgK7: The rebuttal addressed concerns about solution sparsity by including corresponding results in the experiments. It also provided more detailed explanations to clarify remaining questions. The reviewer acknowledged that all concerns had been addressed.

**Outstanding:**

The main remaining issue is the scope of applicability, raised by reviewers carC, jE3k, and bhTF. The method fundamentally relies on convexity and strong duality of the lower-level problem, and therefore does not extend to nonconvex inner problems that are common in modern hyperparameter optimization. Relatedly, although the added large-scale experiment improves coverage, empirical validation is still largely confined to relatively simple models (elastic net, SVM, logistic regression), leaving performance in more complex learning pipelines untested.

Finally, concerns about strong assumptions, raised by reviewers carC, and jE3k persist—particularly boundedness-type conditions (bounded dual/multiplier sequences in ADMM, boundedness of certain sequences for feasibility analysis). While the authors weaken Assumption 3.6 to bounded multipliers and argue that such assumptions are standard, reviewers may still view them as restrictive rather than fully eliminated.

**Reviewer Scores:**

**Reviewer carC (Score: 4  -> Est. unchanged):**

Reviewer carC would maintain the same overall score (4: marginally below acceptance). The rebuttal addressed concerns about the missing baseline comparisons and scalability by adding new experiments on a larger dataset. As a result, the reviewer's confidence in the paper might increase slightly (e.g., from 3 → 4), as part of their concerns have been resolved. However, the rebuttal did not fully address the reviewer’s core concern raised in Weakness 1 and 2, namely that the applicability of the proposed method is restricted to special cases where the lower-level problems are convex and linearly structured. Consequently, the reviewer’s original concerns remain.

**Reviewer jE3K (Score: 4  -> Est. unchanged):**

Reviewer jE3K would maintain the same overall score (4: marginally below acceptance). The reviewer remained engaged in the discussion and reiterated concerns about the strength of the assumptions despite the authors’ clarifications.

**Reviewer bhTF (Score: 2 -> Est. 4):**

Reviewer bhTF would likely raise the score from (2: reject, not good enough) to (4: marginally below acceptance). The rebuttal addressed the concerns about scalability by adding experimental results on higher-dimensional datasets. Additionally, the reviewer noted the lack of comparisons with recent single-loop or envelope-based approaches. The authors clarified that such comparisons were included with two recent BLO methods of this type. Regarding specific questions, the reviewer asked about the choice of the parameter ($\lambda$), which the authors clarified is an optimization variable rather than a penalty parameter. The reviewer also raised concerns about the duality gap, which the authors addressed with supplementary experiments. For the remaining questions, the authors provided detailed explanations. Overall, the responses effectively addressed most of the reviewer’s concerns, and I think this justifying a score increase to 4.

**Reviewer vgK7 (Score: 8  -> Est. unchanged):**

Reviewer vgK7 would maintain the high overall score (8: accept, good paper (poster)). The reviewer remained actively engaged in the discussion and upheld their original positive evaluation, ultimately expressing clear support for the paper’s acceptance.

---

### Decision · Program_Chairs · 2026-01-26

Reject